# Biomolecular condensate drives polymerization and bundling of the bacterial tubulin FtsZ to regulate cell division

Beatrice Ramm [1,2,4] ✉, Dominik Schumacher [3,4] ✉, Andrea Harms[3], Tamara Heermann [1], Philipp Klos[3], Franziska Müller[3], Petra Schwille [1,5] ✉ & Lotte Søgaard-Andersen [3,5] ✉

Cell division is spatiotemporally precisely regulated, but the underlying mechanisms are incompletely understood. In the social bacterium *Myxococcus xanthus*, the PomX/PomY/PomZ proteins form a single megadalton-sized complex that directly positions and stimulates cytokinetic ring formation by the tubulin homolog FtsZ. Here, we study the structure and mechanism of this complex in vitro and in vivo. We demonstrate that PomY forms liquid-like biomolecular condensates by phase separation, while PomX self-assembles into filaments generating a single large cellular structure. The PomX structure enriches PomY, thereby guaranteeing the formation of precisely one PomY condensate per cell through surface-assisted condensation. In vitro, PomY condensates selectively enrich FtsZ and nucleate GTP-dependent FtsZ polymerization and bundle FtsZ filaments, suggesting a cell division site positioning mechanism in which the single PomY condensate enriches FtsZ to guide FtsZ-ring formation and division. This mechanism shares features with microtubule nucleation by biomolecular condensates in eukaryotes, supporting this mechanism's ancient origin.

Although bacterial cells generally lack membrane-bounded organelles, they are spatiotemporally highly organized with proteins localizing dynamically to distinct subcellular locations to spatially restrict their activities[1,2]. However, our understanding of how this spatiotemporal organization is accomplished is incomplete. Recently, biomolecular condensates formed by phase separation have emerged as an important mechanism to spatially organize intracellular processes in eukaryotic cells[3–6], while they are only beginning to be identified and explored in bacteria[7].

The formation of biomolecular condensates in bulk solution involves the concentration-dependent, switch-like demixing of proteins from solution into high-density condensates that coexist with a remaining dilute protein phase above a critical saturation concentration ($C_{sat}$)[3–6]. Alternatively, several studies observed that condensates seemed to be nucleated through a form of surface-assisted condensation[8–12]. This might arise from a phenomenon termed prewetting[13,14], in which the binding of a protein to a surface results in a local enrichment, thereby stimulating condensate formation on the surface. As the bulk concentration is below $C_{sat}$, condensate formation is restricted to the surface, which could provide a means of spatiotemporal regulation. If the bulk concentration is above $C_{sat}$, condensates formed in the bulk can also associate with the surface to wet it[15,16].

For all condensates, the interface between the two phases forms a boundary that serves as a selective barrier for some molecules but not

[1]Department of Cellular and Molecular Biophysics, Max Planck Institute of Biochemistry, Am Klopferspitz 18, 82152 Martinsried, Germany. [2]Department of Physics, Princeton University, Princeton, NJ 08544, USA. [3]Department of Ecophysiology, Max Planck Institute for Terrestrial Microbiology, Karl-von-Frisch Str. 10, 35043 Marburg, Germany. [4]These authors contributed equally: Beatrice Ramm, Dominik Schumacher. [5]These authors jointly supervised this work: Petra Schwille, Lotte Søgaard-Andersen. ✉e-mail: bramm@princeton.edu; dominik.schumacher@mpi-marburg.mpg.de; schwille@biochem.mpg.de; sogaard@mpi-marburg.mpg.de

for others, resulting in selective enrichment of so-called client proteins and/or RNA molecules[3–6]. In this way, biomolecular condensates can enhance chemical reactions, sequester molecules, or act as hubs to nucleate microtubule or actin polymerization[8,9,17–21].

Protein condensates form by cumulative, specific, transient, low-affinity interactions between multivalent proteins that may self-interact via homotypic interactions or interact with other proteins via hetero-typic interactions using folded domains, intrinsically disordered regions (IDRs), and/or repetitive protein motifs in low-complexity regions[4]. Above the $C_{sat}$, the multivalent interactions give rise to large non-stoichiometric networks of interacting proteins that lead to con-densate formation[3–6]. Many condensates have liquid-like properties and are of spherical shapes due to the surface tension acting to minimize the surface area[3–6]. Such liquid-like condensates can be highly dynamic and exchange molecules with the dilute phase and undergo internal reorganization, fusion, fission, and/or disintegration[3–6].

Positioning of the cell division site in bacteria is spatiotemporally precisely regulated. Bacterial cell division depends on the localization of the tubulin-homolog FtsZ at the incipient division site to form, in a GTP-dependent manner, the so-called Z-ring[22,23], a ring-like structure of short treadmilling FtsZ filaments[24,25]. Subsequently, the Z-ring directly or indirectly recruits other proteins that help to execute cytokinesis[22,23]. The systems that spatiotemporally control cell division accomplish their function by directly interacting with FtsZ to modulate its GTP-dependent polymerization. Negative regulation systems such as the MinC/MinD/MinE system in *Escherichia coli* and the MipZ/ParB system in *Caulobacter crescentus* position an FtsZ inhibitor at the cell poles, whereby Z-ring formation is restricted to midcell[26,27]. By con-trast, MapZ in the pathogen *Streptococcus pneumoniae*[28,29] and the PomX/PomY/PomZ proteins in the rod-shaped cells of the social, predatory bacterium *Myxococcus xanthus* localize at midcell to directly guide and promote Z-ring formation and cell division between two segregated chromosomes[30–32].

The PomX/PomY/PomZ system is a representative of a large group of systems that include the MinC/MinD/MinE system for cell division placement and the ParA/ParB system for chromosome and plasmid segregation, and in which a ParA/MinD-type ATPase together with its cognate ATPase activating protein (AAP) positions macro-molecular structures in bacteria[33]. In the PomX/PomY/PomZ system, the AAPs PomX and PomY form a single large cytoplasmic complex that is recruited to the nucleoid by the ATP-bound dimeric ParA/MinD-type ATPase PomZ[31,32]. PomX, PomY and PomZ are present in multiple copies in this complex, which has an average size of ~15MDa and is visible by widefield fluorescence microscopy with the three proteins colocalizing[31]. After cell division, the PomX/PomY/PomZ complex is bound to the nucleoid close to the new cell pole[31]. Subsequently, it translocates by biased random motion to midcell, where it switches to constrained motion and stimulates Z-ring formation by an unknown mechanism[31]. Upon division, the PomX/PomY/PomZ complex was suggested to undergo fission with both daughters "receiving" part of the complex[31]. The three Pom proteins interact in all pairwise combi-nations and have distinct functions[31,34]: As shown by negative stain transmission electron microscopy (TEM), PomX self-assembles to form filaments in vitro[31,34]. In vivo, PomX assembles into a single cel-lular cluster independently of PomY and PomZ and stimulates the assembly of the PomY and PomZ clusters by direct interaction to generate the PomX/PomY/PomZ complex[31,34]. PomY interacts directly with FtsZ and is essential for its recruitment to the incipient division site[31]. PomZ interacts directly with PomX and PomY and associates the PomX/PomY complex with the nucleoid[31]. By stimulating ATP hydro-lysis by DNA-bound PomZ, PomX and PomY promote cluster translocation[31,34].

While PomX/PomY/PomZ cluster translocation is well-understood based on experiments and theory[31,35,36], the structure of the PomX/PomY/PomZ complex and its function in Z-ring formation remain unclear. Here, by focusing on PomX and PomY and combining in vivo and in vitro approaches, we show that PomY forms liquid-like bio-molecular condensates by phase separation. These condensates selectively enrich FtsZ thereby nucleating GTP-dependent FtsZ poly-merization and bundle FtsZ filaments. The filamentous PomX structure serves as a scaffold to locally enrich PomY, thereby nucleating phase separation by PomY via surface-assisted condensation and ensuring the formation of precisely one PomY condensate per cell that guides Z-ring formation and cell division.

## Results

### PomX and PomY clusters are non-stoichiometric and scale with cell size

To understand the structure and function of the PomX/PomY/PomZ complex in vivo, we focused on PomX and PomY because they form a single cytoplasmic complex independently of PomZ that is able to stimulate Z-ring formation, while the PomZ ATPase is important for translocation of this complex[31]. In strains expressing active mCherry (mCh) fusions, i.e., mCh-PomX and PomY-mCh, at native levels, we observed that cluster dimensions were above the diffraction limit of high-resolution structured illumination microscopy (SIM) (Fig. 1a, b, Supplementary Fig. 1a–c). Individual cells generally contained a single cluster, but smaller cells sometimes lacked a visible cluster (Supple-mentary Fig. 1d). For both proteins, cluster shape varied from spherical to spheroidal (Fig. 1a, b). Generally, mCh-PomX clusters were more elongated than PomY-mCh clusters while the short axes were com-parable (Fig. 1b). Accordingly, the mean aspect ratio of mCh-PomX clusters was significantly higher than that of PomY-mCh clusters (Fig. 1b). Interestingly, for both proteins, the spherical clusters tended to be present in shorter cells while longer cells contained the larger spheroidal clusters (Fig. 1c). Because the short cluster axes were independent of cell length (Fig. 1c), we conclude that cluster shape varies from spherical to spheroidal and cluster size scales with cell size.

We next examined the size-scaling of the mCh-PomX and PomY-mCh clusters by quantitative widefield time-lapse microscopy and using fluorescence intensity as a metric for cluster size. All cells had mCh-PomX and PomY-mCh clusters at midcell immediately before cell division (Fig. 1d). For mCh-PomX, most daughters contained a cluster after cell division that, generally, was smaller than the one of the mother (Fig. 1d). Because the summed fluorescence intensities of the daughter clusters added up to $87 \pm 15\%$ of the mother's cluster (Fig. 1d), we conclude that the mCh-PomX clusters undergo fission during cell division. Sometimes the fission process was asymmetric with two daughters receiving clusters of unequal sizes (Fig. 1d). Occasionally, a daughter did not "receive" a mCh-PomX cluster; such cells subse-quently formed a cluster de novo (Fig. 1d). mCh-PomX clusters essentially doubled in size over the cell cycle as did total cellular fluorescence, and cytoplasmic fluorescence (Fig. 1d, e). By normalizing fluorescence intensities to the respective areas to obtain a metric for protein concentration (referred to as fluorescence concentration), we found that the fluorescence concentration of mCh-PomX in the cell, the cytoplasm, and the cluster remained constant during the cell cycle (Fig. 1f). Remarkably, most PomY-mCh clusters disappeared during cell division resulting in daughters with only diffuse PomY-mCh signals that added up to $89 \pm 11\%$ of the mother's total cellular fluorescence (Fig. 1d), supporting that PomY-mCh in clusters is not proteolytically degraded during division, but rather that the clusters undergo disin-tegration during division. Within hours of a cell division, a single weak cluster emerged de novo and grew in size before the subsequent cell division, resulting in cycles of cluster growth and disintegration (Fig. 1d). For PomY-mCh, total cellular fluorescence and cytoplasmic fluorescence also essentially doubled over the cell cycle, while cluster fluorescence intensity (and therefore cluster size) increased ~4-fold; this increase represents a minimum estimate because it is calculated from the first appearance of a cluster until the subsequent cell division

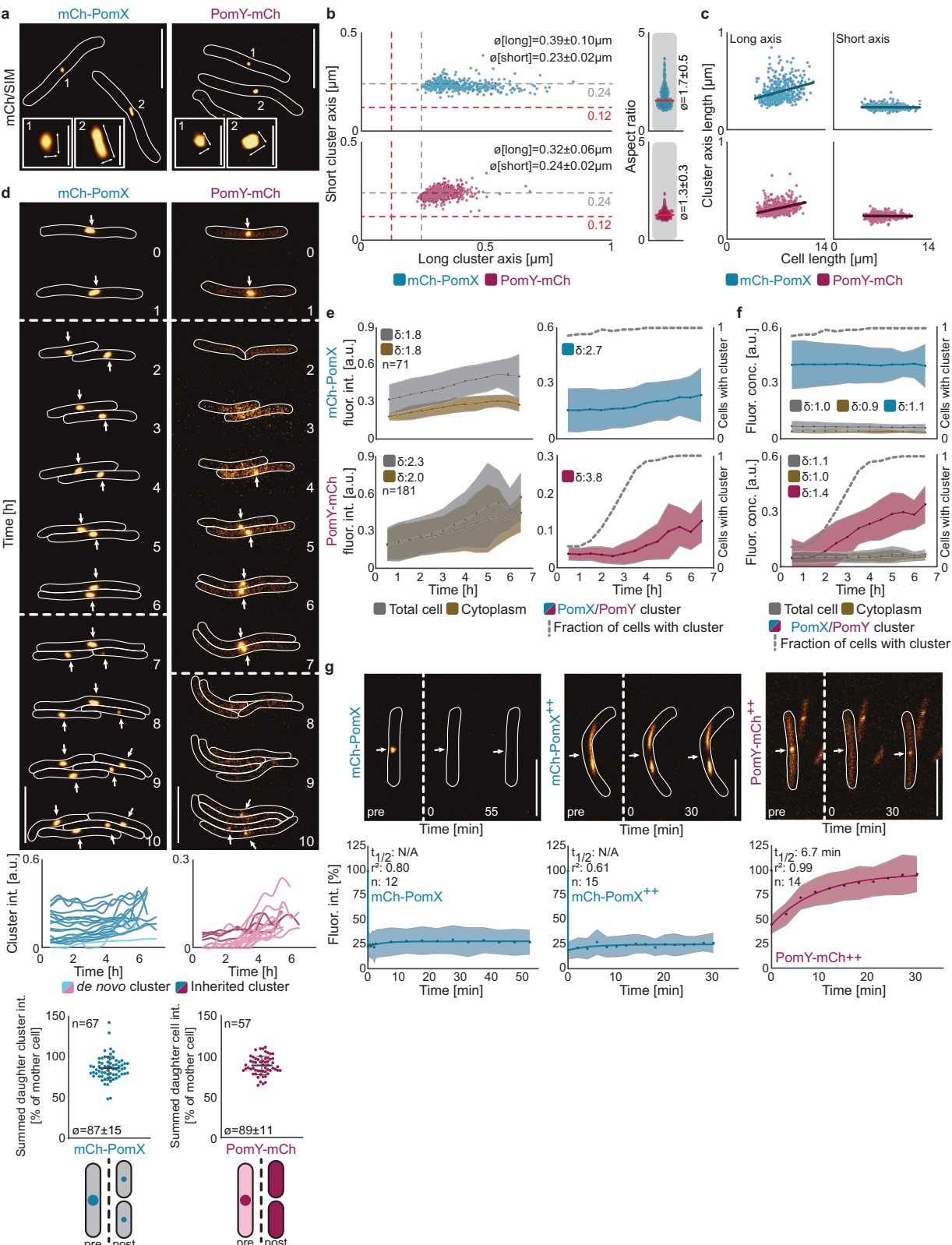

(Fig. 1e). Moreover, the cellular and cytoplasmic PomY-mCh fluorescence concentrations remained constant over the cell cycle, while cluster fluorescence concentration increased ~40% (Fig. 1f); as above, this represents a minimum estimate.

In snapshot images of >1000 cells and using cell length as a proxy for the cell cycle stage, we confirmed the results from the time-lapse microscopy for total cellular, cytoplasmic, and cluster fluorescence as well as for fluorescence concentrations (Supplementary Fig. 2). Based on these experiments, we calculated a mean enrichment factor in a cluster relative to the cytoplasm of 12.1 for mCh-PomX and 4.1 for PomY-mCh. Based on the estimated PomX and PomY concentrations in wild-type (WT) cells[31], the mean mCh-PomX concentration in a cluster and the cytoplasm is 1.0 µM and 0.08 µM, respectively, and the corresponding PomY-mCh concentrations 1.4 µM and 0.3 µM.

**Fig. 1 | PomX and PomY clusters are structurally flexible, non-stoichiometric and scale in size with cell size. a** SIM images of mCh-PomX or PomY-mCh clusters. Numbers indicate the insets and arrows the long and short cluster axes used for quantification. Scale bars, 5 μm, 0.5 μm in insets. **b** Quantification of cluster axes and cluster aspect ratios based on SIM images. Red and gray stippled lines in the left panels indicate the theoretical resolution of SIM and epifluorescence microscopy, respectively; red solid lines in right panels, median. $n > 400$ clusters in at least two independent replicates. ø, mean ± standard deviation (STDEV). **c** Quantification of cluster axes as function of cell length based on SIM images. Dark-colored lines indicate linear regression. Data as in (**b**). **d** Epifluorescence time-lapse microscopy of mCh-PomX or PomY-mCh clusters. Stippled lines, cell divisions. Arrows point to mCh-PomX and PomY-mCh clusters. Scale bars, 5 μm. Middle panels, quantification of mCh-PomX and PomY-mCh cluster intensity in 25 representative cells from birth to division. Dark-, and light-colored lines, intensity of clusters inherited from the mother and de novo synthesized, respectively. Lower panels, (left) quantification of mCh-PomX cluster fluorescence intensity pre- and post-division in mothers and their corresponding daughters; (right) quantification of total cellular PomY-mCh fluorescence intensity pre- and post-division in mothers and their corresponding daughters. $n$, number of divisions analyzed; error bars, mean ± STDEV.
**e** Fluorescence intensity of mCh-PomX or PomY-mCh over the cell cycle. Colored dotted lines, mean and light-colored areas STDEV. δ-values, fold increase between birth and division. $n$, number of cells analyzed. **f** Fluorescence concentrations of mCh-PomX or PomY-mCh over the cell cycle. Same cells as in (**e**). **g** FRAP experiments. Cluster fluorescence intensity in a region of interest (ROI) of 0.52 μm in diameter before bleaching was set to 100%. Colored dots, mean and light-colored areas, STDEV. Dark lines, recovery fitted to a single exponential equation. ++, proteins were moderately overexpressed from the *pilA* promoter. Stippled lines, bleaching events marked by white arrows. $n$, number of bleaching events. Scale bars, 5 μm. Unless otherwise mentioned, mCh-PomX and PomY-mCh were expressed at native levels.

Because the PomX and PomY clusters grow in size with time, we asked whether the proteins dynamically exchange with the cytoplasm. In fluorescence recovery after photobleaching (FRAP) experiments, neither mCh-PomX clusters at native protein levels nor when moderately overexpressed showed recovery (Fig. 1g). PomY-mCh signals were too weak for FRAP analyses at native protein levels; however, when PomY-mCh was moderately overexpressed, PomY-mCh in clusters dynamically exchanged with the cytoplasm with a half-maximal recovery time ($t_{1/2}$) of 6–7 min (Fig. 1g).

In summary, PomX and PomY clusters share some characteristics, i.e., cluster shape varies from spherical to spheroidal and cluster size scales directly with cell size despite constant cytoplasmic and cellular concentrations. However, PomX clusters undergo fission during division and then regrow, while PomY clusters disintegrate and then reform de novo followed by growth. Moreover, PomX in clusters does not dynamically exchange molecules with the cytoplasm, while PomY does. We conclude that the PomX/PomY complex is structurally and compositionally highly flexible, neither the PomX nor the PomY part of this complex has a fixed stoichiometry, and the combined complex lacks a fixed PomX/PomY stoichiometry. This is unlike structures with a fixed stoichiometry, e.g., the ribosome, that are held together by precise, stereospecific interactions and do not exhibit size scaling. Importantly, spherical to spheroidal cellular clusters, non-stoichiometric complexes, direct size scaling with cell size in combination with a constant cytoplasmic concentration[37,38], dynamic exchange of components, as well as cluster fission and disintegration are defining features of biomolecular condensates formed by liquid-liquid phase separation (LLPS), raising the possibility that LLPS might also be involved in PomX/PomY complex formation.

## PomX forms filamentous structures in vitro
To determine whether PomX and PomY share characteristics with proteins that undergo LLPS, we analyzed the two proteins bioinformatically. PomX consists of an N-terminal, disordered region (PomX^IDR), which includes the conserved N-terminal 22 residues (PomX^NPEP) that are sufficient to stimulate ATP hydrolysis by PomZ[34], as well as a C-terminal α-helical region predicted to form a coiled-coil (PomX^CC) (Fig. 2a, Supplementary Fig. 3a). This architecture is conserved in PomX orthologs[34]. PomX^CC is required and sufficient for PomX polymerization into filaments in vitro and also interacts with PomY; both parts are required for PomX function[34]. PomY comprises an N-terminal α-helical region predicted to form a coiled-coil (PomY^CC), followed by a region with six HEAT-repeats (PomY^HEAT), and a C-terminal mostly disordered region (PomY^IDR) (Fig. 3a, Supplementary Fig. 3a). This architecture is conserved in PomY orthologs (Supplementary Fig. 3b). All five parts of the two proteins have a biased amino acid composition and also contain a large number of charged residues (Supplementary Fig. 3a) resulting in a pI of 4.9 and 9.3 for PomX and PomY, respectively. Thus, both proteins possess the

hallmarks of proteins that undergo LLPS, including multiple domains that might engage in protein-protein interactions, IDRs, and low sequence complexity.

To determine whether PomX and/or PomY undergo LLPS with condensate formation, we used an in vitro assay with purified His$_6$-tagged full-length proteins without (referred to as PomX, PomY) or with an mCh-tag (referred to as mCh-PomX, PomY-mCh), as well as a His$_6$-tagged PomX variant with a Cys-tag for labeling with the Alexa488-maleimide fluorophore (PomX-A488) (Supplementary Fig. 4). These proteins were used to collect phase diagrams based on turbidity measurements and fluorescence microscopy with increasing concentrations of purified protein close to the cellular concentrations (see above) and the crowding agent polyethylene glycol 8000 (PEG8000) to mimic the crowded intracellular environment[39] as well as for TEM (Figs. 2b and 3b).

The turbidity of PomX and mCh-PomX solutions was similar for all protein and crowding agent concentrations (Fig. 2c, Supplementary Fig. 5a). In fluorescence microscopy, mCh-PomX and PomX-A488 were inhomogeneously distributed and formed large filamentous structures with increasing PEG8000 concentrations (Fig. 2d, Supplementary Fig. 5b, c). These structures were scarce, and thus did not cause a marked increase in turbidity (Fig. 2c, Supplementary Fig. 5a). To quantify the effect of PEG8000 on the formation of these structures, we determined the coefficient of variation (CV) of the fluorescence intensity in the images. Indeed, the CV increased with the PEG8000 concentrations (Supplementary Fig. 5d, e). Importantly, spherical mCh-PomX and PomX-A488 condensates were not detected under any condition. Next, we used TEM to investigate the large filamentous PomX structures. In agreement with previous TEM analyses[31,34], in the absence of PEG8000, PomX formed regular filaments with a width of ~10 nm that are too thin to be resolved by fluorescence microscopy (Fig. 2e; cf. Fig. 2d, Supplementary Fig. 5b, c). However, in the presence of 4% PEG8000, the PomX filaments were bundled into larger filamentous structures in which individual filaments could still be distinguished (Fig. 2e). Thus, PomX in vitro forms filaments with sizes below the diffraction limit that become increasingly bundled as the PEG8000 concentrations are increased; but PomX does not form spherical condensates (Fig. 2f). Together with the observation that PomX clusters grow but do not exchange molecules with the cytoplasm in vivo, these data strongly suggest that PomX neither undergoes LLPS in vitro nor in vivo. Rather PomX forms protein filaments, which in vivo assemble to form the single PomX cluster.

## PomY forms biomolecular condensates in vitro
In turbidity measurements of PomY (Supplementary Fig. 6a) and PomY-mCh (Fig. 3c) solutions, the turbidity increased in a protein and PEG8000 concentration-dependent manner. By light microscopy, both proteins formed micrometer-sized, spherical condensates

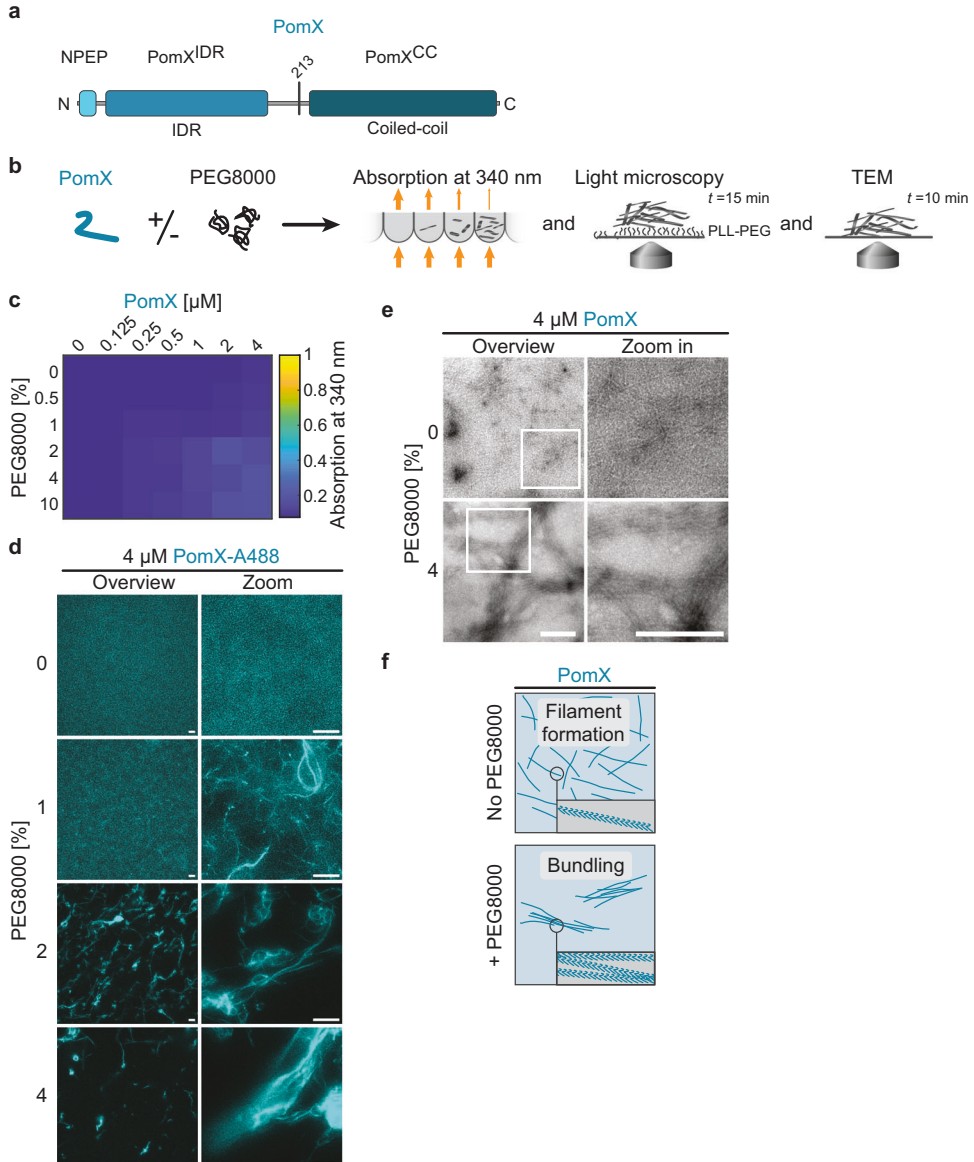

**Fig. 2 | PomX forms filaments that are bundled by PEG8000 in vitro. a** Domain architecture of PomX. PomX domain truncations as used in ref. [34] and here are indicated by vertical lines at the corresponding amino acid position. **b** Schematic of the experimental setup. **c** Turbidity measurements of PomX for increasing protein and PEG8000 concentrations. The heat map displays average absorption at 340 nm for $n = 3$ replicates. **d** PomX forms filaments that are increasingly bundled with increasing PEG8000 concentrations. Representative maximum intensity projections of confocal z-stacks (left) and high-resolution images (right) of 4 μM PomX-A488 in the presence of increasing concentrations of PEG8000. The experiment was performed three times. Scale bars, 50 μm. **e** Representative TEM images of PomX in the presence and absence of 4% PEG8000. Frames indicate regions in the zoomed images. Experiments were performed three times. Scale bars, 0.5 μm. **f** Schematic representation of the bundling effect of PEG8000 on PomX filaments.

indicative of LLPS under the conditions with increased turbidity (Fig. 3d, Supplementary Fig. 6b, c), suggesting that the dense phase has liquid-like properties with the surface tension seeking to minimize the surface area. As a measure for the phase separation behavior of PomY-mCh, we quantified the total amount of protein that segregated into the dense phase by determining the separation factor defined as the integrated fluorescence intensity of the condensates over the total integrated fluorescence intensity in the images. In the resulting phase diagram, conditions with a high separation factor coincided with those that also showed increased turbidity (Fig. 3c, e). At these PomY and PomY-mCh concentrations, which are close to the cellular concentration, the proteins only formed condensates in the presence of PEG8000. Importantly, at a concentration of 14 μM, which is far higher than the estimated cellular concentration, PomY-mCh also formed small spherical condensates in the absence of PEG8000 (Fig. 3f). In

agreement with these observations, we observed by TEM that PomY-mCh in the presence of PEG8000 formed dense spherical structures, which were absent in the absence of PEG8000 (Fig. 3g).

In addition to their spherical shape, PomY-mCh condensates also displayed other liquid-like behaviors with the fusion of condensates followed by relaxation into larger spherical condensates to minimize surface area within seconds (Fig. 3h; Supplementary Movie 1). Likewise, PomY-mCh condensates were dynamic, as shown by FRAP with recovery on a minute timescale ($t_{1/2}$: 4–5 min), similar to PomY-mCh clusters in vivo (Fig. 3i, cf. Fig. 1g). Based on (1) the protein and crowding agent concentration-dependent condensate formation, (2) their liquid-like properties as evidenced by their sphericity, fusion/relaxation events driven by surface tension, and the dynamic exchange of molecules, and (3) the bioinformatic analysis, we conclude that PomY in the absence of crowding agent undergoes LLPS at very high

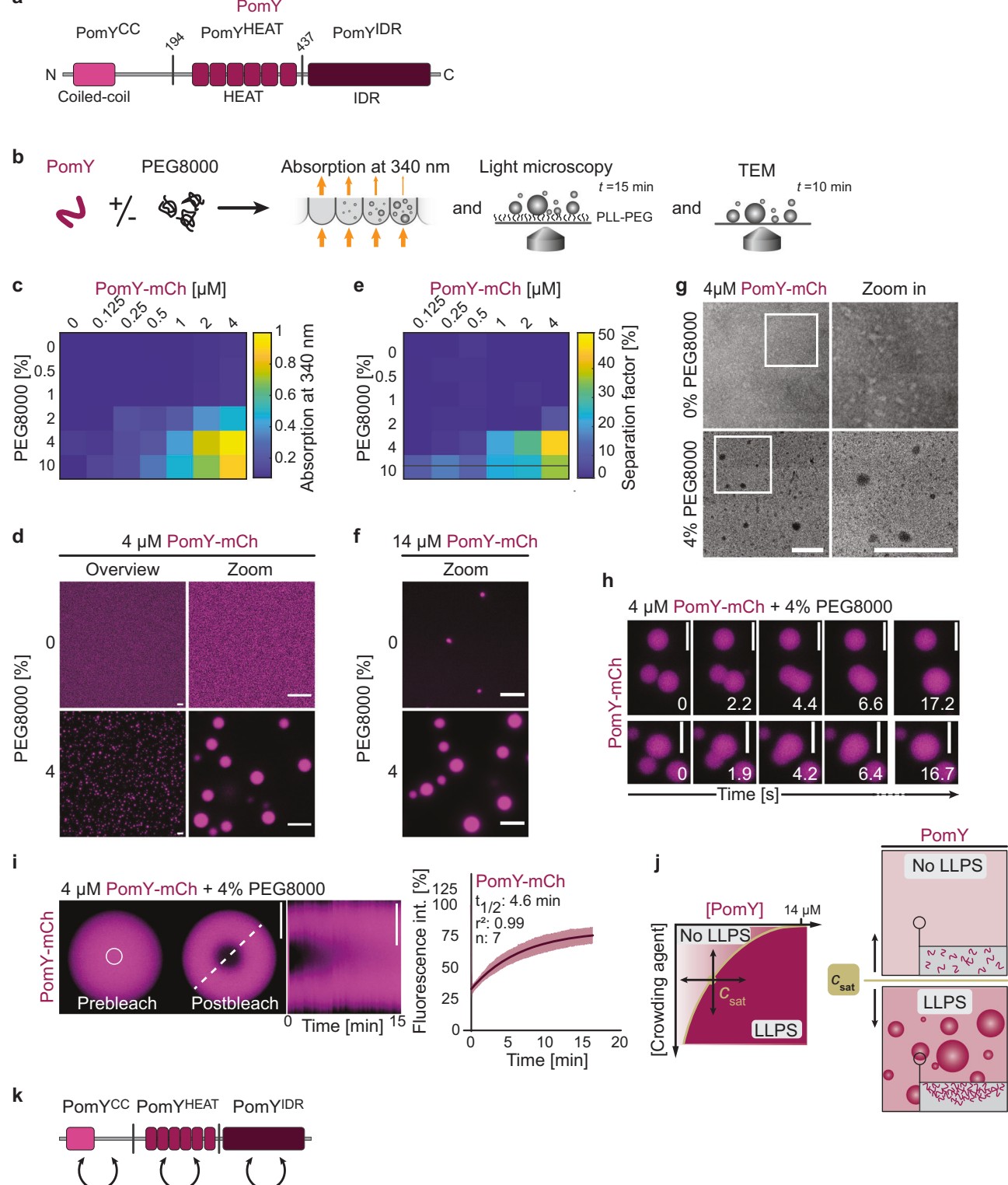

concentrations and that $C_{sat}$ for LLPS by PomY is lowered in the presence of crowding agent in vitro (Fig. 3j).

Phase separation of PomY-mCh was not specific to the crowding agent used (Supplementary Fig. 6d) and occurred with various buffers in the physiological pH range of 6.5–8.0 (Supplementary Fig. 6e). It was independent of MgCl$_2$ and decreased with increasing KCl concentration (Supplementary Fig. 6f), as well as with increasing concentrations of the aliphatic alcohol 1,6-hexanediol, which perturbs hydrophobic interactions[40] (Supplementary Fig. 6g), suggesting that PomY

condensate formation depends on electrostatic as well as hydrophobic interactions.

The phase separation behavior of PomY in vitro suggests that PomY self-interacts. Accordingly, using a bacterial adenylate cyclase two-hybrid (BACTH) assay with truncated PomY variants, we observed that full-length PomY as well as the three individual PomY domains, i.e., PomY$^{CC}$, PomY$^{Heat}$ and PomY$^{IDR}$ (Fig. 3a, Supplementary Fig. 3a), self-interact (Fig. 3k, Supplementary Fig. 7). We conclude that (1) PomY self-interacts via multivalent homotypic interactions, allowing the

**Fig. 3 | PomY forms biomolecular condensates with liquid-like properties in vitro. a** Domain architecture of PomY. PomY domain truncations are indicated by vertical lines at the corresponding amino acid position. **b** Schematic of the experimental setup. **c–f** PomY phase separates in a protein and crowding-agent-dependent manner. **c** Turbidity measurements of PomY-mCh with increasing protein and PEG8000 concentrations. The heat map displays average absorption at 340 nm for $n = 6$ replicates. **d** Representative maximum intensity projections of confocal z-stacks (left) and high-resolution images (right) of 4 μM PomY-mCh in the absence and presence of 4% PEG8000. Scale bars, 5 μm. **e** Heat map of PomY-mCh phase separation with increasing protein and PEG8000 concentrations. Values are the average separation factor in percent calculated from the maximum intensity projection of confocal z-stacks. Data from two independent experiments with $n = 8$ analyzed images per condition. **f** Representative images of 14 μM PomY-mCh in the absence and presence of 4% PEG8000. Experiments were performed three times. Scale bars, 5 μm. **g** Representative TEM images of 4 μM PomY-mCh in the presence and absence of 4% PEG8000. Frames indicate regions in the zoomed images.

Experiments were performed three times. Scale bars, 0.5 μm. **h** Time-series of PomY-mCh condensates undergoing fusion (4 μM PomY-mCh, 4% PEG8000). The experiment was performed five times. Scale bars, 5 μm. See Supplementary Movie 1. **i** Partial FRAP of PomY-mCh condensates (4 μM PomY-mCh, 4% PEG8000). Representative images of a PomY-mCh condensate before and after bleaching (white circle indicates ROI). Kymograph of the fluorescence recovery was created along the white stippled line. In the right diagram, the fluorescence intensity in the ROI was set to 100% before bleaching; the dark magenta line indicates the mean recovery fitted to a single exponential equation with the light-colored area showing the STDEV from $n = 7$ bleaching events. Scale bars, 5 μm. **j** Schematic representation of the phase separation behavior of PomY-mCh. Below $C_{sat}$, PomY is homogeneously distributed, above $C_{sat}$, PomY phase separates into a condensed phase. $C_{sat}$ can be surpassed by increasing the protein or the crowding agent concentration. **k** Schematic representation of the interactions between PomY domains as observed in BACTH analyses.

formation of a network of interacting PomY molecules without a precise stoichiometry (Supplementary Fig. 7), and (2) PomY phase separation is driven by homotypic electrostatic and hydrophobic interactions.

## PomY condensates wet PomX filaments in vitro

In vivo PomX cluster formation is independent of PomY, but PomY depends on PomX for cluster formation at native protein levels[31]. Prompted by these observations, we asked whether PomX stimulates phase separation by PomY in vitro (Fig. 4a). By fluorescence microscopy, at low PEG8000 concentrations (0–1%), where PomX-A488 alone forms (bundled) filaments (cf. Fig. 2e, f) and PomY-mCh alone is homogeneously distributed (Fig. 3d, g, j), the two proteins colocalized in filamentous structures with PomY-mCh coating and bundling the PomX-A488 filaments (Fig. 4b). At higher PEG8000 concentrations (2–10%), where PomX-A488 alone is bundled by PEG8000 into large filamentous structures and PomY-mCh alone forms spherical condensates, PomY-mCh not only coated the large PomX-A488 structures but also formed condensates that often localized on these PomX-A488 structures (Fig. 4b). Qualitatively these PomY condensates tended to be less spherical than those formed by PomY alone (Fig. 4b), suggesting that PomY-mCh condensates wet the PomX-A488 structures. As shown by analysis of the CV of PomX-A488 fluorescence intensity in the images, PEG8000 and PomY-mCh had an additive effect on the bundling of the PomX filaments especially at lower PEG8000 concentrations (Fig. 4c). Similarly by TEM, and in agreement with previous observations[31], we observed that PomY-mCh in the absence of PEG8000 bundled PomX filaments (Fig. 4d, cf. Fig. 2e). In the presence of both PomY-mCh and 4% PEG8000, PomX formed large filamentous structures in which individual filaments could no longer be distinguished (Fig. 4c) as opposed to the network formed in the presence of 4% PEG8000 only (Fig. 2e). Based on these two methods, we conclude that PEG8000 and PomY-mCh both have a bundling effect on PomX filaments and that PomY-mCh is enriched on and coats these bundled filaments independently of PEG8000.

To investigate PomY condensate formation on the PomX-A488 bundles, we performed fluorescence time-lapse microscopy in the presence of both proteins and PEG8000. We observed that PomY-mCh condensates formed in solution associated with and wetted the filamentous PomX-A488 structures (Fig. 4e). These PomY-mCh condensates also underwent fusion on the PomX-A488 structures, becoming more elongated over time (Fig. 4e, Supplementary Movie 2) suggesting that they have liquid-like properties. Indeed, PomY-mCh coating the PomX-A488 structures was dynamic, as shown by FRAP on a minute timescale ($t_{1/2}$: 2.3 min) (Fig. 4f; cf. Fig. 3i).

We conclude that in vitro at bulk PomY concentrations below $C_{sat}$ for LLPS, PomX filaments enrich PomY resulting in a high local concentration of PomY on the PomX structure, which, in turn, is bundled

by PomY. Above the bulk $C_{sat}$, PomY condensates form in solution and associate with and wet the PomX filament bundles while maintaining their liquid-like properties.

## PomY can form functional condensates in vivo in the absence of PomX

Based on the observations that PomY cluster formation depends on PomX in vivo and that phase separation might be involved in PomY cluster formation, we hypothesized that the PomY concentration in WT cells is below $C_{sat}$ for LLPS. Therefore, in the absence of PomX, PomY is homogeneously distributed. This hypothesis predicts that if the cellular PomY concentration is increased sufficiently to exceed $C_{sat}$, then PomY should be able to undergo LLPS with condensate formation independently of PomX. To test this hypothesis, we expressed *pomY-mCh* from a vanillate-inducible promoter in strains with or without PomX. At the highly elevated PomY-mCh concentration at 9 h of induction (Supplementary Fig. 8a, b) and in the presence of PomX, PomY-mCh formed a single cluster in most cells, but 12% of cells contained multiple clusters (Fig. 5a, b). Intriguingly, at this elevated concentration, PomY-mCh formed one or more clusters in the absence of PomX at random positions along the cell length in 41% of cells, while mCh alone was homogeneously distributed (Fig. 5a, b, Supplementary Fig. 8c, d). These PomY-mCh clusters were highly dynamic, grew in intensity, and were stable over long periods or rapidly disintegrated (Fig. 5c), indicating a dynamic rearrangement of PomY molecules between clusters and the cytoplasm and that these clusters are biomolecular condensates.

To investigate the functionality of the "pure" PomY-mCh condensates formed in the absence of PomX, we analyzed the localization of cell division constrictions in cells with a highly elevated PomY-mCh concentration. Stunningly, not only PomX/PomY-mCh clusters but also the "pure" PomY-mCh clusters were proficient in determining the site of cell division (Fig. 5d, e, Supplementary Fig. 8e), demonstrating that cell division site positioning solely depends on the presence of a PomY condensate, irrespective of the presence of PomX.

Altogether, we conclude that homotypic interactions among PomY molecules are sufficient to drive phase separation into biomolecular condensates in vivo if the cellular PomY concentration surpasses $C_{sat}$. These condensates are functional and support cell division. At physiological concentrations, PomY does not form condensates independently of PomX. Therefore, we infer that PomX is a scaffold that drives condensate formation by PomY in vivo when the cellular PomY concentration is below $C_{sat}$ for LLPS. In this mechanism, PomX enriches PomY locally to a concentration above $C_{sat}$ via heterotypic interactions, thereby enabling PomY condensate formation on the PomX scaffold. This suggests that PomX filaments nucleate PomY condensate formation via surface-assisted condensation. As cells

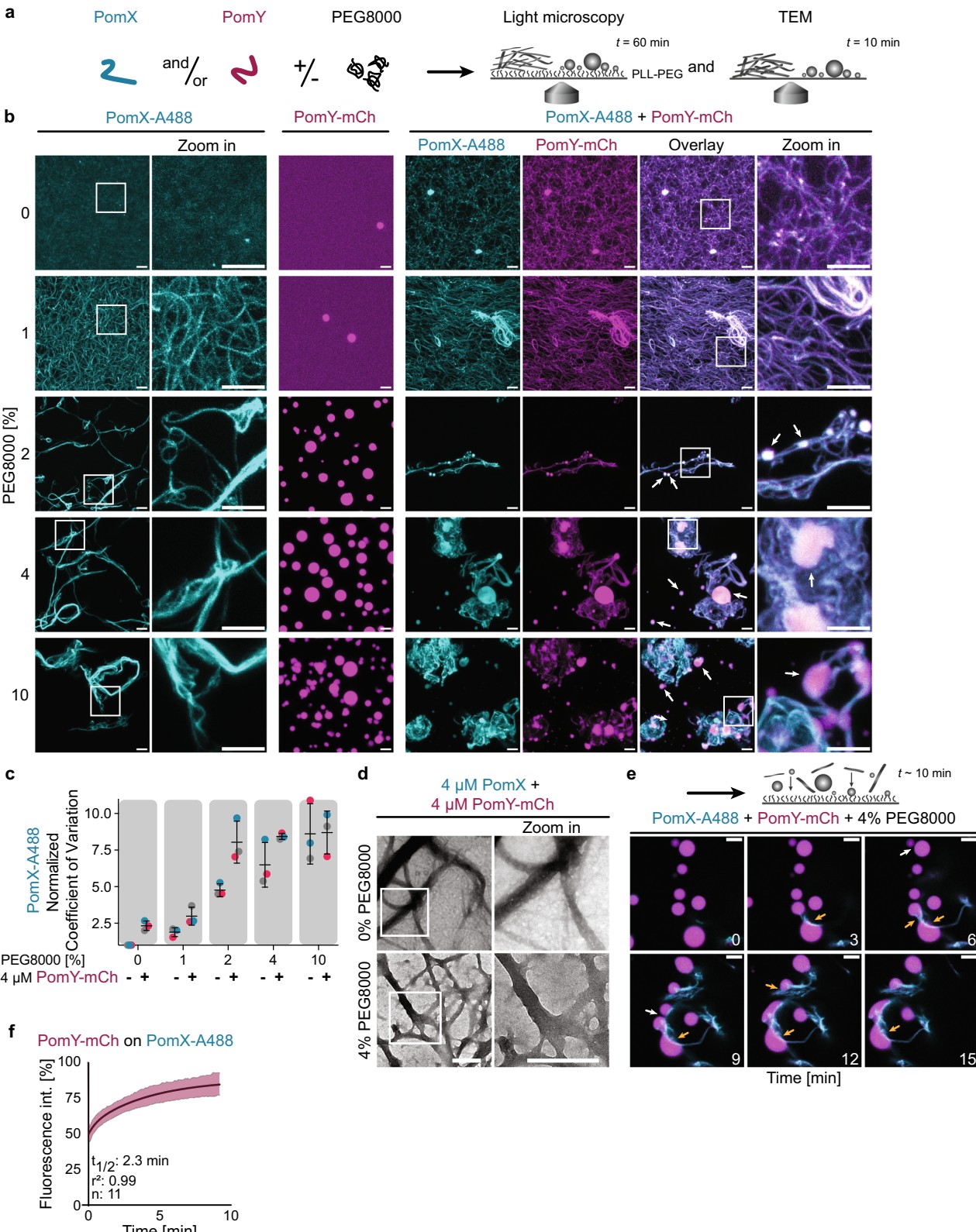

contain only a single large PomX scaffold, PomY condensate formation is restricted to the single position of the PomX scaffold. Importantly, this mechanism is also in agreement with the observations in vitro that PomY is enriched on the PomX filament bundles giving rise to a high local PomY concentration at concentrations below $C_{sat}$ for LLPS, and is forming condensates that wet PomX filaments at PomY concentrations above $C_{sat}$.

## Homotypic PomY interactions and heterotypic interactions with PomX drive PomY condensate formation in vivo

Next, we dissected PomY to understand how the PomY condensates are formed and to identify the domains that mediate or contribute to condensate formation in vitro and in vivo. Because PomY at physiological concentrations depends on PomX for condensate formation, we first mapped in more detail how PomY interacts with PomX. We

**Fig. 4 | PomY condensates wet and bundle PomX filaments in vitro. a** Schematic of the experimental setup. **b** Representative high-resolution maximum intensity projections of confocal z-stacks of 4 μM PomX-A488 and 4 μM PomY-mCh alone and in combination in the presence of increasing PEG8000 concentrations. White frames, regions shown in zoomed images. White arrows, PomY-mCh condensates. Scale bars, 5 μm. **c** Quantification of the CV of PomX-A488 fluorescence intensity in images in relation to PomY-mCh and PEG8000 based on the maximum intensity projection of confocal z-stacks. Data from three independent experiments with in total $n = 6$ analyzed images per condition. All values were normalized to the CV of the 0%PEG8000 and 0 μM PomY-mCh condition. Error bars, mean ± STDEV. **d** Representative TEM images of 4 μM PomX and 4 μM PomY-mCh in the presence and absence of PEG8000. White frames, regions in zoomed images. Scale bars, 0.5 μm. Experiments were performed three times. **e** Time-series of PomY-mCh

condensates wetting PomX-A488 filament bundles and deforming upon contact (4 μM PomY-mCh, 4 μM PomX-A488, 4% PEG8000) 10 min after mixing of proteins. White arrows, PomY-mCh condensate fusion events, orange arrows, PomY-mCh condensates deforming upon interaction with PomX-A488 filament bundle. Scale bars, 5 μm. See Supplementary Movie 2. Note that in this recording, the PomY-mCh condensates do not stimulate PomX-A488 filament bundle formation, instead the PomX-A488 filament bundle sediments from the solution into the focal plane for imaging. The experiment was performed twice. **f** Partial FRAP of PomY-mCh on PomX-A488 bundles in the presence of 2% PEG8000. In the diagram, the fluorescence intensity in an ROI was set to 100% before bleaching; the dark magenta line indicates the mean recovery, fitted to a single exponential equation with the light-colored area showing the STDEV from $n = 11$ bleaching events.

previously demonstrated that PomX interacts with full-length PomY; specifically, PomX$^{CC}$ interacts with PomY, while we have not detected interactions between PomX$^{IDR}$ and PomY[31,34]. We purified the three individual PomY domains as well as combinations of these domains with mCh- and His$_6$-tags (from hereon PomY$^{CC}$-mCh, PomY$^{Heat}$-mCh, PomY$^{IDR}$-mCh, PomY$^{CC-Heat}$-mCh and PomY$^{Heat-IDR}$-mCh) as well as a Strep-tagged full-length PomX variant (Supplementary Fig. 3a, Supplementary Fig. 4). In in vitro pull-down assays, we observed that PomX-Strep interacted not only with full-length PomY-mCh but also with all the truncated PomY-mCh variants (Fig. 6a, Supplementary Fig. 9).

To test for condensate formation in vitro, we used the in vitro LLPS assay with varying concentrations of PEG8000 together with the different mCh- and His$_6$-tagged PomY variants. All constructs containing PomY$^{CC}$ formed condensates in a PEG8000-dependent manner (Fig. 6b, c). Neither PomY$^{HEAT}$, PomY$^{IDR}$ nor both combined formed condensates (Fig. 6b, c). While PomY$^{CC}$ only formed condensates at 4% PEG8000, PomY$^{CC-HEAT}$ did so at 2% PEG8000 (Fig. 6c). Moreover, the separation factor of PomY$^{CC-HEAT}$ at 2% PEG8000 was higher than that of full-length PomY, but lower at 4% PEG8000. We conclude that in vitro homotypic interactions between PomY$^{CC}$ domains are required and sufficient for PomY condensate formation and that the two additional domains modulate phase separation by PomY.

To delineate which PomY domains are important for condensate formation in vivo, we expressed the same domains and combinations thereof fused to mCh in the presence and absence of PomX using the strong, constitutively active *pilA* promoter. The truncated proteins accumulated at levels equal to or moderately higher than that of PomY-mCh expressed from its native promoter (except possibly PomY$^{HEAT}$-mCh, which accumulated but to levels unknown) (Supplementary Fig. 10a). None of the truncations complemented the Δ*pomY* cell length and division defects, demonstrating that all three domains are required for full PomY function (Supplementary Fig. 10b, c). PomY$^{CC-HEAT}$- and PomY$^{HEAT-IDR}$-mCh, almost as efficiently as PomY-mCh, formed a single cluster per cell in a PomX-dependent manner, while cluster formation was largely abolished for the other truncated variants (Fig. 6d, e). None of the individual domains formed a cluster, demonstrating that the interactions between individual PomY domains and PomX are of low affinity and insufficient to drive cluster formation. These findings agree with cluster formation being mediated by multiple heterotypic interactions between PomX and PomY at native protein levels.

The observation that PomY$^{CC}$ did not form a cluster in vivo, even though it undergoes LLPS in vitro, albeit to a lower extent than the WT protein, suggests that the expression level in vivo is insufficient to surpass $C_{sat}$ even in the presence of PomX. Surprisingly, PomY$^{HEAT-IDR}$ formed a cluster in vivo, even though this variant lacks PomY$^{CC}$ required for phase separation in vitro. High-resolution SIM demonstrated that PomY-mCh and PomY$^{CC-HEAT}$-mCh (both phase separate in vitro) clusters were spherical to spheroidal, while PomY$^{HEAT-IDR}$-mCh (does not phase separate in vitro) clusters were strongly elongated

(Fig. 6f, g, cf. Fig. 1a, b). The spherical shape of the PomY$^{CC-HEAT}$-mCh clusters supports that these clusters, similar to those of PomY-mCh, reflect surface-assisted condensate formation by PomY on the PomX scaffold. By contrast, the highly elongated shape of the PomY$^{HEAT-IDR}$ clusters suggests that this protein also does not phase separate in vivo but instead simply interacts with the PomX scaffold without undergoing phase separation with condensate formation. Consistently, we observed in in vitro experiments in which PomX-A647 (PomX-Cys labeled with the Alexa647-maleimide fluorophore) was mixed with PomY$^{HEAT-IDR}$-mCh, PomY$^{CC-HEAT}$-mCh or PomY-mCh in the presence of 4% PEG8000 that PomY$^{CC-HEAT}$-mCh, similarly to PomY-mCh, coated the PomX-A647 filament bundles and also formed condensates that localized on the PomX-A647 bundles. By contrast, PomY$^{HEAT-IDR}$-mCh simply coated the PomX structures but did not form condensates (Fig. 6h, Supplementary Fig. 11). We conclude that not the presence but the shape of the cellular PomY clusters in vivo indicates phase separation.

In total, our data support that cluster formation by PomY in vivo is a two-pronged mechanism: (1) multivalent heterotypic interactions between PomX and at least two PomY domains locally enrich PomY on the PomX scaffold, and (2) homotypic, multivalent PomY interactions enable phase separation, and, thus, surface-assisted condensate formation on the PomX scaffold below $C_{sat}$. Importantly, for (2) to occur, the homotypic interactions between PomY$^{CC}$ domains are essential. We also speculate that the surface tension of the PomY condensates formed on the PomX scaffold is likely to drive their reorganization into the spheroidal condensates observed in vivo.

## PomY condensates enrich FtsZ

The PomX/PomY complex recruits FtsZ to form the Z-ring and initiate cell division, while FtsZ in Δ*pomY* cells localizes in a speckled pattern, and cell division occurs away from the PomX cluster[31]. Because the "pure" cellular PomY condensates in the absence of PomX are sufficient for cell division site positioning, we sought to determine how they accomplish this function. We first asked whether PomY condensates can recruit FtsZ as a client protein using a Cys-tagged FtsZ (Cys-FtsZ) labeled with Alexa488 (A488-FtsZ) in our in vitro system. In control experiments, we found that purified Cys-FtsZ behaved as untagged FtsZ and at 10 μM sedimented in a GTP-dependent manner in high-speed centrifugation experiments and exhibited concentration-dependent GTPase activity (Supplementary Fig. 12a–c), supporting that it polymerizes in a GTP-dependent manner to form filaments.

Next, we formed PomY-mCh condensates in the presence of PEG8000 and A488-FtsZ or the control proteins EGFP or a commercial Streptavidin Alexa488 conjugate (Strep-A488) (Fig. 7a). None of the three tested proteins formed condensates in the absence of PomY (Supplementary Fig. 13). A488-FtsZ partitioned into the PomY-mCh condensates and was significantly more

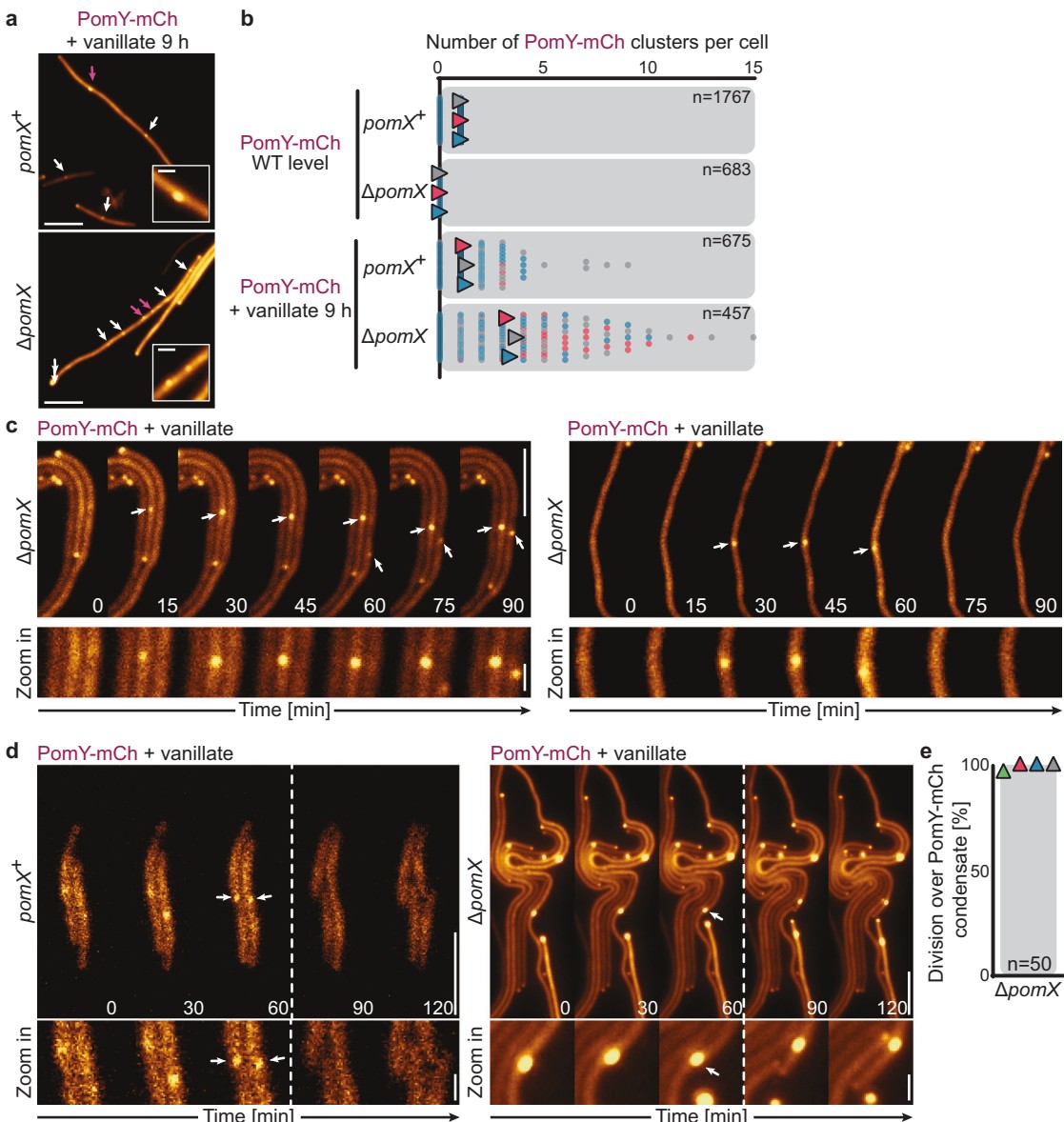

**Fig. 5 | PomY forms functional condensates in a concentration-dependent manner in vivo independently of PomX. a** Fluorescence microscopy of cells with a highly elevated PomY-mCh concentration after 9 h of induction with 1 mM vanillate. White arrows, PomY-mCh clusters; magenta arrows, clusters shown in insets. Scale bars, 5 μm and 0.5 μm in insets. **b** Quantification of the number of PomY-mCh clusters in cells in a and in cells accumulating PomY-mCh at native levels. Data from three replicates are shown in different colors and with triangles indicating the mean. For calculating the mean, only cells with clusters were included. *n*, number of cells analyzed. **c** Fluorescence time-lapse microscopy of cells with a highly elevated PomY-mCh concentration. Cells grown in suspension without vanillate were transferred to agarose pads containing 1 mM vanillate at 32 °C. Images show cells after ~10 h of induction. White arrows, cell regions with clusters

shown in zoomed images. Scale bars, 5 μm, 0.5 μm in the zoomed images. The experiment was performed three times, and representative cells are shown. **d** Fluorescence time-lapse microscopy of cells with a highly elevated PomY-mCh concentration during division. Cells were treated as in (**c**). White arrows indicate regions with clusters immediately before cell division. Zoomed images show cluster regions during the cell division event. Stippled lines indicate cell division events. Scale bars, 5 μm and 0.5 μm in zoomed images. The experiments were performed three times (left panel) and four times (right panel) and representative cells are shown. **e** Quantification of cell divisions over PomY-mCh condensates. Experiments were performed as in (**c**). Data from four biological replicates are shown in different colors with triangles indicating the mean. Only cells with PomY-mCh condensates were used for the analysis. *n*, number of cell divisions analyzed.

enriched than the EGFP and Strep-A488 controls, as the fluorescence of A488-FtsZ inside condensates was much higher than the background (Fig. 7b). To precisely quantify the partitioning of proteins into the PomY condensates, we calculated the enrichment of the proteins defined as the average intensity in the condensates over the average intensity of the background. Indeed, A488-FtsZ was significantly more enriched than the two control proteins (Fig. 7c). Of note, if the A488-FtsZ enrichment had been due to the A488 labeling, the enrichment of Step-A488

should have been ~38-fold higher than for A488-FtsZ (Fig. 7b). PomY-mCh phase separation was not influenced by the presence any of the proteins tested (Fig. 7d). Finally, A488-FtsZ fluorescence in the condensates was more variable than for the control proteins, as indicated by the CV of the client proteins' fluorescence in the dense phase (Fig. 7e). We speculate that strong interactions between FtsZ and PomY cause this heterogeneity. We conclude that PomY condensates concentrate FtsZ in vitro as a client protein.

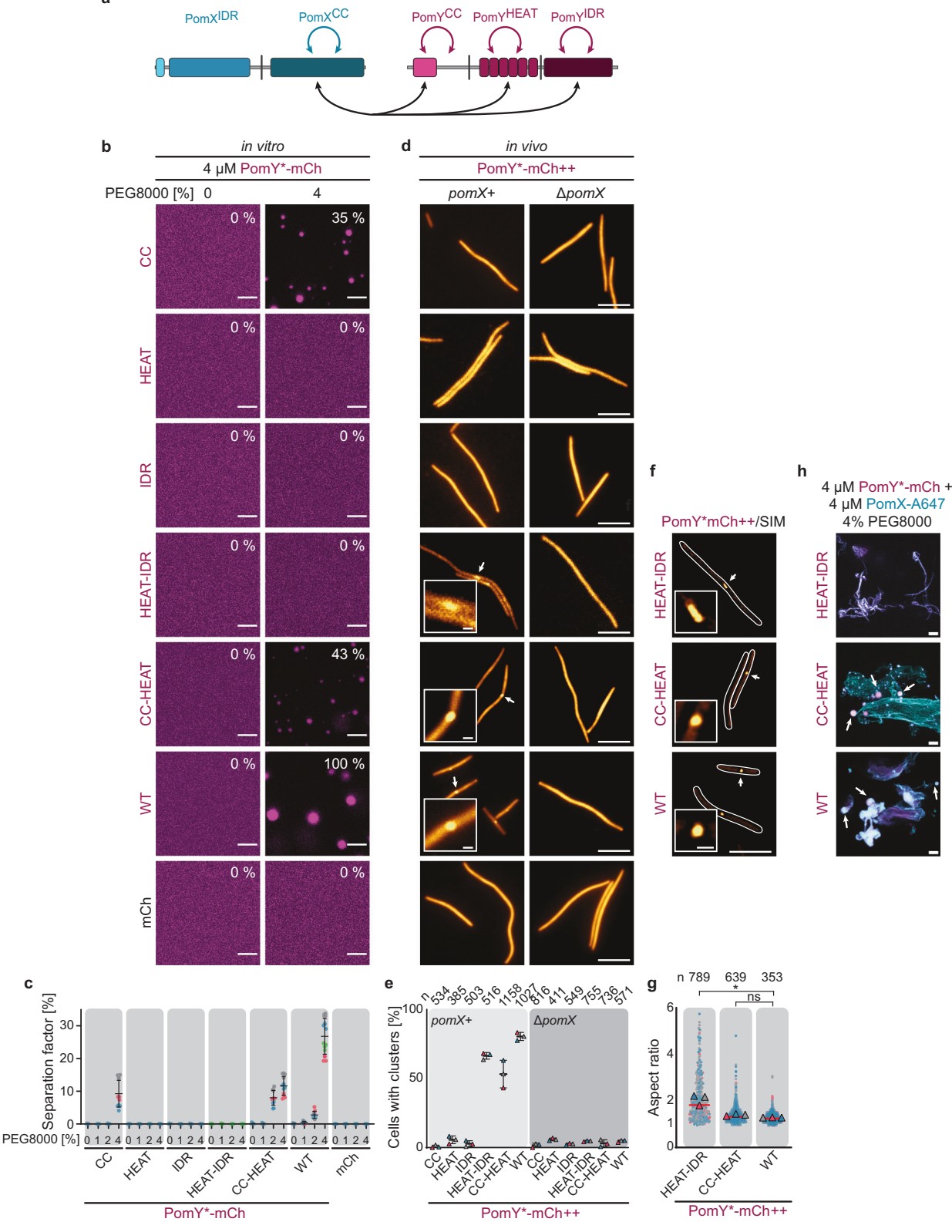

## PomY condensates nucleate GTP-dependent FtsZ polymerization and bundle FtsZ filaments

To assess whether PomY condensates act as local nucleation centers for FtsZ polymerization, we investigated their effect on FtsZ filament formation in the presence of GTP in vitro (Fig. 8a). Since individual FtsZ filaments are only one monomer wide, i.e., 5–7 nm, and 50–100 nm long, and thus cannot be resolved by fluorescence microscopy[41,42], we

first tested the effect of PomY on FtsZ polymerization using TEM. A488-FtsZ incubated alone without or with PEG8000 occasionally formed a very small number of filamentous structures (Fig. 8b); thus, under these conditions, 1 μM A488-FtsZ and 0.5 mM GTP, FtsZ does not efficiently polymerize by itself. Similarly, in the presence of PomY-mCh, but without PEG8000, 1 μM A488-FtsZ also only formed a very small number of filamentous structures (Fig. 8b). However, in the

**Fig. 6 | Multivalent homotypic PomY interactions and multivalent heterotypic interactions with PomX are required for PomY condensate formation in vivo.** **a** Schematic of the interactions between PomX and PomY. Cyan, magenta and black arrows indicate interactions mapped in ref. [34], in the BACTH analysis and in the pull-down experiments, respectively. **b, c** PomY$^{CC}$ is required and sufficient for phase separation in vitro, while PomY$^{HEAT}$ and PomY$^{IDR}$ modulate phase separation. **b** Representative high-resolution images of PomY-mCh variants with or without 4% PEG8000 in vitro. The average separation factor in % normalized to that of PomY-mCh is indicated in the images and calculated from the maximum intensity projection of confocal z-stacks of the individual conditions. Scale bars, 5 μm. **c** Analysis of the separation factor of PomY-mCh variants in vitro in the presence of different PEG8000 concentrations. The average separation factor is calculated as in (**b**). Data from at least three independent experiments each shown as colored dots with $n = 12$ images per condition except $n_{WT(4\%PEG8000)} = 16$, $n_{HEAT-IDR(2\%PEG8000)} = 10$ and $n_{HEAT-IDR(4\%PEG8000)} = 10$. Error bars, mean ± STDEV. **d** Fluorescence microscopy of

cells expressing the PomY-mCh variants in (**b**). Arrows, clusters shown in the insets. ++, PomY-mCh variants were expressed from *pilA* promoter. Scale bars, 5 μm, 0.5 μm in insets. **e** Percentage of cells in d with a cluster. Cells from three independent experiments with the mean shown as colored triangles, error bars, mean ± STDEV. **f** SIM images of cells expressing PomY-mCh variants as in (**d**). White arrows, clusters shown in the insets. Scale bars, 5 μm, 0.5 μm in inset. **g** Aspect ratio of clusters of the indicated PomY-mCh variants based on SIM images from (**f**). Measurements from three independent experiments are shown as colored dots together with the mean. Red line, the median. $n$, number of analyzed clusters. *, significant difference ($P = 0.00005$), ns, not significant ($P = 0.2864$) in two-way ANOVA test. **h** Representative maximum intensity projections of confocal z-stacks of indicated PomY-mCh variants mixed with PomX-A467 and 4% PEG8000 in vitro. Images show overlays of PomY-mCh (magenta) and PomX-A647 (cyan) images shown in Supplementary Fig. 11. White arrows, PomY*-mCh condensates. The experiment was performed three times. Scale bars, 5 μm.

presence of PomY-mCh and PEG8000, i.e., under conditions where PomY-mCh forms condensates, A488-FtsZ filaments and filament bundles were highly abundant forming an extended network (Fig. 8b). These filaments and filament bundles were associated with protein-dense spherical and lenticular structures that tended to elongate along the A488-FtsZ structures.

Similar to the TEM analysis, fluorescence microscopy at ~60 min of incubation revealed that A488-FtsZ formed an extensive network of filamentous structures only in the presence of both PEG8000 and PomY-mCh, but not in the absence of either PomY-mCh or PEG8000 (Fig. 8c). Consistently, high average fluorescence intensity was only observed when both PEG8000 and PomY-mCh were present (Fig. 8d). A488-FtsZ was enriched in the PomY-mCh condensates (Fig. 8c) and the entire network of FtsZ-A488 structures was covered by PomY-mCh (Fig. 8c). This extensive filament bundle/condensate network even extended away from the coverslip surface in 3D (Fig. 8e, Supplementary Fig. 14a), and, importantly, no "free" FtsZ structures that were not connected to this network could be detected. Moreover, the PomY-mCh condensates were often deformed into lenticular shapes (Fig. 8c), suggesting that the spherical to lenticular-shaped protein-dense structures observed by TEM (Fig. 8b) are PomY-mCh condensates. Together, the two methods support that PomY-mCh condensates stimulate FtsZ polymerization into filaments and further bundle these filaments.

To investigate how these A488-FtsZ networks are formed, we performed time-lapse fluorescence microscopy. A488-FtsZ was enriched in PomY-mCh condensates that sedimented onto the coated surface close to the coverslip (Fig. 8f). Filamentous A488-FtsZ structures formed in solution sedimented into the focal plane and associated with PomY condensates (Fig. 8f, Supplementary Fig. 14b). Importantly, these A488-FtsZ filament bundles were coated with PomY-mCh and/or connected to PomY-mCh condensates. Similarly, within minutes A488-FtsZ filament bundles emerged from PomY-mCh condensates, which were deformed into lenticular shapes in this process (Fig. 8g, Supplementary Fig. 14c, Supplementary Movie 3). These A488-FtsZ filament bundles were also coated with PomY-mCh (Fig. 8f). A488-FtsZ enriched PomY-mCh condensates also underwent fusion/relaxation events into a spherical shape indicating their liquid-like nature (Fig. 8f, Supplementary Fig. 14b, Supplementary Movie 3).

In time-lapse recordings of the larger field of views, it was apparent that the extensive filamentous A488-FtsZ network was generated over time from three types of interactions: (1) A488-FtsZ filament bundles emerging from condensates, (2) fusion and alignment of A488-FtsZ filament bundles, and (3) A488-FtsZ filament bundles associating with a condensate (Fig. 8h, Supplementary Fig. 14d, Supplementary Movies 3 and 4).

We conclude that PomY condensates are enrichment centers for FtsZ and drive local FtsZ filament nucleation in the presence of GTP. In this process, PomY condensates are deformed, and PomY associates

with the FtsZ filaments, thereby bundling these filaments. In this way, PomY condensates stimulate GTP-dependent FtsZ filament formation and bundling at concentrations that are too low for the unaided formation of FtsZ filaments.

To investigate the link between PomY phase separation, FtsZ recruitment, and cell division in vivo, we took advantage of PomY variants that form a cluster in the presence of PomX and can (PomY-mCh, PomY$^{CC-HEAT}$-mCh) or cannot undergo phase separation (PomY$^{HEAT-IDR}$-mCh) (Fig. 6d–g). We analyzed these variants for their ability to recruit FtsZ and stimulate Z-ring formation by scoring the colocalization of PomY*-mCh clusters and cell division constrictions (Fig. 8i, j). Like PomY-mCh, PomY$^{CC-HEAT}$-mCh clusters mostly colocalized with division constrictions, suggesting that this variant can recruit FtsZ and stimulate Z-ring formation. By contrast, PomY$^{HEAT-IDR}$-mCh clusters did not colocalize with constriction sites. Although we cannot rule out that PomY$^{CC}$, in addition to being central for condensate formation in vitro and in vivo, is also central for the interaction to FtsZ, these observations suggest that PomY phase separation is required to recruit FtsZ to the PomY/PomX/PomZ complex to stimulate Z-ring formation. We also note that phase separation by PomY is not sufficient to stimulate cell division to WT levels as PomY$^{CC-HEAT}$ does not complement the ΔpomY mutant (Supplementary Fig. 10b, c).

In total, our data show that PomY condensates act as concentration centers for FtsZ, resulting in the localized nucleation of FtsZ filament formation, and subsequent bundling of FtsZ filaments, and they strongly support that this is the underlying mechanism by which Z-ring formation is stimulated precisely over the PomX/PomY/PomZ cluster in vivo.

## Discussion

Here, we identified a mode of cell division regulation in bacteria that depends on a selective biomolecular condensate. This condensate emerges from the PomY protein undergoing LLPS based on multivalent interactions with itself and the molecular PomX scaffold. The single PomX scaffold locally enriches PomY and thus allows the formation of the single PomY condensate below $C_{sat}$ for LLPS. Thereby this PomX-mediated surface-assisted condensation restricts condensate formation to a single position in the cell. This localized PomY condensate, in turn, selectively enriches the tubulin homolog FtsZ, the key protein in bacterial cell division, to nucleate and bundle GTP-dependent FtsZ filaments, thereby guiding Z-ring formation and subsequent cell division precisely over the PomX/PomY cluster.

That PomY undergoes LLPS to form a biomolecular condensate is supported by several observations, all hallmarks of LLPS[3–6]. In vivo and at physiological concentrations, the PomX-dependent PomY cluster is spherical to spheroidal, non-stoichiometric, scales in size with cell size, exchanges molecules with the cytoplasm, disintegrates during division and is subsequently formed de novo. Finally, at very high cellular concentrations PomY by itself forms spherical clusters. In vitro PomY

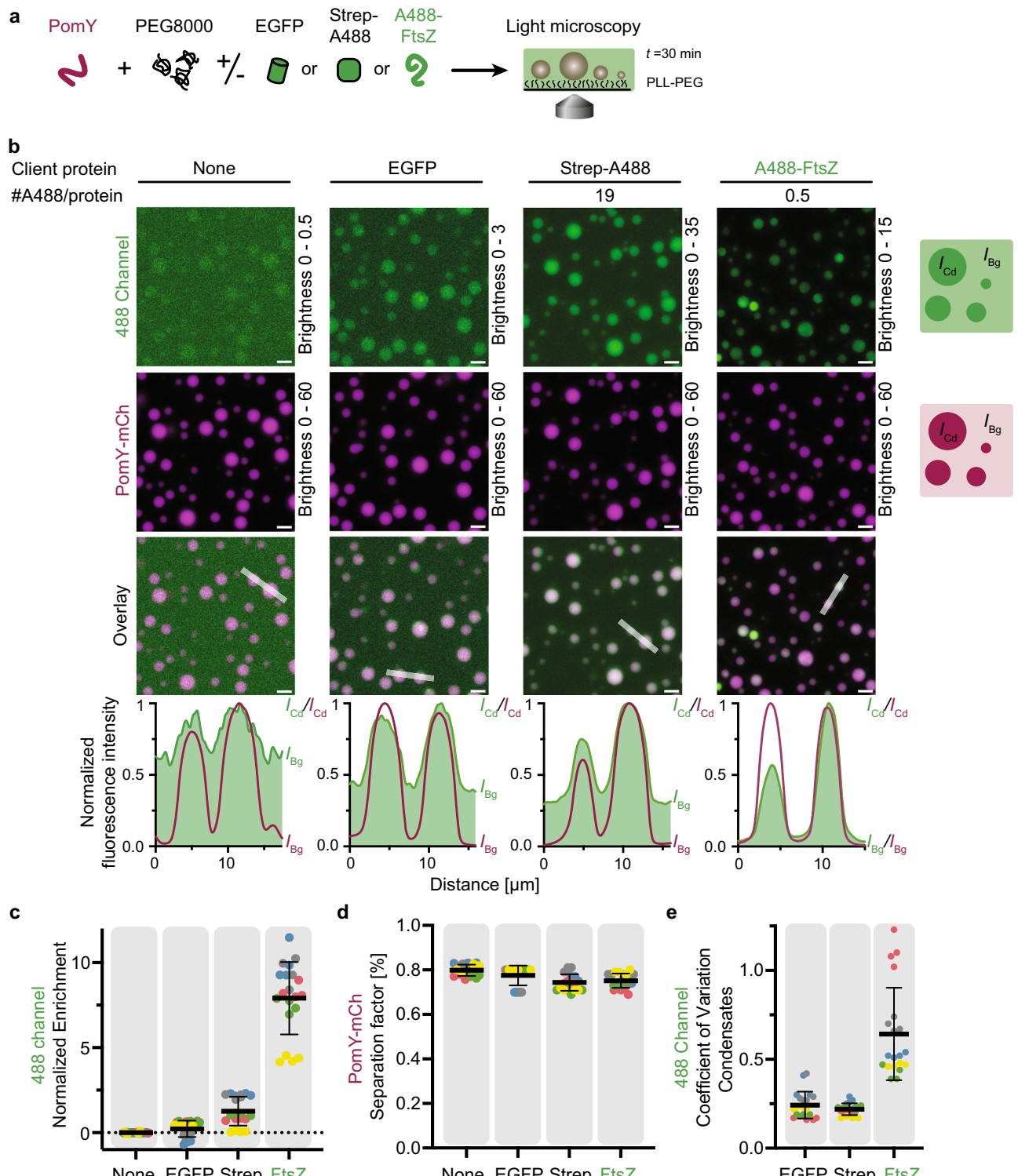

**Fig. 7 | PomY condensates enrich FtsZ. a** Schematic of the experimental setup. **b** Representative high-resolution images and fluorescence intensity line plots (lower panels, smoothed) of 4 μM PomY-mCh in the presence of 4% PEG8000 and either no client protein (bleedthrough control), EGFP, Strep-A488 or A488-FtsZ. Note that Strep-A488 and A488-FtsZ are labeled with on average 19 and 0.5 molecules per protein, respectively. Therefore, brightness and contrast settings for the 488 channel were individually adjusted to enable the display of all images. Fluorescence intensity inside the condensates ($I_{Cd}$) and of the background ($I_{Bg}$) is indicated in the schematics on the right and in the line plots. The line plots are from scans across two condensates as shown in the overlay images and are normalized to the highest intensity value. Scale bars, 5 μm. **c** Quantification of the enrichment of EGFP, Strep-A488 and A488-FtsZ in PomY-mCh condensates from confocal z-stacks. For normalization, the respective enrichment of the bleedthrough control was subtracted from each value. The average enrichment of the bleedthrough control was close to an equal partitioning of 1 (1.7 ± 0.5). **d** The presence of A488-FtsZ, EGFP or Strep-A488 does not alter the extent of PomY phase separation. Quantification of the separation factor of PomY-mCh in the presence of 4% PEG8000 alone and in the presence of A488-FtsZ, EGFP or Strep-A488. **e** FtsZ-A488 fluorescence in the condensates is more variable than that of EGFP or Strep-A488. Quantification of the CV of the client protein fluorescence in PomY-mCh condensates. **c**–**e** Data from five independent experiments as indicated with colors with in total $n = 20$ analyzed images per condition in which on average $n = 553 ± 139$ condensates were identified per analyzed image. Error bars, mean ± STDEV.

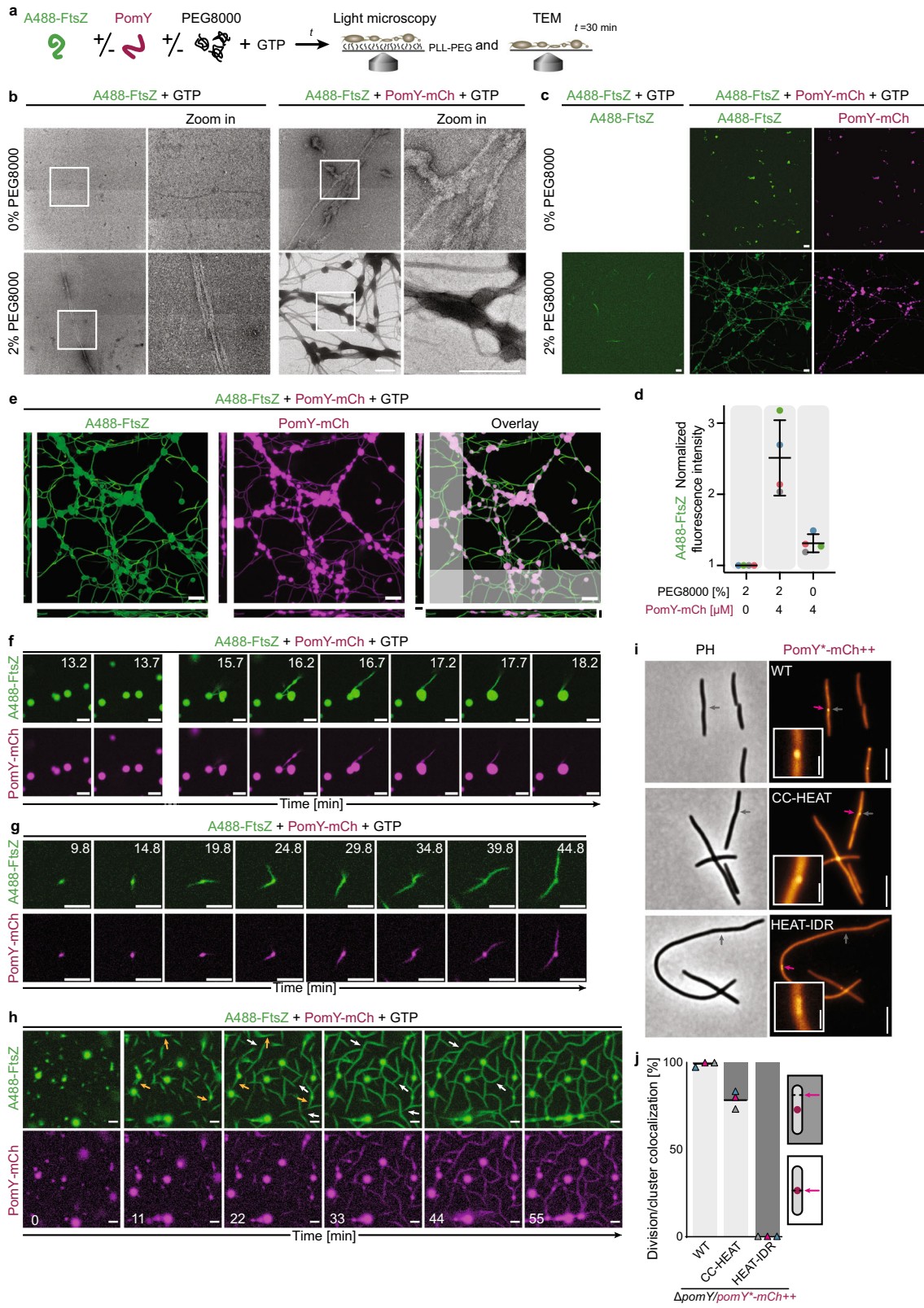

forms spherical, liquid-like condensates in a protein and crowding agent concentration-dependent manner. Moreover, phase separation both in vitro and in vivo is driven by multivalent homotypic interactions. Specifically, PomY$^{CC}$ is both necessary and sufficient for LLPS in vitro, while PomY$^{HEAT}$ and PomY$^{IDR}$ modulate LLPS. Similar domains have previously been implicated in phase separation behavior[20,43,44]. The many charged residues and the unusually high isoelectric point of PomY together with the strong salt dependence of PomY LLPS and its sensitivity to 1,6-hexanediol in vitro all suggest that electrostatic interactions between charged sequence motifs, as well as hydrophobic interactions contribute to its phase separation.

In contrast to PomY, and despite the overall similar structural features of PomX and PomY, PomX does not phase separate. Rather we confirmed that PomX self-assembles into filaments that are bundled by

**Fig. 8 | PomY condensates nucleate GTP-dependent FtsZ polymerization and bundle FtsZ filaments. a** Experimental setup. **b** TEM images of 1 μM A488-FtsZ with 0.5 mM GTP with or without 4 μM PomY-mCh and/or 2% PEG8000. Frames, regions in zoomed images. Structures shown are rare except those formed in the presence of PomY-mCh and PEG8000. Experiments were performed three times. Scale bars, 0.5 μm. **c** Maximum intensity projections of confocal z-stacks of 1 μM A488-FtsZ with 0.5 mM GTP with or without 4 μM PomY-mCh and/or 2% PEG8000 at ~60 min incubation. The experiment was performed four times. Scale bars, 10 μm. **d** Average fluorescence intensity of A488-FtsZ from experiments in (**c**) normalized to that of A488-FtsZ in the absence of PomY-mCh and PEG8000. Data from four maximum intensity projections per condition shown as colored dots. Error bars, mean ± STDEV. **e** Maximum intensity and z-projection (of shaded areas) of a confocal z-stack (conditions and replicates as in (**c**). Scale bars, 10 μm. **f, g** Representative time-series acquired close to coverslip surface showing fusion of A488-FtsZ-enriched PomY-mCh condensates and sedimenting A488-FtsZ filament bundles (**f**)

and the emergence of filamentous A488-FtsZ structures from a PomY-mCh condensate (**g**). Conditions as in (**c**). The experiment was performed four times. Scale bars, 5 μm. See Supplementary Movie 3. **h** Lower resolution time-series of a larger field of view showing the emergence of extended filament networks. A488-FtsZ filament bundles emerge from condensates (orange arrows) and fuse and align with each other (white arrows). Conditions as in (**c**). Scale bars, 5 μm. See Supplementary Movie 4. **e–h** Images gamma-corrected for visualization, unaltered images in Supplementary Fig. 14a–d. **i** Representative images of cells with cell division constrictions (gray arrows) and PomY*-mCh clusters (magenta arrow). ++ indicates expression from *pilA* promoter. Scale bars, 5 μm, 1 μm in insets. **j** Quantification of the colocalization of clusters of PomY*-mCh variants with division constrictions. Histogram, black lines indicate the mean of three independent experiments as in (**i**) shown as triangles in different colors. Light gray and dark gray, indicate constrictions colocalizing or not colocalizing with a PomY*-mCh cluster, respectively. *n*, 141, 102 and 18 (left to right).

PomY in vitro[31,34] and associate into a highly stable, filamentous scaffold in vivo. This structure undergoes fission during cell division and then doubles in size over the cell cycle by incorporating PomX molecules.

Why then do two distinct mechanisms of macromolecular complex formation exist within the PomX/PomY cluster, i.e., filament formation and phase separation, and how do their interplay lead to the formation of a single complex in vivo? In the absence of PomX, PomY forms condensates only when the bulk concentration is above $C_{sat}$. In vitro $C_{sat}$ is surpassed either by elevation of protein concentration or addition of crowding agents that are believed to increase the effective protein concentration by an excluded volume effect[39] (Fig. 9a). Analogously, in vivo, PomY only forms condensates independent of PomX if it is highly overexpressed, suggesting that native PomY levels, where PomY by itself is homogeneously distributed, are well below $C_{sat}$. Importantly, in the presence of PomX, PomY also forms structures below $C_{sat}$. In vitro, PomY was enriched on the PomX filaments, coated and bundled these structures below $C_{sat}$; above $C_{sat}$, PomY condensates also formed in the bulk and wetted the PomX filament bundles. Similarly, in vivo PomY only forms a single cluster per cell which colocalizes with the PomX scaffold[31] at native concentrations, i.e., below $C_{sat}$; when PomY is overexpressed it can also form multiple ones, suggesting that some of them form independently of PomX (Fig. 9b).

Altogether, this implies that the PomX/PomY cluster at physiological concentrations is formed by surface-assisted condensation of PomY on the PomX scaffold (Fig. 9b). The dissection of PomY suggests that this process proceeds in two steps. First, PomX interacts with all three PomY domains, and the opposing isoelectric points of PomX (pI 4.9) and PomY (pI 9.3) suggest electrostatic interactions between the proteins. These multivalent heterotypic interactions between PomX and PomY locally enrich PomY on the PomX scaffold resulting in a high local PomY concentration. Second, because the PomX-dependent PomY clusters display all the hallmarks of biomolecular condensates, we infer that PomY undergoes phase separation locally on the PomX scaffold, and this phase separation specifically requires PomY[CC]. Importantly, because the bulk concentration is well below $C_{sat}$ for LLPS, PomY forms a single condensate precisely on the PomX scaffold. Similar surface-assisted condensate formation has been observed in the spatiotemporal regulation of eukaryotic protein condensates on microtubules, membranes and DNA[10–12,16,18,45–48]. Specifically, it has been suggested that surface-assisted condensation can occur via prewetting[10,45,48]. Our data indicate that prewetting also underlies the formation of the PomX/PomY complex observed here. Of note, we observe that PomY is more spheroidal than PomX in the PomX/PomY cluster, supporting that PomY indeed forms a single condensate on the PomX scaffold below $C_{sat}$ that reorganizes into spherical shapes due to its surface tension. However, in vitro we observed PomY coating the PomX filaments in a seemingly uniform layer below $C_{sat}$. We also did not observe this PomY layer breaking up into more round condensed

phases akin to the (regular) droplet formation of condensates coating fibers or filaments based on hydrodynamic instability[10,18,46,49]. We speculate that these discrepancies can be explained by the different geometries of these systems. Compared to individual DNA strands or microtubules on which round condensates have been observed[10,18,46,49], PomX structures in vitro and likely also in vivo consist of many individual filaments. The capillary phenomena occurring during (pre) wetting of fibrous networks are more complicated ranging from stable droplets to liquid columns depending on the geometry[50,51]. This may also explain the more round PomY-mCh condensates we observed on the short PomX clusters in vivo, compared to the extended PomX filamentous structures in vitro that are coated and bundled by PomY-mCh.

PomY condensates in vitro enrich the tubulin homolog FtsZ thereby locally stimulating GTP-dependent FtsZ polymerization. PomY also coats and wets the FtsZ filaments thereby bundling them. The local nucleation of FtsZ filament bundles by PomY condensates is phenomenologically similar to synthetic coacervates that artificially enrich FtsZ[52]. More importantly, it is reminiscent of eukaryotic condensates that stimulate the polymerization of cytoskeletal proteins like tubulin[9,18,20] or actin[8,17,19,21,53], suggesting a broadly conserved role of condensates in the spatiotemporal regulation of protein filament formation and that spatiotemporally regulated tubulin polymerization by condensates is an ancient mechanism. In vitro PomY condensates deformed during FtsZ filament formation and covered the FtsZ filament bundles, likely leading to the alignment, fusion and bundling of growing filaments into larger networks. However, whether bundling of FtsZ filaments by PomY has a functional role in vivo remains to be uncovered. Nevertheless, we speculate that PomY could act as a crosslinker that bundles FtsZ similar to *E. coli* ZapA[54].

Taken together, we propose a model for the regulation of bacterial cell division by a biomolecular condensate (Fig. 9c). High-affinity interactions among PomX molecules drive the assembly of a single filamentous PomX scaffold per cell (Fig. 9c, I). During the cell cycle, additional PomX molecules are incorporated into this scaffold allowing it to double in size. PomY associates with the PomX scaffold via multivalent heterotypic interactions, thereby locally reaching a high concentration resulting in surface-assisted condensate formation on the PomX scaffold based on homotypic PomY[CC] interactions (Fig. 9c, II). The PomY condensate exchanges molecules with the cytoplasm, but overall there is a net gain of PomY molecules in the condensate allowing it to grow over time. The PomX/PomY complex translocates on the nucleoid toward midcell via PomX/PomY-stimulated ATP hydrolysis by the DNA-bound PomZ ATPase[31,34] (Fig. 9c, III). At midcell, the PomY condensate recruits FtsZ by direct interaction and enriches FtsZ locally (Fig. 9c, IV). Upon an unknown signal, FtsZ filaments emerge from the PomY condensates to form the Z-ring consisting of treadmilling FtsZ filaments on the membrane (Fig. 9c, V). Subsequently, FtsZ recruits other proteins required to execute cytokinesis.

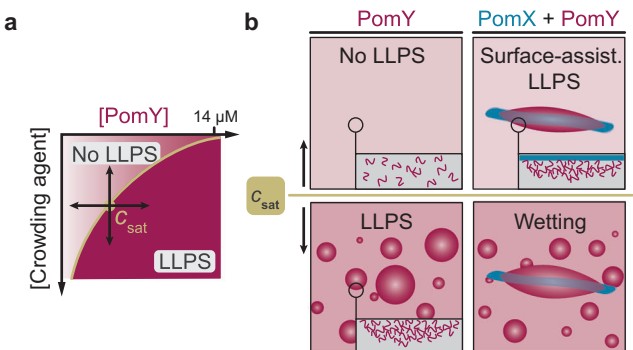

During cytokinesis, the PomX scaffold undergoes fission with both daughters "receiving" a smaller scaffold, while the PomY condensate disintegrates (Fig. 9c, VI). We speculate that the advantage of the PomX scaffold fission is that the daughters are each born with a single PomX scaffold, thereby ensuring that only a single PomY condensate forms. Similarly, we suggest that PomY condensate disintegration represents an elegant mechanism to prevent the immediate stimulation of

additional divisions after a completed division event. Altogether, this mechanism represents a bona fide positive mechanism for spatio-temporal regulation of Z-ring formation and cell division in bacteria. We speculate that a similar mechanism might be employed by analogous positive regulatory systems, especially in unusually shaped bacteria since it does not require cellular symmetry as in negative regulatory systems[26,27].

**Fig. 9 | A phase-separated biomolecular PomY condensate nucleates polymerization of the tubulin homolog FtsZ to spatiotemporally regulate bacterial cell division. a** Schematic phase diagram for PomY phase separation dependent on crowding agent and protein concentration. **b** Schematic representation of PomY phase separation behavior. PomY by itself is homogeneously distributed below $C_{sat}$, and phase separates above $C_{sat}$ (left panels). In the presence of PomX, PomY is enriched on the PomX scaffold below $C_{sat}$ and undergoes surface-assisted condensation on the PomX scaffold; above $C_{sat}$, PomY condensates also form in the bulk and wet the PomX scaffold (right panels). **c** Model for the regulation of cell division by a biomolecular condensate in *M. xanthus*. I, a single PomX scaffold per cell is formed via high-affinity interactions among PomX molecules. II, PomY associates with the PomX scaffold via multivalent heterotypic interactions, thereby locally reaching a high concentration resulting in surface-assisted condensate formation on the PomX scaffold via homotypic PomY$^{CC}$ interactions. III, the PomX/PomY complex translocates on the nucleoid toward midcell via PomX/PomY-stimulated ATP hydrolysis by the DNA-bound PomZ ATPase. IV, at midcell, the PomY condensate enriches FtsZ locally via direct interaction. V, upon an unknown signal, FtsZ filaments emerge from the PomY condensates to form the Z-ring that recruits other proteins to execute cytokinesis. VI, during cytokinesis, the PomX scaffold undergoes fission and the PomY condensate disintegrates.

Our study raises several intriguing questions for future research avenues. First, we do not know how the remarkable fission of the PomX scaffold occurs during cell division. Our data support that this event does not involve a duplication event, unlike centrioles that duplicate before segregation[55]. PomX scaffold splitting follows a cell division-specific signal[31,34], which we suspect is of mechanical nature, e.g., septum ingrowth. Second, the mechanism underlying PomY condensate disintegration also remains unknown. Even though disintegration occurs less often at increased PomY concentration[31], it is unlikely to be caused by dilution of PomY because total cell volumes do not change during division, as opposed to the sudden volume increase during nuclear envelope breakdown in eukaryotes that results in disintegration of the nucleolus condensates[56]. We speculate that PomY condensate disintegration could be triggered by other cell division proteins disrupting PomX-PomY or PomY-PomY interactions. A sudden increase in the surface-to-volume ratio of the PomY condensate as a consequence of the PomX scaffold fission could also lead to PomY condensate disintegration. Alternatively, the association of PomY with FtsZ filaments as we observed in vitro could influence PomY condensate stability. Third, we cannot explain why the PomY condensate only becomes competent for stimulating Z-ring formation at midcell. Maturation of PomY condensates, or the PomZ ATPase might allow Z-ring stimulation only at midcell. Alternatively, the nucleoid, similar to nucleoid occlusion effects in other systems, could influence Z-ring formation. Finally, it remains to be uncovered how the PomZ ATPase associates with the PomX/PomY complex to stimulate its translocation on the nucleoid to midcell.

Interestingly, AAPs of other ParA/MinD-type ATPases, i.e., ParB and McdB, have recently been shown to undergo phase separation[57–59]. Although PomY, ParB and McdB share little sequence homology, these observations suggest that surface-assisted condensate formation by AAPs of MinD/ParA-like ATPase is a general mechanism of these proteins in orchestrating the spatiotemporal organization of bacteria.

## Methods
### *M. xanthus* and *E. coli* strains and growth
*M. xanthus* strains are listed in Supplementary Table 1. Plasmids and oligonucleotides used are listed in Supplementary Tables 2 and 3, respectively. All *M. xanthus* strains are derivatives of DK1622 (WT)[60]. *M. xanthus* strains were cultivated in 1% CTT medium (1% casitone, 10 mM Tris·HCl pH 7.6, 1 mM KPO$_4$ pH 7.6, 8 mM MgSO$_4$) or on 1% CTT 1.5% agar plates[61]. Kanamycin, oxytetracycline and gentamycin were added at concentrations of 50 μg/ml, 10 μg/ml and 10 μg/ml, respectively. Growth was measured as an increase in optical density (OD) at 550 nm. *M. xanthus* was transformed by electroporation with a BioRad Micro-Pulser™ (BioRad) at 0.65 Ω. In-frame deletions were generated as described[62]. All plasmids were verified by sequencing. All strains were verified by PCR. All *M. xanthus* strains used are non-motile to allow time-lapse microscopy for several hours by deletion of *mglA*[63,64]. Plasmids for the expression of genes in *M. xanthus* were integrated by site-directed integration at the Mx8 *attB* site or the *mxan18-19* intergenic region (*mxan18-19*).

*E. coli* strains were grown in LB or 2xYT medium in the presence of relevant antibiotics or on LB plates containing 1.5% agar[65]. Plasmids were propagated in *E. coli* NEB Turbo cells (New England Biolabs) (F′ *proA*$^+$*B*$^+$ *lacI*$^q$ *ΔlacZM15/fhuA2 Δ(lac-proAB) glnV galK16 galE15 R(zgb-210::Tn10)* Tet$^S$ *endA1 thi-1 Δ(hsdS-mcrB)5*) or *E. coli* OneShot TOP10 (Invitrogen, Thermo Fisher Scientific, Waltham, USA) (F-*mcrA* Δ(*mrr-hsd*RMS-*mcr*BC) Φ80*LacZ*ΔM15 Δ *Lac*X74 *rec*A1 *ara*D139 Δ(*araleu*) 7697 *gal*U *gal*K *rps*L (StrR) *end*A1 *nup*G).

### Fluorescence microscopy and live-cell imaging
Fluorescence microscopy was performed as described[31]. Briefly, exponentially growing cultures were transferred to slides with a thin 1.0% agarose pad (SeaKem LE agarose, Cambrex) with TPM buffer (10 mM Tris·HCl pH 7.6, 1 mM KH$_2$PO$_4$ pH 7.6, 8 mM MgSO$_4$), covered with a coverslip and imaged using a temperature-controlled Leica DMi6000B inverted microscope with a 100× HCX PL FLUOTAR objective at 32 °C. Phase-contrast and fluorescence images were recorded with a Hamamatsu ORCA-flash 4.0 sCMOS camera using the LASX software (Leica Microsystems). Time-lapse microscopy was performed as described[64]. Briefly, cells were transferred to a coverslip mounted on a metallic microscopy slide and covered with a pre-warmed 1% agarose pad supplemented with 0.2% casitone in TPM buffer. Slides were covered with parafilm to retain the humidity of the agarose, and live-cell imaging was performed at 32 °C. Image processing was performed with Metamorph v 7.5 (Molecular Devices). For image analysis, cellular outlines were obtained from phase-contrast images using Oufti[66] and manually corrected if necessary. Fluorescence microscopy image analysis was performed with a custom-made Matlab script (Matlab R2018a, MathWorks) as described[34].

### Structured illumination microscopy (SIM)
SIM was performed on a temperature-controlled AxioObserver with Zeiss Elyra 7 Lattice SIM module with perfect focus and an α Plan-Apochromat 63x/1.46 oil Korr M27. Images were recorded with a pco.edge 4.2 sCMOS camera (PCO). Cells were mounted onto a 1% agarose pad as described. mCh-PomX and PomY-mCh images were acquired using a 561 nm 500 mW laser with 100 ms exposure time and 1.5% or 3% laser power, respectively. SIM images were reconstructed from 13 frames without rescaling using Zen software (Zeiss). Image processing was performed as described.

### In vivo fluorescence recovery after photobleaching (FRAP)
In vivo FRAP experiments were performed as described with a temperature-controlled Nikon Ti-E microscope with Perfect Focus System and a CFI PL APO 100×/1.45 Lambda oil objective at 32 °C with a Hamamatsu Orca Flash 4.0 camera using NIS Elements AR 2.30 software (Nikon). Photobleaching of PomY-mCh was performed with two laser pulses with 30% laser power (561 nm) and a dwelling time of 500 msec. For mCh-PomX, bleaching was performed with two laser pulses with 35% laser power (561 nm) using the same dwelling time. For every image, the total integrated cellular fluorescence in a region of interest (ROI) within the outline of the cell was measured together with the total integrated background fluorescence of an ROI of the same size placed outside of the cell. Additionally, fluorescence intensity was measured in the bleached region together with the background fluorescence of an ROI of the same size placed on the background.

After background correction, the corrected fluorescence intensity of the bleached ROI was divided by total corrected cellular fluorescence, which corrects for bleaching effects during picture acquisition. This relative fluorescence was correlated to the initial fluorescence in the bleached ROI. The mean relative fluorescence of several cells was plotted as a function of time [min]. The recovery rate for the tested fluorescent protein (t1) was determined by fitting the plotted data to a single exponential equation (y = y0+A*e-x/t) using GraphPad Prism 9.0.2 (161). Half-maximal recovery ($t_{1/2}$) was calculated from the recovery rate.

## Bacterial adenylate cyclase two-hybrid (BACTH) experiments

BACTH experiments were performed as described[67]. Relevant genes were cloned into the appropriate vectors to construct N-terminal and C-terminal fusions with the 25 kDa N-terminal or the 18 kDa C-terminal adenylate cyclase fragments of *Bordetella pertussis*. Restored cAMP production was observed by the formation of blue color on LB agar supplemented with 80 μg/ml 5-bromo-4-chloro-3-indolyl-β-D-galacto-pyranoside (X-Gal) and 0.25 mM isopropyl-β-D-thiogalactoside (IPTG). As a positive control, the leucine zipper from GCN4 was fused to the T18 and the T25 fragment. As a negative control, plasmids were co-transformed that only expressed the T18 or T25 fragment. All tested interactions were spotted on the same LB agar plates with positive control and all corresponding negative controls. Proteins were tested for their interaction in *E. coli* BTH101 (F- *cya-99 araD139 galE15 galK16 rpsL1* (StrR) *hsdR2 mcrA1 mcrB1* cells.

## Immunoblot analysis

Samples were prepared by harvesting exponentially growing *M. xanthus* cultures and subsequent resuspension in SDS lysis buffer to an equal concentration of cells. Western blot analyses were performed as described[65] with rabbit polyclonal α-PomX (1:10000)[31], α-PomY (1:10000)[31], α-PilC (1:3000)[68], or α-mCh (1:10000; Biovision) primary antibodies together with horseradish-conjugated goat α-rabbit immunoglobulin G (1:25000) (Sigma-Aldrich) as the secondary antibody. For PomY blots, protein transfer was performed with a Tris/CAPS buffer system (BioRad). Protein transfer was done with Trans-Blot Turbo buffer (BioRad) for all other blots. Blots were developed using Luminata Forte Western HRP Substrate (Millipore) and visualized using a LAS-4000 luminescent image analyzer (Fujifilm).

## Protein purification

PomY-His6 and native FtsZ were purified as described[31]. Cys-FtsZ has an N-terminal MGKCKGSG extension that can be labeled with a fluorophore after purification. Cys-FtsZ was purified as native FtsZ. To purify PomY-mCh-His6 (analogous to the active PomY-mCh fusion in vivo) plasmid pAH194 was propagated in *E. coli* ArcticExpress(DE3) RP cells (Agilent Technologies), grown in 2xYT medium with 50 μg/ml kanamycin at 30 °C to an OD600 of 0.6–0.7. Protein expression was induced with 1 mM IPTG for 16 h at 18 °C. Cells were harvested by centrifugation at 5000 × *g* for 20 min at 4 °C and washed in lysis buffer 1 (50 mM NaH2PO4, 300 mM NaCl, 10 mM imidazole, 20 mM β-mercaptoethanol, pH 8.0). Cells were then lysed in 50 ml lysis buffer 2 (lysis buffer 1, 100 μg/ml phenylmethylsulfonyl fluoride (PMSF), 1× complete protease inhibitor (Roche Diagnostics GmbH), 10 U/ml benzonase, 0.1% Triton X-100, pH 8.0) by three rounds of sonication for 5 min with a UP200St ultrasonic processor (pulse 80%, amplitude 80%) (Hielscher) on ice. Cell debris was removed by centrifugation (4150 × *g* for 45 min at 4 °C) and filtration with a 0.45-μm sterile filter (Millipore Merck). PomY-mCh-His6 was purified with a 5 ml HiTrap Chelating HP (Cytiva, 17040901) column preloaded with NiSO4 and equilibrated with lysis buffer 1. The column was washed with 20 CV (column volumes) wash buffer 1 (lysis buffer 1, 20 mM imidazole) and five CV wash buffer 2 (lysis buffer 1, 50 mM imidazole). Protein was eluted with elution buffer (lysis buffer 1, 300 mM imidazole). Protein

was then applied to a HiLoad 16/600 Superdex 200 pg (Cytiva; 28989335) equilibrated in dialysis buffer (50 mM HEPES/NaOH, 150 mM KCl, 0.1 mM EDTA, 20 mM β-mercaptoethanol, 10% (v/v) glycerol, pH 7.2). Fractions with PomY-mCh-His6 were snap-frozen in liquid nitrogen and stored at −80 °C until use. Protein concentration was determined immediately before use using Protein Assay Dye Reagent Concentrate (BioRad). Before use, proteins were dialyzed in dialysis buffer 2 (dialysis buffer 1, 50 mM KCl). PomY$^{CC}$-mCh-His6, PomY$^{CC-HEAT}$-mCh-His6, PomY$^{HEAT}$-mCh-His6, PomY$^{HEAT-IDR}$-mCh-His6, and PomY$^{IDR}$-mCh-His6 were purified as PomY-mCh-His6 using the plasmids pAH196, pAH187, pAH197, pAH195, and pAH198, respectively.

PomX-His6, PomX-His6-Cys, and His6-mCh-PomX (which is similar to the active mCh-PomX fusion in vivo) were purified under denaturing conditions and subsequently refolded. To purify them, the plasmids pEMR3, pBR39 and pBR36, respectively, were propagated in Nico21 (DE3) (NEB). Cells were grown in LB media with 50 μg/ml kanamycin at 30 °C until an OD600 of 0.5. After cooling of cells to 18 °C, expression was induced with 0.4 mM IPTG for 16 h at 18 °C. Cells were harvested by centrifugation at 4500 × *g* for 10 min at 4 °C. Cell pellets were subsequently frozen at −80 °C. For purification, cell pellets were resuspended in PomX lysis buffer (50 mM NaH2PO4, 300 mM NaCl, 10 mM imidazole, 8 M urea, pH 8.0, 1× complete protease inhibitor (EDTA free) (Roche Diagnostics GmbH), 0.4 mM Tris(2-carboxyethyl) phosphine hydrochloride (TCEP), 10 U/ml DNase 1, 100 μg/ml lysozyme) and lysed by sonication with a tip sonicator (2.30 min, 20 s pulses, 30% amplitude). After centrifugation to clear cell debris (25,000 × *g*, 45 min, 4 °C), the lysate was incubated with Ni-NTA agarose (Qiagen, Hilden, Germany) for 16 h in a 50 ml plastic tube. Subsequently, the Ni-NTA agarose was washed six times with 45 ml of PomX wash buffer (50 mM NaH2PO4, 300 mM NaCl, 20 mM imidazole, 6 M urea, 0.4 mM TCEP, pH 8.0) by resuspension and centrifugation (500 × *g*, 4 °C, 5 min). In the last step, the Ni-NTA agarose was applied to a gravity column and eluted with PomX elution buffer (50 mM NaH2PO4, 300 mM NaCl, 250 mM imidazole, 6 M urea, 0.4 mM TCEP, pH 8.0). PomX-His6-Cys containing fractions were used for labeling before refolding. Protein-containing fractions were pooled for refolding during dialysis in Slide-A-Lyzer cassette (Thermo Fisher Scientific) with a 10,000 MWCO. Refolding was achieved with four subsequent dialysis steps with decreasing amounts of urea (1st dialysis: 4 M urea for 4 h, 2nd: 2 M urea for 16 h, 3rd and 4th: 0 M urea for 4 h each) in the PomX dialysis buffer (50 mM HEPES/NaOH, 50 mM KCl, 0.1 mM EDTA, urea, 10% (v/v) glycerol, pH 7.2) at 4 °C. Proteins were frozen in liquid nitrogen and stored at −80 °C until further use.

For purification of PomX-Strep, plasmid pAH205 was propagated in *E. coli* Rosetta2(DE3) cells, grown in 2xYT medium with 50 μg/ml kanamycin, 30 μg/ml chloramphenicol, and 0.5% glucose at 37 °C to an OD600 of 0.6–0.7. Protein expression was induced with 0.5 mM IPTG for 18 h at 18 °C. Cells were harvested and washed in StrepTag lysis buffer 1 (100 mM Tris-HCl pH 8.0, 150 mM NaCl, 1 mM EDTA, 1 mM β-mercaptoethanol) and lysed in StrepTag lysis buffer 2 (StrepTag lysis buffer 1, 100 μg/ml PMSF, 1× complete protease inhibitor (Roche Diagnostics GmbH), 10 U/ml benzonase) by four rounds of sonication for 5 min with a UP200St ultrasonic processor (pulse 80%, amplitude 80%) (Hielscher) on ice. Cell debris was removed by centrifugation (4150 × *g* for 45 min at 4 °C). PomX-Strep was purified from a batch using 5 ml Strep-TactinXT 4Flow resin (iba), equilibrated with StrepTag lysis buffer 1. The resin was incubated with the cleared lysate for 18 h at 4 °C on a rotary shaker. Contaminating proteins were eluted from the resin by washing 5× with 5 ml StrepTag lysis buffer 1. The protein was eluted with 1× BXT buffer (100 mM Tris-HCl pH 8.0, 150 mM NaCl, 1 mM EDTA, 50 mM biotin) (iba). Fractions containing PomX-Strep were pooled and dialyzed against 3× 5 l of dialysis buffer at 4 °C. Proteins were frozen in liquid nitrogen and stored at −80 °C until used.

## Protein labeling

For labeling of PomX-His$_6$-Cys the protein-containing fractions were mixed with 125 µg Alexa Fluor 488 or 647 C$_5$ Maleimide (Thermo Fisher Scientific) and incubated for 2 h at RT under denaturing conditions. Subsequently, the unreacted dye was removed by repeated concentration and dilution in Amicon Ultra 100 kDa MWCO filters (Merck Millipore, Darmstadt, Germany) using dialysis buffer (50 mM HEPES/NaOH, 50 mM KCl, 0.1 mM EDTA, 6 M urea, 2.5% (v/v) glycerol, pH 7.2) at 4 °C. Afterward, the protein was refolded via dialysis as described above. The labeled protein is indicated as PomX-A488 or PomX-A647. Cys-FtsZ contains an N-terminal MGKCKGSG extension to FtsZ. For labeling of Cys-FtsZ, the protein was mixed with 125 µg Alexa Fluor 488 C$_5$ Maleimide (Thermo Fisher Scientific) for 2 h at 4 °C before unreacted dye was separated from the protein using an Econo-Pac 10DG desalting column (Biorad, Hercules, USA).

## FtsZ GTPase assay

GTP hydrolysis by FtsZ was determined using a 96-well NADH-coupled enzymatic assay with modifications. Assays were performed in reaction buffer (50 mM HEPES/NaOH pH 7.2, 50 mM KCl, 10 mM MgCl$_2$) with 0.5 mM nicotinamide adenine dinucleotide (NADH) and 2 mM phosphoenolpyruvate and 3 µl of a pyruvate kinase/lactate dehydrogenase mix (PYK/LDH; Sigma). Buffer was premixed with proteins in low-binding microtubes (Sarstedt) on ice. If necessary, a dialysis buffer was added to correct for glycerol in the assays. Then, 100 µl mixtures were transferred into transparent UV-STAR µCLEAR 96-well microplates (Greiner bio-one). The addition of 1 mM GTP started the reaction. Measurements were performed in an infinite M200PRO (Tecan) for 2 h in 30 s intervals at 32 °C, shaking at 340 nm wavelength. Assays were also performed without the addition of FtsZ or Cys-FtsZ, and these measurements were subtracted to account for background by spontaneous GTP hydrolysis and UV-induced NADH decomposition. The light path was determined to be 0.248 cm with known NADH concentrations. The extinction coefficient of NADH $\varepsilon 340 = 6220$ M$^{-1}$cm$^{-1}$ was used. Experiments were replicated at least three times with independent protein preparations.

## FtsZ sedimentation assay

Sedimentation assays were performed as described[34]. Briefly, before sedimentation experiments, FtsZ and Cys-FtsZ were applied to a clear spin at 20,000 × g for 10 min at 4 °C. Proteins at a final concentration of 10 µM in a total volume of 50 µl were premixed and incubated for 2 min at 32 °C in a buffer (50 mM Hepes/NaOH pH 7.2, 50 mM KCl, 10 mM MgCl$_2$, 10 mM CaCl$_2$, 1 mM β-mercaptoethanol) and after 10 min GTP was added to 5 mM. Samples were separated into soluble and pellet fractions by high-speed centrifugation at 160,000 × g for 1 h at 25 °C. Soluble and pellet fractions were separated, and volumes were adjusted with 1×SDS sample buffer. Fractions were separated by SDS-PAGE and stained with Instant Blue (expedion) for 10 min. Experiments were replicated at least three times with independent protein preparations.

## Negative stain transmission electron microscopy (TEM)

For TEM, proteins of interest alone or in combination were premixed in a low-binding microtube (Sarstedt) with or without PEG8000 in reaction buffer (50 mM Hepes/NaOH pH 7.2, 50 mM KCl, 10 mM MgCl$_2$, 1 mM β-mercaptoethanol). Reactions including PomX-A488 and/or PomY-mCh for Figs. 2–4 were incubated for 10 min at 32 °C. Reactions including A488-FtsZ and/or PomY-mCh for Fig. 8 were incubated for 30 min at 32 °C. 1% glutaraldehyde was added immediately before 10 µl of the reactions were applied onto an EM grid (Plano) and incubated for 1 min at 25 °C. Residual liquid was blotted through the grid by applying the grid's unused side on Whatman paper. The grid was washed twice with double-distilled H$_2$O. For staining, 10 µl of 1:4 diluted UAR-EMS uranyl acetate replacement stain (Electron Microscopy

Sciences) was applied onto the grid for 40 s and blotted through with a Whatman paper. Electron microscopy was performed with a JEM-1400 Plus electron microscope (JEOL) at 80 kV.

## In vitro pull-down experiments

For this, 10 µM protein alone or premixed as indicated were incubated for 1 h at 32 °C in reaction buffer (50 mM HEPES/NaOH pH 7.2, 50 mM KCl, 10 mM MgCl$_2$) in a total volume of 200 µl and applied to 20 µl 5% (v/v) MagStrepXT beads (iba) for 30 min. Magnetic beads were washed 10× with 200 µl reaction buffer. Proteins were eluted with 200 µl 1× BXT buffer (100 mM Tris-HCl pH 8.0, 150 mM NaCl, 1 mM EDTA, 50 mM biotin) (iba).

## Turbidity measurements

Turbidity measurements were set up using a liquid handler (Freedom EVO platform, Tecan) in flat bottom UV-STAR 384 well plates (#781801, Greiner BioOne, Frickenhausen, Germany). For assays, reaction buffer, storage buffer for adjustment, and artificial crowding agent were added to individual wells and mixed before the addition of the proteins after which the mixture was mixed again by pipetting. All reactions were set up in triplicates on the plate. Protein was thawed immediately before addition and the protein and the 384-well plate were cooled to 4 °C during the liquid handling process. All experiments were performed at a constant final buffer composition of 50 mM HEPES pH 7.25, 50 mM KCl, 3% Glycerol, 7 mM MgCl$_2$ with or without artificial crowder (final concentration given in % w/v). Subsequently, the turbidity was determined via absorption at 340 nm using a plate reader (model Spark, Tecan).

## Setup of in vitro phase separation assays by microscopy

In vitro phase separation assays were either conducted in 384 well glass bottom plates (#781892, Greiner BioOne, Frickenhausen, Germany) or in sticky-Slide 18 well (# 81818 ibidi GmbH, Gräfelfing, Germany) attached to cleaned cover slides (rinsed with ethanol and ddH$_2$O). The plates and chambers were further cleaned and hydrophilized by plasma cleaning with oxygen as the process gas (model Zepto, Diener Electronic). Directly afterward, chambers were incubated with 25–50 µl of 0.5 mg/ml PLL(20)-g[3.5]-PEG(2) (SuSoS AG, Duebendorf, Switzerland). After incubation for more than 30 min at RT, chambers were washed three times with reaction buffer (50 mM HEPES pH 7.2, 50 mM KCl, 10 mM MgCl$_2$). The concentration of crowding agent stock solution (PEG8000, Ficol 70, Ficol 400, Dextran T500) was determined by measuring the refractive indices. For assays reaction buffer, storage buffer for adjustment, and artificial crowding agent were added to the chambers and mixed by pipetting before the addition of the proteins. Phase separation was induced by adding the proteins and gently mixing by pipetting. All experiments were performed at a constant final buffer composition of 50 mM HEPES pH 7.25, 50 mM KCl, 3% glycerol, 7 mM MgCl$_2$ with or without crowding agent (final concentration given in % w/v) at a constant room temperature of 23 °C.

For experiments with PomX-A488, or experiments involving mixtures of several proteins, i.e., PomY and PomX or PomY and FtsZ or Streptavidin, experiments were performed in the presence of 4 mM TCEP to prevent protein crosslinking.

For experiments containing PomX-A488 or PomX-A647, the PomX-His$_6$-Cys labeled with the respective fluorophore was premixed with PomX-His$_6$ by rigorous pipetting to generate a protein mixture containing 30% of the labeled PomX-His$_6$-Cys and 70% of the non-labeled PomX-His$_6$. Hence, for example, "4 µM PomX-A488" corresponds to 1.2 µM PomX-A488 and 2.8 µM PomX.

To generate heat maps of the behavior of PomY and PomY-mCh, and PomX, PomX-A488 and mCh-PomX in response to different protein and PEG8000 concentrations, individual chambers were pipetted in rows of equal protein concentration starting from the lowest

crowder concentration. Images were acquired after 15 min of incubation starting from the highest crowder concentrations.

FRAP of PomY-mCh condensates was performed in the presence of 4 mM TCEP, an oxygen scavenger system (3.7 U/ml pyranose oxidase, 90 U/ml catalase, 0.8% glucose)[69] and Trolox to reduce the aging of PomY condensates. Condensates were formed under standard conditions, but particularly large condensates were selected for FRAP measurements.

For experiments with PomY-mCh and PomX-A488, 4 µM PomX-A488 was added first to the chamber then 4 µM PomY-mCh and subsequently the chamber was well mixed by pipetting up and down 7–10 times, followed by incubation for 1 h before image acquisition.

FRAP of PomY-mCh coating PomX structures was performed with 4 µM PomY-mCh and 4 µM PomX, 4 mM TCEP, and the oxygen scavenger system. Particularly large structures were selected for measurements.

In experiments for the determination of the enrichment of A488-FtsZ in PomY condensates, 4 µM PomY-mCh was first added to the chamber, then either no protein, or 1 µM EGFP, Strep-A488 or A488-FtsZ were added, and subsequently gently mixed by pipetting, followed by incubation for 30 min before image acquisition.

Experiments showing FtsZ filament formation from PomY condensates were conducted with 2% PEG8000, 4 µM PomY-mCh, 1 µM A488-FtsZ, 4 mM TCEP.

## Fluorescence microscopy of in vitro phase separation assays

All images unless otherwise mentioned were acquired on a Zeiss LSM780 confocal laser scanning microscope using a Zeiss C-Apochromat 40×/1.20 water-immersion objective (Carl Zeiss AG, Oberkochen, Germany). Tile scans and time-series were recorded using the built-in definite focus system. All two-color images were acquired with alternating illumination to avoid cross-talk. FtsZ-A488/PomX-A488 were excited using the 488 nm Argon laser, PomY-mCh or mCh-PomX constructs using the 561 nm DPSS laser. Images were usually acquired as a tile-scan and confocal Z-stack with about 10–50 planes and optimized slicing and a large field of view (212.55 × 212.5 µm or 106.3 × 106.3 µm). Images for analysis were typically recorded with a pinhole size of 14 Airy units, 512×512 pixel resolution, and a scan rate of 1.27 µs per pixel. Images for display were typically recorded with the same settings, but also at a scan zoom of 4 and a line averaging of 4 resulting in a smaller field of view of 53.1 × 53.1 µm, unless otherwise noted. Images to visualize FtsZ filament formation from PomY-mCh condensates were typically recorded with a pinhole size of 1 Airy unit, 512×512 pixel resolution, at scan zooms from 1 to 8 and a scan rate of 1.27 µs per pixel. Time-series to observe the fusion of PomY-mCh condensates were typically recorded with ~1 s intervals, while time-series of FtsZ filament formation originating from PomY-mCh condensates were typically acquired with ~15–30 s intervals.

For FRAP experiments photobleaching of large PomY-mCh condensates (diameter: ~9–15 µm) or PomY-mCh coating PomX structures was performed with ten laser pulses with 100% laser power (561 nm). The region of interest (ROI) selected for FRAP was centered in the middle of the condensates or on large PomX structures (diameter: ~3.5 µm) and time-series to record fluorescent recovery were acquired at ~2–5 s intervals. FRAP analysis was performed as in the in vivo FRAP.

## Image analysis for in vitro assays

All images were processed using Fiji[70] (version v1.53q). Images were analyzed using custom-written Fiji macros. Analysis was conducted on images with a large field of view (212.55 × 212.5 µm or 106.3 × 106.3 µm).

To quantify filament bundling of mCh-PomX or PomX-A488 in response to PEG8000 or PEG8000 and PomY-mCh, we determined the CV. The STDEV and the mean fluorescence in the PomX channel of the entire image were measured from maximum intensity projections of individual z-stacks and used to calculate the CV defined as the STDEV

over the average intensity of the image. All values obtained for each PomX concentration were normalized to the CV of the 0% PEG8000 (and no PomY-mCh) condition for that concentration.

The phase separation of PomY-mCh under various conditions was quantified by estimating the separation factor. The condensates in each image were identified and segmented from the maximum intensity projection of individual z-stacks, by means of the StarDist plugin, using the 2D pretrained model *2D_versatile_fluo*[71]. The segmented condensates were then screened, and those with an intensity below three STDEV of the background signal or a size of less than nine pixels (corresponding to a diameter of 300 nm) were removed. Intensity values inside and outside the condensates were measured separately, and used to calculate the separation factor, in percentage, as the integrated intensity of the condensates over the total integrated intensity of the image which reflects the total amount of protein that is located in the condensed phase.

To quantify the recruitment of proteins into PomY-mCh condensates, we used a similar approach, with the following differences. Instead of using the maximum intensity projection of the images, the slice that was best in focus, i.e., with the maximum STDEV of the intensity, was selected for analysis of enrichment and separation rate. The PomY-mCh channel was used to determine the location and size of the condensates, as before. The segmentation was used to estimate the STDEV and the average of the fluorescence intensity in the condensates as well as the average fluorescence intensity in the background for all five signals of interest, PomY-mCh, bleedthrough control (no added protein), EGFP, Strep-A488 and A488-FtsZ. The separation factor for PomY-mCh, under all these conditions, was estimated as described above. The enrichment of EGFP, Strep-A488 and A488-FtsZ in the condensates was computed as the ratio between the average intensity of the intensity in all the condensates and the average intensity in the remaining areas of the field of view, that is the background. For normalization, the enrichment of the respective bleedthrough control (no added protein) was substracted from each value. The average enrichment of all bleedthrough controls was 1.7 ± 0.5 and thus close to equal partitioning, suggesting that bleedthrough is very low. To quantify heterogeneity in the enrichment of EGFP, Strep-A488 and A488-FtsZ in PomY-mCh condensates the CV of the intensity in the condensates in the 488 channel was determined which is defined as the STDEV over the mean fluorescence intensity in the condensates.

To quantify FtsZ filament polymerization and bundling by PomY-mCh condensates the average fluorescence intensity in the FtsZ-488 channel was obtained from maximum intensity projections of individual z-stacks.

## Statistics

The mean and STDEV were calculated with GraphPad Prism 9.0.2 (161). Localization patterns from fluorescence microscopy data, cell length distribution, and constrictions frequency were quantified based on the indicated number of cells per strain in three independent experiments unless otherwise indicated. Scatter dot plots were generated with GraphPad Prism 9.0.2 (161). Statistical analysis was performed with GraphPad Prism 9.0.2 (161). All data sets were tested for significant differences using a two-way ANOVA with multiple comparisons. A $P$-value < 0.01 was used to determine statistically significant difference. All $P$-values are provided in the figure legends.

## Bioinformatics

PomY orthologs were identified in a best-best hit reciprocal BlastP analysis from fully-sequenced genomes of myxobacteria[72]. Protein domains were predicted using SMART[73] and Interpro[74]. Disorder prediction was performed with IUPred2A using the IUpred2 and Anchor2 algorithms[75,76]. Isoelectric points were predicted with the ProtParam tool using the Expasy webserver[77].

## Plasmid constructions

pBR plasmids were propagated in *E. coli* OneShot TOP10 (Invitrogen, Thermo Fisher Scientific, Waltham, USA). Seamless assembly was used to introduce larger fragments into plasmid backbones. DNA fragments and vector backbones were amplified by PCR with primers that contained 15–20 bp overlaps between adjacent fragments. The PCR products were combined using GeneArt Seamless Cloning and Assembly Enzyme Mix (Thermo Fisher Scientific, Waltham, USA) according to the manufacturer's instructions. All pDS, pAH, pKA and pPK plasmids were propagated in *E. coli* NEB Turbo cells (New England Biolabs). These plasmids were generated using standard restriction enzymes and ligated using T4-Ligase (New England Biolabs). All DNA fragments, generated by PCR were verified by sequencing.

For the introduction of point mutations or small peptide sequences, blunt end cloning was employed. The entire vector was amplified with two primers extended by the sequence to be introduced. After PCR the product was digested with DpnI and the blunt ends of the PCR products were phosphorylated using T4 Phosphokinase and subsequently ligated with T4 DNA Ligase (Thermo Fisher Scientific).

For pDS79 *mCh* was amplified from pKA28 with mCherry fwd Start XbaI and mCherry stop rev HindIII and cloned into pKA45 with XbaI and HindIII.

For pDS132 *pomY^{CC-HEAT}-mCh* was amplified from pSWU30-*pomY^{CC-HEAT}-mCh* with MXAN_0634-6 and mCherry stop rev HindIII and cloned into pKA45 with XbaI and HindIII.

For pSWU30-*pomY^{CC-HEAT}-mCh pomY^{CC-HEAT}* was amplified from genomic DNA with DS1 and DS2 and cloned into pKA28 with EcoRI and BamHI.

For pDS331 *pomY-mCh* was amplified from pDS7 with DS14 and DS40 and cloned into pMR3690 with NdeI and EcoRI.

For pAH14 *mCh* was amplified from pKA28 with KA477 and KA478 and cloned into pET45b+ with BamHI and HindIII.

For pAH180 Cys-*ftsZ* was amplified from genomic DNA with AH159 and KA500 and cloned into pET24b+ with NdeI and BamHI.

For pAH187 *pomY^{CC-HEAT}-mCh* was amplified from pDS132 with NdeI PomY fwd and DS274 and cloned into pET24b+ with NdeI and HindIII.

For pAH194 *pomY-mCh* was amplified from pDS8 with DS14 and DS274 and cloned into pET24b+ with NdeI and HindIII.

For pAH195 *pomY^{HEAT-IDR}-mCh* was amplified from pDS7 with DS12 and DS274 and cloned into pET24b+ with NdeI and HindIII.

For pAH196 *pomY^{CC}-mCh* was amplified from pPK29 with DS14 and DS274 and cloned into pET24b+ with NdeI and HindIII.

For pAH197 *pomY^{HEAT}-mCh* was amplified from pPK45 with DS12 and DS274 and cloned into pET24b+ with NdeI and HindIII.

For pAH198 *pomY^{IDR}-mCh* was amplified from pDS7 with DS271 and DS274 and cloned into pET24b+ with NdeI and HindIII.

For pAH205 *pomX-Strep* was amplified from gen. DNA DK1622 with DS278 and DS276 and cloned into pRSF-Duet1 with NcoI and HindIII.

For pAH225 *mCh* was amplified from pDS7 with mCherry fwd NdeI and DS40 and cloned into pMR3690 with NdeI and EcoRI.

For pPK29 *pomY^{CC}* was amplified from pDS7 with DS14 and CC(nat) linker rev BglII and cloned into pKA45 with NdeI and BamHI.

For pPK30 *pomY^{HEAT}* was amplified from pDS7 with 17 HEAT(nat) XbaI fwd and 18 HEAT(nat) linker rev BglII and cloned into pKA45 with XbaI and BglII. For cloning pKA45 was opened with XbaI and BamHI.

For pPK36 *pomY^{CC-HEAT}* was amplified from pDS7 with the primers DS14 and 18 HEAT(nat) linker rev BglII and cloned into pKA28 with NdeI and BamHI. The obtained plasmid was used as a template for the following PCR with the primers DS4 and mCherry stop rev HindIII and cloned into pKA45 with XbaI and HindIII.

For pPK45 *pomY^{HEAT}-mCh* was amplified from pPK30 with the primers 27 HEAT fwd and mCherry stop rev HindIII and clone into pDS7 with XbaI and HindIII.

For pPK47 *pomY^{HEAT-IDR}-mCh* was amplified from pDS7 with the primers 27 HEAT fwd and mCherry stop rev HindIII and cloned into pDS7 with XbaI and HindIII.

For pPK46 *pomY^{IDR}-mCh* was amplified from pDS7 with the primers FM4 and mCherry stop rev HindIII and cloned into pDS7 with XbaI and HindIII.

pBR35_pET45b_His6-PomZ-mCherry was generated by seamless assembly. The backbone was amplified by PCR with primers BR68 and BR70 from pKA3, and the insert was amplified from pDS43 with primers BR64 and BR69.

pBR36_pRSF-Duet-1_His_6-mCh-PomX encodes *His_6-mCh-pomX* and was amplified by PCR from the plasmid pDS37 with the primers BR71 and BR72 and subsequently ligated using blunt end cloning.

pBR38_pET24b_PomY-YFP-His6 was generated by seamless assembly. The backbone was amplified by PCR with primers BR62 and BR75 from pDS3, and the fragment encoding *pomY-yfp* was amplified from pDS46 with primers BR64 and BR77.

pBR39_pET24b- PomX-His-GS-Cys encodes *pomX-His_6-Cys* and was amplified by PCR from the plasmid pEMR3 with the primers BR87 and BR153 and subsequently ligated using blunt end cloning.

pBR72_pET24_PomY-mCherry-His6 encodes *pomY-mCh-His_6.* This plasmid was generated by seamless assembly. The backbone was amplified from pBR38_pET24b_PomY-YFP-His_6 with primers BR62 and BR185, the insert encoding *mCh* was amplified from pBR35_pET45b_His6-PomZ-mCherry with primers BR64 and BR184.

pDS121, and pDS123. For both plasmids *pomY* was amplified from genomic DNA using primers Mxan0634 fwd BACTH and Mxan_0634-17 pomY-rev. The resulting PCR fragment was digested with KpnI and XbaI and cloned into pKT25 and pKNT25.

pDS261, pDS262, pDS263, and pDS264. For all four plasmids, *pomY^{CC}* was amplified from genomic DNA using the primers Mxan0634 fwd BACTH and Mxan0634 Coil KpnI BACTH. The resulting PCR fragment was digested with KpnI and XbaI and cloned into pUT18, pUT18C, pKT25, and pKNT25 using the same restriction sites.

pDS265, pDS266, pDS267, and pDS268. For all four plasmids, *pomY^{HEAT}* was amplified from genomic DNA using the primers Mxan0634 HEAT XbaI BACTH and Mxan0634 HEAT KpnI BACTH. The resulting PCR fragment was digested with KpnI and XbaI and cloned into pUT18, pUT18C, pKT25, and pKNT25 using the same restriction sites.

pDS269, pDS270, pDS271, and pDS272. For all four plasmids, *pomY^{IDR}* was amplified from genomic DNA using the primers Mxan0634 Pro XbaI BACTH and Mxan_0634-17 pomY-rev. The resulting PCR fragment was digested with KpnI and XbaI and cloned into pUT18, pUT18C, pKT25, and pKNT25 using the same restriction sites.

## Reporting summary

Further information on research design is available in the Nature Portfolio Reporting Summary linked to this article.

## Data availability

The authors declare that all data supporting this study are available within the article, its Supplementary Information file or in the Source Data file. Source data are provided with this paper.

## Code availability

The custom-made Matlab script (Matlab R2018a, MathWorks) for the analysis of bacterial fluorescence microscopy images is available on GitHub [https://github.com/SBergeler/ImageAnalysisMyxo]. All other codes are available from the corresponding authors upon request.

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

## Acknowledgements

We thank Giovanni Cardone (MPIB Imaging Facility) and Silke Bergeler for assistance with image analysis, Leopold Urich (MPIB Core Facility) for assistance with the operation of the liquid handler, Gabriele Malengo (MPITM Core Facility for Flow Cytometry and Imaging) for help with structured illumination microscopy, Kerstin Andersson and Katharina Nakel for assistance with protein purification, and Lei Kai, Philipp Blumhardt and Leon Babl for helpful discussions. B.R. was supported by a fellowship from the German Research Council (DFG) through the Graduate School of Quantitative Biosciences Munich and in part by the National Science Foundation grant no. PHY-1734030. P.S. and L.S.-A. acknowledge financial support by the DFG through the Transregio 174 "Spatiotemporal dynamics of bacterial cells" (Project ID No. 269423233) and the Max Planck Society. P.S. acknowledges the support of the research network MaxSynBio via a joint funding initiative of the German Federal Ministry of Education and Research (BMBF). We thank the undergraduate class "Building the Synthetic Cell", University of Michigan for very constructive and helpful feedback.

## Author contributions

The order of the two shared first authors is alphabetical and has no further meaning. B.R., D.S., P.S. and L.S.-A. conceived the study. B.R. and D.S. designed in vitro experiments; D.S. and P.K. designed in vivo experiments. B.R., D.S., T.H., A.H., P.K. and F.M. performed experiments. B.R. and D.S. analyzed data and wrote the original manuscript draft. B.R., D.S., P.S. and L.S.-A. reviewed and edited the manuscript. All authors discussed the results and revised the manuscript.

## Funding

## Competing interests

The authors declare no competing interests
