## [Peer Review File · Nature Communications]

Biomolecular condensate drives polymerization and bundling of the bacterial tubulin FtsZ to regulate cell divisionREVIEWER COMMENTS

Reviewer #1 (Remarks to the Author):

This paper reports studies of the PomXYZ system, which serves to localize the Z ring in *Myxococcus*. The *in vitro* structures of X and Y separately and mixed are examined by fluorescence LM, concluding that X forms a condensate-like scaffold that concentrates Y. Y itself is shown to form condensate clusters, which are interpreted as the active agent in nucleation of FtsZ and localization of the Z ring. I have two major concerns compromise the major interpretations. See points 8-9 and 12-15. Since I don't find the major interpretations to be convincing, I cannot recommend publication.

1. Ref 24 is the Bisson-Filho study of treadmilling in *Bs*. It would be fair to add the accompanying Yang.. Xiao ref for treadmilling in *Ec*
2. Lines 100-106 list several points for how the Pom's interact with each other and with FtsZ. These probably come from refs 30 and 33. It would be helpful to state which is more important for each association.
3. Is there an estimate for the average number of molecules of PomXYZ per cell? This would be very helpful (essential) for understanding the structures. This can perhaps be deduced from the 1.0-1.4 μM concentration of X and Y in clusters, but this raises a problem. At this concentration X or Y subunits will be on average ~ 100 nm apart. What holds the condensate together?
4. X does not show any exchange by FRAP, which I think is unusual for a condensate. Condensates typically show exchange within the cluster, and with the solution.
5. Does 2c contradict 2d? In 2c there is only a very slight turbidity at the highest concentrations of X and PEG. However, 2d shows prominent filamentous clusters. Why don't these cause turbidity? These large, irregular aggregates and fibrils seen by LM are likely composed of some irregular network of the thin X filaments seen by EM in previous studies, but based on the estimated concentration the thin X filaments in the fibrils is probably very sparse. I would suggest reserving the term filaments for the thin filaments seen by EM, and fibrils for the LM structures. A brief discussion of the relation of filaments to fibrils would help.
6. The BACTH data (ED Fig. 7) are confusing. Full length PomY does not seem to interact with itself, raising a question of how it could make a condensate. Also, a couple of the domains do not interact with full length Y. Moreover, to make a condensate, one needs more than pairwise interactions. If the CC and IDR of one molecule could both bind to the CC and IDR of another, this would just give a dimer. This needs some thought.
7. Fig. 3 is really complicated. *In vivo* X forms spherical clusters on its own, and Y can join these clusters if present. *In vivo* X forms irregular fibrils on its own, and Y forms large spherical clusters. When mixed (in 2% PEG) the large Y clusters disappear, and are redistributed, at least somewhat, as small spherical aggregates on the X fibrils. I don't see that this *in vitro* phenomenon is related to the very different structures *in vivo*.
8. Fig. 4 shows a clever and important result. Y cannot make clusters without X at its normal expression level, but when overexpressed it can. Moreover, the Y cluster could apparently attract and localize FtsZ and achieve division at that site. This is an important new result, but I am not convinced by the anecdotal data shown. In particular, I wonder how frequently the Y clusters lead to division. Two examples of division at Y are shown in Fig. 4d, however these are at very weak Y clusters. Two examples of strong Y clusters in Fig. 4c do not seem to lead to division. Moreover, X-minus cells still divide at a reduced frequency in the complete absence of Y clusters (at the normal expression levels). I think we need a large survey with solid statistics to make a convincing case that the division site is localized at Y clusters.

9. "cell division site positioning solely depends on the presence of a PomY condensate, irrespectively of the presence of PomX." As noted above, this broad interpretation needs extensive counting and statistics. From the present images it is not convincing.

10. L 1309-10: "PomY^{heat} and PomY^{idr} stimulate phase separation." Y-heat may show a slight stimulation based on the numbers in Fig. 5b (although the single image looks less than cc); cc-idr does not seem to be tested. I guess this conclusion for idr may be based on wt being stronger than cc-heat; this should be spelled out.

11. Fig. 6 shows that FtsZ is enriched in Y clusters, but surprisingly Strep was also enriched. FtsZ was 12x, while Strep was lower at 3x. Still, this suggests a lack of specificity in the recruitment, which may be useful to note. (Minor suggestion: In 6a the top panel is labeled Protein, which suggested to me that it may be measuring total protein. It would be better to label it A488.)

12. Interpreting Fig. 6d the authors state: "Remarkably, filamentous FtsZ structures emerged from PomY-mCh condensates within minutes, later on forming an extended filamentous network." This is a very important conclusion, central to the study, but I am not convinced. In the time point at 11 min, the majority of the FtsZ bundles seem to arise spontaneously in the solution, not from Y clusters. This is even more obvious in the early stages of video 3. I see no hint that the Y clusters nucleate the bundles. There is only a vague hint that they attach to bundles when they contact, but that would have nothing to do with the interpretation that Y clusters are involved in nucleating FtsZ.

13. Note also that these beautiful time lapse movies are on a much slower scale than FtsZ nucleation. FtsZ protofilaments reach a plateau of assembly within ~10 s of nucleation. What these LM movies are showing is bundling of these protofilaments, which progresses much more slowly than nucleation and initial assembly of protofilaments.

14. My observation that FtsZ bundles appear mostly outside Y clusters seems contradicted by Fig. 6e, which shows that no FtsZ bundles appear in the absence of Y. However, this probably means only that soluble Y mediates FtsZ bundling. In the absence of Y, FtsZ assembles protofilaments, which are not resolved by LM. In the presence of Y, the protofilaments are slowly clustered into discrete bundles, which are resolved by LM. This should be tested by repeating the assay with lower concentrations of Y, going below the C_{eff} for cluster formation. It would also be interesting to test the effect of PEG, lowering it to below cluster formation.

15. In previous papers the Sogaard-Andersen lab has presented high quality negative stain EM. This would be an ideal tool to study the bundling as a function of Y, Y-cc, and PEG concentrations. EM might also reveal details of the substructure of Y and Y-cc clusters.

Reviewer #2 (Remarks to the Author):

In this manuscript, Ramm and colleagues investigated how the PomX/Y/Z complex directs the formation of the Z-ring, a structure formed through the polymerization of FtsZ for bacterial division. Using *Myxococcus xanthus* as a model organism, they found that the cluster of PomY, but not of PomX, dynamically exchanged molecules with the cytoplasm and was assembled before and disassembled after the cell division. In the presence of a crowding reagent such as PEG8000, purified PomX polymerized into filaments, whereas PomY phase-separated into liquid droplets. The filaments concentrated PomY and induced deformation of PomY condensates. The CC region mediated the LLPS of PomY, whereas the HEAT and IDR regions promoted the LLPS through homotypic interactions. All the regions contributed to the full interaction affinity with PomX. PomY condensates formed in PomX-depleted cells through overexpression was sufficient to induce cell division. Furthermore, PomY condensates enriched FtsZ and promoted its GTP-dependent polymerization *in vitro*. The authors thus proposed a model in which the cluster of PomX filaments in cells enriched PomY and induced its local phase separation to form a single condensate, which further recruited FtsZ and promoted its polymerization to form the Z-ring.

While the authors proposed an interesting model for the mechanism of Z-ring formation from the angle of LLPS, experimental results still do not sufficiently support their claims due to lack of controls, solid evidence, or detailed information. I do not suggest publication of the manuscript in its current form, unless the concerns listed below can be properly addressed.

Major concerns:

1. In the images presented in Figure 3a and 3b/Video 2, fluorescent signals of the A488 and mCh channels superimposed nicely at fibrous structures and condensates. While this may reflect an extensive interplay between PomX and PomY, it might be due to a leak of fluorescent signals as well, unless the latter possibility is excluded. I would suggest that the authors demonstrate that the signals are not interdependent by bleaching A488 and mCh, respectively, in different small regions of fibers/condensates, followed by dual-color imaging of the entire field. Alternatively, mCh could be mixed with PomX-A488 fibers and an irrelevant A488-labelled protein with PomY-mCh condensates, followed by imaging. If leaking is not a problem, they might want to clarify whether PomY droplets concentrated PomX to in turn facilitate PomX polymerization. In Video 2, PomX filaments appear to display enhanced polymerization after contacting with droplets.
2. In a previous publication (PMID: 28486132), the group shows through TEM that PomX alone forms 8.3-nm-thick filaments, which are further bundled into 150-nm ones in the presence of PomY. Fluorescent images in Figure 3a indicate that the fibrous structures formed with PomX and PomX+PomY in the presence of PEG differ markedly from those in the absence of PEG. I would suggest that the authors at least examine the samples of 1% PEG with TEM to clarify how the induced phase separation impacts the filamentous structures.
3. Figure 4d: the conclusion that PomY condensates are functional and support cell division in the absence of PomX is drawn without the support of quantification results. Could the authors show how tightly the cell division event is correlated with PomY punctum? What happens in cells with multiple puncta?
4. The enrichment of Strep-A488 into PomY condensates (Figure 6a) raises the question whether the enrichment of A488-FtsZ is specific, which needs to be further clarified. As pointed out earlier, fluorescence leaking might be a problem. If this possibility is excluded, I would suggest that the authors clarify whether the observed partitioning was due to the A488 label. Condensates usually display selectivity on proteins that can be partitioned in them. Using FtsZ and a control protein tagged with a fluorescent protein, e.g., GFP, would help to clarify the issue. Although the authors managed to show less enrichment of Strep-A488 (Figure 6b), I am not convinced by the quantification results. In the images in Figure 6a, I do not see that Strep-A488 is much less enriched in the droplets than A488-FtsZ, as implicated in Figure 6b. I also don't understand why the authors suddenly chose to use selected optical sections, instead of maximum intensity-projected images, and a different calculation method to measure the partition efficiency of the two proteins (Figure 6b), while measuring the usual separation factor to show the partition efficiency of PomY-mCh (Figure 6c). Without a clear demonstration of the partition specificity, the polymerization results of FtsZ (Figure 6d-g) could just be artifacts of its non-specific enrichment.
5. According to Video 3 (Figure 6d), FtsZ polymerizations appear to occur from short pre-existing seeds but not from the condensates as described by the authors (line 405). If PomY condensates served as nucleation centers (line 418), one would expect that the condensates produce radial arrays of FtsZ bundles, analogous to the situation of abLIM1 condensates on actin (PMID: 35858327). Therefore, PomY may only bundle spontaneously-polymerized FtsZ filaments, which were too small for light microscopy to reveal. EM examinations on samples as those in Figure 6e will clarify whether FtsZ polymerized autonomously in the absence of PomY and how the bundles formed in the presence of PomY look like. Light scattering assays (e.g., PMID: 9922245) would also be very informative to understand how PomY impacts FtsZ polymerization. A nucleator should usually induce polymerization at FtsZ concentrations below the critical concentration of spontaneous polymerization. In addition, as PomY was gradually recruited to the elongating FtsZ bundles following the polymerization process (Video 3), its colocalization with FtsZ bundles does not necessarily underscore the liquid nature of PomY condensates (line 415). The authors need to tone down this or provide evidence that the bundle-associated PomY molecules possess liquid properties, e.g., by monitoring the behavior of photoconverted PomY fusion proteins like demonstrated in (PMID: 35858327).
6. I would suggest that authors briefly discuss how the PomY condensate disintegrates during or after cytokinesis.

Minor concerns:

7. Line 164: "In snapshots of 1000s of cells": what does 1000s mean here ?

8. Extended Data Figure 3b: full names of the species should be listed instead of abbreviations to increase readability.
9. Figure 3a: the white arrow in the PomY channel of the bottommost panel should be either removed or relocated to the merged channel to indicate a condensate.
10. It is not clear whether the initial time point (0 min) of Figure 3b (and video 2) started after the 1-hour incubation (as indicated in the diagram in Figure 3a) or quickly after the mixing of the reagents.
11. Figure 4c: one of the two arrows at 60 min does not point to a punctum.
12. Figure 7: The model does not include the bundling effect of PomY on PomX filaments. The authors might want to consider adding this to the revised manuscript.
13. Line 1359: "under the conditions in d and de Scale bar".

Reviewer #3 (Remarks to the Author):

In this well-written manuscript, the Andersen and Schwille groups have collaborated to explore the PomXYZ system, which aids in the positioning of the Z-ring and therefore division in Myxo. Specifically, their focus is on the phase separating properties of PomY and its interactions with PomX and FtsZ filaments. The researchers first characterize the growth and division of PomX and PomY clusters through mCh labeled PomX and PomY, finding that while the PomX cluster divides between daughter cells without dissolution, PomY foci dissolve into the cytoplasm following division. Next the authors demonstrated through in vitro fluorescent imaging and FRAP that PomX forms a cluster of filaments while PomY undergoes phase separation. They go on to show that Csat of PomY can be lowered through PEG8000 induced crowding and/or the presence of PomX. In vivo, they then show that over-expression of PomY allows for condensates to form without PomX that can still position the Z-ring. To characterize the interactions between PomX and PomY, the authors produce PomY domain fragments and show that all three domains bind PomX through pull-down experiments. These PomY fragments were then studied in vitro and in vivo, analyzing which domains and combinations of domains best undergo phase separation. Finally, the authors show PomY condensates can locally induce/stimulate FtsZ polymerization, and that PomY subsequently coats and condensates around the FtsZ bundles. This impressive body of work significantly contributes to our knowledge of the PomXYZ system; primarily through their identification of the phase separation behavior of PomY, its surface-assisted condensation provided by PomX, and the localized induction of FtsZ polymerization both in vivo and in vitro. The authors are successful in effectively introducing the most relevant information about condensates to their study and in contrasting the PomXYZ system from other means of division localization in bacteria. This work is significant as it details a thus-far understudied means of divisome positioning in bacteria. The work's significance extends beyond the immediate scope of bacterial division and has implications on the potential role of phase separation for the regulation of other ParA/MinD ATPase systems. These systems are so widespread that this paper will pave the way for future studies of phase separation roles in bacterial subcellular organization. I think the paper is suitable for publication as is. But there is, as always, a few major (to better justify conclusions made) and several minor suggestions that would improve the manuscript.

Major Issues:

1. The authors claim that while PomY clusters depend on PomX to form, PomX clusters are independent of PomY. Data in columns 1 and 3 in figure 3B slightly conflict with the claim that PomY has no effect on PomX clusters. At 0% PEG8000, PomX mixed with PomY is much more filamentous than PomX alone. Likewise, at 4% and 10% PEG8000, PomX mixed with PomY forms more clumped and aggregated rather than long and thin filaments compared to PomX alone. It is a potentially important finding that PomY may affect PomX filament morphology, the researchers should provide an explanation as to why these PomX morphologies differ.
2. (Line 311) "Stunningly, not only PomX/PomY-mCh clusters but also the "pure" PomY-mCh clusters were proficient in determining the site of cell division (Fig. 4d), demonstrating that cell division site positioning solely depends on the presence of a PomY condensate, irrespectively of the presence of PomX." This is a strong statement coming from only two divisions shown in 4C. Please quantify and

provide statistics for the locations of pure PomY clusters and the locations of division events across a cell population (similar to what was done in 6h).

3. Based on the data in Fig. 5C, the authors claim that both the IDR and HEAT domains of PomY stimulate phase separation. However, the PomY CC-HEAT fragment alone separates better at PEG8000 2% than the full PomY CC-HEAT-IDR protein. While it's true at 4% PEG8000 the full PomY protein separates more completely, this discrepancy casts doubt on the claim that the IDR region is improving separation. I would argue the IDR assists in fluidizing the condensates and makes them larger due to increasing solvent interactions, as opposed to stimulating condensation (Line 347). Comparisons of droplet fluidity, such as FRAP, could provide insight.

4. I am very excited about the PomY(HEAT-IDR) mutant (Fig 5C-D, specifically). Although the authors don't explicitly mention this in the paper, could it be that this LLPS- mutant still forms the oblong structure in vivo because it is still interacting with PomX (pre-wetting interactions)? If so, do you think this mutant has allowed you to separate pre-wetting interactions from PomY's condensation activity?

5. (Line 372-376) The claim that LLPS of PomY is necessary for Z ring positioning by the PomXYZ system hinges on the idea that the cluster seen in Figure 5D with PomY [HEAT-IDR-mCh] is not a condensate given its oblong shape. This evidence of a lack of phase separation is weak given that Figure 1B already explored the fact that the PomY condensates form oblong shapes with long and short axes. Perhaps FRAP could be used to show the PomY [HEAT-IDR] cluster lacks fluidity? Or PomX could be mixed with PomY HEAT-IDR in vitro (as in Fig 3A) to show that PomY is associating with PomX, but not phase separating? This would be a nice addition because it shows that with this truncation mutant you have likely separated the initial (pre-wetting) PomX association and surface-assisted condensation activity of PomY.

Minor Issues:

1. (Line 42) "Although bacterial cells generally lack organelles...." Protein-based organelles are ubiquitous in bacteria and archaea. Please change to – "Although bacterial cells generally lack membrane-bound organelles."

2. (Lines 45-46) The authors should change "LLPS" to just "phase separation" or "condensation". Many eukaryotic condensates indeed form via phase separation, but many (most) of these papers do not conclusively show that Liq-Liq phase separation is the actual mechanism.

3. Why weren't the 6-His tags cleaved off? These tags have been shown in several papers to have drastic effects on all features of condensates (assembly, maintenance, dissolution). The authors are making relative comparisons among protein variants that all have the tags, so I think the results still support the conclusions made. However, I'd be cautious of any definitive statements regarding empirical measurements such as a partitioning coefficient or csat for these proteins.

4. Figure 1: Do you think PomZ pulling on the PomXY cluster (specifically PomX) is responsible for the elongated PomX structure at mid-cell? For example, if you delete the N-terminal peptide on PomX do you still get an elongated structure? It would be nice to propose a hypothesis for the elongation of the PomX cluster, as one is provided for the elongation of the PomY cluster.

5. Figure 6a: Do you have any hypotheses as to why partitioning of FtsZ into PomY condensates is heterogeneous? It's intriguing. Do you think FtsZ in the PomY condensates is dynamic (liquid-like?) before the addition of GTP?

6. Figure 6: Why not add PomX to the in vitro reconstitution with PomY and FtsZ? I know you show in vivo that overexpressed PomY is necessary and sufficient. But since you have all three labelled and purified, did you try putting all 3 together? Seemed like the obvious thing to do next.

7. Do you think overexpression of both PomY and FtsZ would allow for multiple division events in a PomX deletion?

8. Any reason why two-color labelling of PomX and PomY wasn't performed, as the authors did previously in the 2017 paper? Would have been helpful for several points made throughout this paper, such as co-localization data and pairwise comparisons of PomX-PomY cluster shape. I have the same question for two-color labelling of PomY and FtsZ.

9. (Line 567) "PomX-PomX" should be "PomX-PomY"

Reviewer #1 (Remarks to the Author):

This paper reports studies of the PomXYZ system, which serves to localize the Z ring in Myxococcus. The in vitro structures of X and Y separately and mixed are examined by fluorescence LM, concluding that X forms a condensate-like scaffold that concentrates Y. Y itself is shown to form condensate clusters, which are interpreted as the active agent in nucleation of FtsZ and localization of the Z ring. I have two major concerns compromise the major interpretations. See points 8-9 and 12-15. Since I don't find the major interpretations to be convincing, I cannot recommend publication.

1. Ref 24 is the Bisson-Filho study of treadmilling in Bs. It would be fair to add the accompanying Yang.. Xiao ref for treadmilling in Ec.

Response: We thank the reviewer for pointing this out. We have now included this reference as #25 (line 77).

2. Lines 100-106 list several points for how the Pom's interact with each other and with FtsZ. These probably come from refs 30 and 33. It would be helpful to state which is more important for each association.

Response: We apologize for not making this clearer. We have now included references for each statement in line 100-108.

3. Is there an estimate for the average number of molecules of PomXYZ per cell? This would be very helpful (essential) for understanding the structures. This can perhaps be deduced from the 1.0-1.4 μM concentration of X and Y in clusters, but this raises a problem. At this concentration X or Y subunits will be on average ~ 100 nm apart. What holds the condensate together?

Response: As experimentally determined in Fig. 1a-b, average PomY and PomX clusters are spheroidal. Using the dimensions of the average spheroids (as determined in Fig. 1a-b), we calculate their volumes using the formula for the volume of an ellipsoid. This gives an average volume of a PomY cluster of $9.6 \times 10^6 \text{nm}^3$ and for PomX $12.0 \times 10^6 \text{nm}^3$.

Based on quantitative immunoblots, we previously determined the average number of molecules per cell to ~ 850 for PomY and ~ 200 for PomX (Schumacher et al, 2017). Using quantitative fluorescence microscopy, we found that $\sim 15\%$ of the total PomY fluorescence and $\sim 47\%$ of the total PomX fluorescence are in clusters (Schumacher et al, 2017). Thus, average clusters contain ~ 130 PomY molecules and ~ 100 PomX molecules.

Given the sizes of PomY (682 aa residues) and PomX (404 aa residues) and assuming they are globular/unfolded, then we calculate hydrodynamic radii of 3.6/9.1nm and 3.1/6.7nm for a PomY and PomX molecule, respectively. If we assume that each molecule in a cluster is not fixed in space but has some degree of freedom to move, e.g. the distance equal to its hydrodynamic radius, this gives "effective" hydrodynamic radii of 7.2/18.2nm and 6.2/13.4nm for a PomY and a PomX molecule, respectively. Using the formula for the volume of a sphere, each PomY molecule then occupies a volume of $1.6/25 \times 10^3 \text{nm}^3$ and each PomX molecule occupies a volume of $1.0/10 \times 10^3 \text{nm}^3$. Multiplying by the experimentally estimated number of molecules per cluster this gives a volume of $0.2/3.3 \times 10^6 \text{nm}^3$ for the PomY cluster and $0.1/1.0 \times 10^6 \text{nm}^3$ for the PomX cluster. Importantly, these back-of-the-envelope calculated cluster volumes are on the same order of magnitude as the experimentally determined average volumes of PomY and PomX clusters of $9.6 \times 10^6 \text{nm}^3$ and $12.0 \times 10^6 \text{nm}^3$, respectively, supporting that the PomY condensates and the PomX scaffold are held together by Y/Y and X/X interactions, respectively.

4. X does not show any exchange by FRAP, which I think is unusual for a condensate. Condensates typically show exchange within the cluster, and with the solution.

Response: We apologize for the confusion. Despite the overall similar structural features of PomX and PomY, all our *in vitro* data suggest that PomX does not phase separate. Rather PomX *in vitro* self-assembles to form filaments and does not form spherical condensates. Therefore, we conclude in line 240-243 “Together with the observation that PomX clusters grow but do not exchange molecules with the cytoplasm *in vivo*, these data strongly suggest that PomX neither undergoes LLPS *in vitro* nor *in vivo*. Rather PomX forms protein filaments, which *in vivo* assemble to form the single PomX cluster”. To make the difference between the behaviors of PomX and PomY clearer, we have divided the main text with the *in vitro* description of the two proteins into two sections with distinct headlines (line 197 and line 244) and two figures (Fig. 2 for PomX and Fig. 3 for PomY).

5. Does 2c contradict 2d? In 2c there is only a very slight turbidity at the highest concentrations of X and PEG. However, 2d shows prominent filamentous clusters. Why don't these cause turbidity? These large, irregular aggregates and fibrils seen by LM are likely composed of some irregular network of the thin X filaments seen by EM in previous studies, but based on the estimated concentration the thin X filaments in the fibrils is probably very sparse. I would suggest reserving the term filaments for the thin filaments seen by EM, and fibrils for the LM structures. A brief discussion of the relation of filaments to fibrils would help.

Response: We apologize for not being clearer. The turbidity is a measure for light scattering in the solution, which should depend on the extinction cross-section of a particle and the particle concentration (Kleizen et al., *Filtration&Separation* 9, 897-901 (1995)). In the absence of PEG8000 PomX forms individual thin filaments, which do not have a large extinction cross-section. At increasing PEG8000 concentrations, PEG8000 acts as a crowder, causing depletion attraction which leads to bundling of the PomX filaments into larger filamentous structures. While these bundled PomX structures are of similar size or even larger than the PomY condensates at 4% PEG8000, they are, as the reviewer suspected, much sparser than the PomY condensates (see images below), and therefore they do not cause an increase in turbidity.

In the original submission, we failed to clarify that the large filamentous PomX structures formed in the presence of PEG8000 are scarce. To clarify this issue we have included lower

magnification images of PomX previously only shown in the Supplementary Information in Fig. 2d to illustrate that the large PomX structures are scarce. We have also included new experiments in which we use negative stain transmission electron microscopy to visualize the PomX filaments in the absence and presence of PEG8000 (Fig. 2e). These TEM images clearly show the bundling effect of PEG8000 on the PomX filaments. Moreover, we now mention in the revised text (line 227) “These structures were scarce, and thus did not cause a marked increase in turbidity”. Finally, we have included a quantification of the bundling effect of PEG8000 by determining the coefficient of variation (CV) of the fluorescence intensity in the images. This analysis indeed showed that the CV increased with the PEG8000 concentrations (Supplementary Fig. 5d, e) (line 228-231).

Concerning the nomenclature for the structures formed by PomX, we decided to use filaments and large filamentous structures of bundled filaments.

6. The BACTH data (ED Fig. 7) are confusing. Full length PomY does not seem to interact with itself, raising a question of how it could make a condensate. Also, a couple of the domains do not interact with full length Y. Moreover, to make a condensate, one needs more than pairwise interactions. If the CC and IDR of one molecule could both bind to the CC and IDR of another, this would just give a dimer. This needs some thought.

Response: We apologize for this confusion. In the BACTH experiments (which are now in Supplementary Fig. 7), we observe that full-length PomY self-interacts. Moreover, we observe that each of the three parts (PomY^{CC}, PomY^{Heat} and PomY^{IDR}) of PomY self-interacts. Thus, we conclude that not only does full-length PomY self-interact, but also each of the three parts of PomY self-interacts making PomY a multivalent protein. We also observed that full-length PomY interacts with PomY^{CC} and PomY^{Heat}. However, we did not detect an interaction between full-length PomY and PomY^{IDR}. Because this is a negative result, it does not allow us to draw conclusions. Based on the positive results, we conclude that each of the three parts (PomY^{CC}, PomY^{Heat} and PomY^{IDR}) of PomY self-interacts and that PomY is a multivalent protein. This conclusion together with all the observations that PomY phase separates to form condensates with liquid-like properties suggest that the interactions between the three parts of PomY are of low affinity, allowing the formation of a network of interacting PomY molecules without a precise stoichiometry. To illustrate the last point, we have included a new schematic in Supplementary Fig. 7 (described in line 285-286) to exemplify how the PomY multivalency can lead to a network of interacting proteins without a precise stoichiometry.

7. Fig. 3 is really complicated. In vivo X forms spherical clusters on its own, and Y can join these clusters if present. In vivo X forms irregular fibrils on its own, and Y forms large spherical clusters. When mixed (in 2% PEG) the large Y clusters disappear, and are redistributed, at least somewhat, as small spherical aggregates on the X fibrils. I don't see that this in vitro phenomenon is related to the very different structures in vivo.

Response: Again, we apologize for this confusion and for not being clearer. And, we agree with the reviewer that Fig. 3 (now Fig. 4) was very hard to understand. Inspired by the reviewer's comment and to improve the presentation of the results in Fig. 3 (now Fig. 4), we introduced several changes and additions.

1. In Fig. 4b, we added zoomed in images of PomX-A488 structures for direct comparison, which demonstrate that PomX alone forms large structures of bundled filaments as the PEG8000 increases (see also comment #5).

2. In Fig. 4c, we included a new quantification in which we determine the coefficient of variation (CV) of PomX-A488 fluorescence intensity in the images. This analysis demonstrates that PEG8000 and PomY have an additive effect especially at lower PEG8000 concentrations on bundling of the PomX filaments.

3. In Fig. 4d, we included new TEM experiments. These experiments together with the TEM images in Fig. 2e, also show that PEG8000 and PomY have an additive effect on bundling of the PomX filaments.

Based on the results in Fig. 4b-d, we conclude (line 309-310) "...PEG8000 and PomY-mCh both have a bundling effect on PomX filaments, and that PomY-mCh is enriched on and coats these bundled filaments independently of PEG8000".

4. In Fig. 4f, we included new FRAP experiments of PomY coating PomX bundles and find that PomY wetting PomX has liquid-like properties (as *in vivo*).

Based on the results in Fig. 4e, f, we conclude (line 321-323) "Above the bulk C_{sat} , PomY condensates form in solution and associate with and wet the PomX filament bundles while maintaining their liquid-like properties".

We also reworked the main text for clarity (line 288-323). We hope that the reworked text, the new experiments and the reworked Fig. 4 more efficiently convey our data.

8. Fig. 4 shows a clever and important result. Y cannot make clusters without X at its normal expression level, but when overexpressed it can. Moreover, the Y cluster could apparently attract and localize FtsZ and achieve division at that site. This is an important new result, but I am not convinced by the anecdotal data shown. In particular, I wonder how frequently the Y clusters lead to division. Two examples of division at Y are shown in Fig. 4d, however these are at very weak Y clusters. Two examples of strong Y clusters in Fig. 4c do not seem to lead to division. Moreover, X-minus cells still divide at a reduced frequency in the complete absence of Y clusters (at the normal expression levels). I think we need a large survey with solid statistics to make a convincing case that the division site is localized at Y clusters.

Response: Thank you for this suggestion. To address this question, we performed new time-lapse microscopy experiments in which we followed cells that overaccumulate PomY-mCh in the absence of PomX. In four independent experiments, we found that in 49 out of 50 division events, cell division occurred over the PomY-mCh condensate, providing quantitative evidence that cell division site positioning solely depends on the presence of PomY-mCh condensate. We included this quantification Fig. 5e (previously Fig. 4). Moreover, we included additional images to illustrate cell divisions over PomY-mCh condensates in the absence of PomX in Fig. 5d and Supplementary 8e. Together these images show that these cell divisions can occur over "weakly" as well as "strongly" fluorescent clusters. The reviewer is correct that the $\Delta pomX$ mutant cells still divide, albeit much less frequently than WT.

We would also like to add that we do not know what determines which PomY condensate (when PomY is overexpressed in the absence of PomX) eventually determines the site of cell division. Similarly, we do not know whether these cells divide as frequently as WT cells. We will address these two questions in future experiments. Importantly, the answers to these two questions, are not essential for the take-home-message of the experiments in Fig. 5, i.e. "cell division site positioning solely depends on the presence of a PomY condensate, irrespectively of the presence of PomX" (line 346-347).

9. “cell division site positioning solely depends on the presence of a PomY condensate, irrespectively of the presence of PomX.” As noted above, this broad interpretation needs extensive counting and statistics. From the present images it is not convincing.

Response: Please see our comments to comment 8.

10. L 1309-10: “PomY^{heat} and PomY^{idr} stimulate phase separation.” Y-heat may show a slight stimulation based on the numbers in Fig. 5b (although the single image looks less than cc); cc-idr does not seem to be tested. I guess this conclusion for idr may be based on wt being stronger than cc-heat; this should be spelled out.

Response: The reviewer is correct in that the conclusion of PomY^{IDR} stimulating phase separation is based on comparing PomY-mCh to the PomY^{CC-HEAT}-mCh variant.

Indeed while at 4% PEG8000 the phase separation of the wildtype PomY-mCh is clearly much increased as compared to PomY^{CC-HEAT}-mCh the situation is reversed at 2% PEG8000. The statement that the HEAT domain stimulates phase separation is based on the ability of PomY^{CC-HEAT}-mCh to phase separate at lower PEG concentrations than PomY^{CC}-mCh alone.

Unfortunately, we have not been able to reliably purify the PomY^{CC-IDR} variant that the reviewer mentioned. We agree that our data do not support the statement that the PomY^{HEAT} and the PomY^{IDR} domains stimulate phase separation, but they do support that these domains alter the phase separation behavior. We have spelled this out more clearly in line 380-386 and changed stimulate to modulate (line 385).

Please also note that trying to compare the small snapshots displayed in Fig. 6b (previously Fig. 5b) is risky. We provide these small snapshots so that the morphology of the condensates is recognizable. The quantification takes into account intensity and total number of pixels classified as condensates and is based on three independent experimental repeats in which 4 z-stacks were acquired, each of a much larger field of view than the one of the displayed images. This quantification is thus a better indicator.

11. Fig. 6 shows that FtsZ is enriched in Y clusters, but surprisingly Strep was also enriched. FtsZ was 12x, while Strep was lower at 3x. Still, this suggests a lack of specificity in the recruitment, which may be useful to note. (Minor suggestion: In 6a the top panel is labeled Protein, which suggested to me that it may be measuring total protein. It would be better to label it A488.).

Response: Many thanks for pointing this out to us. In response to this comment, we have now significantly expanded on this experiment (see new Fig. 7) and reworked the main text (line 431-457). The changes include

1. an improved explanation of the methods.
2. the addition of EGFP as a second control protein to test its enrichment in PomY-mCh condensates.
3. a clear description of the degree of labeling of the two Alexa488-labeled proteins (A488-FtsZ and Strep-A488) to demonstrate that it is not the A488 label that causes the enrichment of A488-FtsZ in the PomY-mCh condensates.
4. a normalization to the enrichment of the sample that did not contain any client protein to demonstrate that the bleedthrough between the fluorescent channels is minor.
5. a quantification of the heterogeneity of the A488-FtsZ fluorescence in the condensates by calculating the coefficient of variation of all three client proteins' fluorescence in the dense

phase (Fig. 7e). We speculate that strong interactions between FtsZ and PomY cause this heterogeneity.

We believe that with these changes we provide very strong arguments that the partitioning of A488-FtsZ into the PomY-mCh condensates in vitro is indeed specific. Please also note the response to reviewer 2's comment 4.

12. Interpreting Fig. 6d the authors state: "Remarkably, filamentous FtsZ structures emerged from PomY-mCh condensates within minutes, later on forming an extended filamentous network." This is a very important conclusion, central to the study, but I am not convinced. In the time point at 11 min, the majority of the FtsZ bundles seem to arise spontaneously in the solution, not from Y clusters. This is even more obvious in the early stages of video 3. I see no hint that the Y clusters nucleate the bundles. There is only a vague hint that they attach to bundles when they contact, but that would have nothing to do with the interpretation that Y clusters are involved in nucleating FtsZ.

Response: Many thanks for pointing this out to us. In response to this comment, we have now significantly expanded on the effect of PomY on FtsZ polymerization. First, we separated the original Fig. 6 into two figures. In the new Fig. 7, we show that A488-FtsZ is enriched in the PomY condensates. In the new Fig. 8, we focus on the effect of PomY on FtsZ polymerization by including GTP in the experiments. We also completely reworked the main text (line 460-508). By including these changes, we now provide extensive additional evidence that PomY condensates stimulate GTP-dependent FtsZ filament formation and bundling at concentrations that are too low for the unaided formation of FtsZ filaments. Specifically, we included the following changes:

1. The addition of new TEM experiments in which we first show that (line 464-469) "A488-FtsZ incubated alone without or with PEG8000 occasionally formed a very small number of filamentous structures (Fig. 8b); thus, under these conditions, 1 μ M A488-FtsZ and 0.5 mM GTP, FtsZ does not efficiently polymerize by itself. Similarly, in the presence of PomY-mCh, but without PEG8000, 1 μ M A488-FtsZ also only formed a very small number of filamentous structures (Fig. 8b)".

In the second part of the TEM experiments, we show that (line 469-473) "in the presence of PomY-mCh and PEG8000, i.e. under conditions where PomY-mCh forms condensates, A488-FtsZ filaments and filament bundles were highly abundant forming an extended network (Fig. 8b). These filaments and filament bundles were associated with protein-dense spherical and lenticular structures that tended to elongate along the A488-FtsZ structures". Thus, A488-FtsZ only polymerizes under conditions where PomY forms condensates.

2. Following up on the TEM experiments, we performed additional snapshot fluorescence microscopy and confirmed that (line 475-477) "A488-FtsZ formed an extensive network of filamentous structures only in the presence of both PEG8000 and PomY-mCh, but not in the absence of either PomY-mCh or PEG8000 (Fig. 8c)". We also confirmed that A488-FtsZ was enriched in the PomY-mCh condensates (Fig. 8c), that (line 478-480) "the entire network of FtsZ-A488 structures was covered by PomY-mCh (Fig. 8c), and that "the PomY-mCh condensates were often deformed into lenticular shapes (Fig. 8c), suggesting that the spherical to lenticular shaped protein-dense structures observed by TEM (Fig. 8b) are PomY-mCh condensates".

From these two sets of experiments, we conclude that neither PomY-mCh nor PEG8000 alone is sufficient for the formation of the large-scale filamentous A488-FtsZ network. Therefore, these two different methods support that PomY-mCh condensates stimulate FtsZ polymerization into filaments and further bundle these filaments.

3. Having established that FtsZ polymerization into filaments and filament bundling only occurs under conditions where PomY forms condensates, we performed additional time-lapse fluorescence microscopy experiments at high temporal resolution to follow the formation of A488-FtsZ filaments and bundles. The results from these new experiments are included in Fig. 8f, g and the new movies in Supplementary Video 3. As described in line 489-495, we observed that “Filamentous A488-FtsZ structures formed in solution sedimented into the focal plane and associated with PomY condensates (Fig. 8f, Supplementary Fig. 14b). Importantly, these A488-FtsZ filament bundles were coated with PomY-mCh and/or connected to PomY-mCh condensates. Similarly, within minutes A488-FtsZ filament bundles emerged from PomY-mCh condensates, which were deformed into lenticular shapes in this process (Fig. 8g, Supplementary Fig. 14c, Supplementary Video 3). These A488-FtsZ filament bundles were also coated with PomY-mCh (Fig. 8f)”. We also find in time-lapse recordings of larger field of views (Video 4) that (line 499-503) “the extensive filamentous A488-FtsZ network was generated over time from three types of interactions: (1) A488-FtsZ filament bundles emerging from condensates, (2) fusion and alignment of A488-FtsZ filament bundles, and (3) A488-FtsZ filament bundles associating with a condensate (Fig. 8h, Supplementary Fig. 14d, Supplementary Video 3 and 4)”.

Based on all the experiments in Fig. 7 and Fig. 8 and the new videos, we conclude (line 504-508) “PomY condensates are enrichment centers for FtsZ and drive local FtsZ filament nucleation in the presence of GTP. In this process, PomY condensates are deformed and PomY associates with the FtsZ filaments, thereby bundling these filaments. In this way, PomY condensates stimulate GTP-dependent FtsZ filament formation and bundling at concentrations that are too low for the unaided formation of FtsZ filaments”.

13. Note also that these beautiful time lapse movies are on a much slower scale than FtsZ nucleation. FtsZ protofilaments reach a plateau of assembly within ~10 s of nucleation. What these LM movies are showing is bundling of these protofilaments, which progresses much more slowly than nucleation and initial assembly of protofilaments.

Response: Many thanks for pointing this out to us. In the revised manuscript, we have included new TEM experiments. The conditions for TEM and the fluorescence microscopy of A488-FtsZ/PomY-mCh are identical, i.e. FtsZ at 1 μ M and GTP at 0.5mM final concentrations. Our new TEM data demonstrate that under these conditions, FtsZ only forms very few filaments under -PEG8000, +PEG8000 and -PEG8000/+PomY (Fig. 8b). By TEM, FtsZ filaments and bundles are only observed under +PEG8000/+PomY conditions (Fig. 8b). Moreover, in the new time-lapse recordings, it is much more evident that PomY-mCh condensates drive local FtsZ filament nucleation in the presence of GTP. In this process, PomY condensates are deformed and PomY associates with the FtsZ filaments, thereby bundling these filaments. Altogether, we believe that our new data provide very strong support that PomY-mCh condensates specifically enrich FtsZ and thus localizes the nucleation of FtsZ filaments. FtsZ filaments that emerge from PomY condensates are coated with PomY-mCh or are connected to PomY-mCh condensates. We now discuss these points in the revised main text to Fig. 8 (line 460-508).

We would also like to mention that under the conditions of the time-lapse recordings, FtsZ polymerization into protofilaments should take much longer than 10sec. For example, using 2 μ M *Caulobacter* FtsZ and 2mM GTP and a similar chamber geometry as used in our experiments, polymer length as assessed by FCS only reached a steady state after 30-40min (Corrales-Guerrero *et al.*, PNAS 2022). Similarly, *Caulobacter* FtsZ polymerization assessed by DLS using 5 μ M FtsZ and 1mM GTP showed that the steady state of polymer length was also only reached after 10min.

14. My observation that FtsZ bundles appear mostly outside Y clusters seems contradicted by Fig. 6e, which shows that no FtsZ bundles appear in the absence of Y. However, this probably means only that soluble Y mediates FtsZ bundling. In the absence of Y, FtsZ assembles protofilaments, which are not resolved by LM. In the presence of Y, the protofilaments are slowly clustered into discrete bundles, which are resolved by LM. This should be tested by repeating the assay with lower concentrations of Y, going below the C_{eff} for cluster formation. It would also be interesting to test the effect of PEG, lowering it to below cluster formation.

Response: Many thanks for pointing this out to us. To clearly determine whether 1 μ M A488-FtsZ forms filament in the presence of GTP, we performed new TEM experiments (Fig. 8b). Our new TEM data demonstrate that A488-FtsZ only forms very few filaments under -PEG8000, +PEG8000 and -PEG8000/+PomY conditions (Fig. 8b). By TEM, extensive A488-FtsZ filaments and bundles are only observed under +PEG8000/+PomY conditions (Fig. 8b). Thus, PomY in the absence PEG8000 does not efficiently stimulate bundling of FtsZ filaments.

Moreover, for the snapshot fluorescence microscopy images, we have now included maximum intensity projections and a quantification of their average A488-FtsZ fluorescence intensity for samples containing GTP, FtsZ and (1) PEG8000, (2) PomY-mCh in absence of PEG8000, and (3) PomY-mCh in presence of PEG8000 (Fig. 8c-e). While a few small filament bundles are observed for cases (1) and (2), only when PomY-mCh phase separates in presence of PEG8000 are large FtsZ filament bundle networks formed. Importantly, in the resulting filament networks, all FtsZ bundles are connected to other PomY-mCh coated filament bundles or PomY-mCh condensates. This strongly suggests that only the highly concentrated FtsZ within PomY-mCh condensates is efficiently polymerizing and is also bundled by PomY-mCh. See also the answer to the comment below and above.

Finally, we would like to add that we did not follow the advice of the reviewer to look at FtsZ filament formation at lower PomY concentrations because even at 4 μ M, PomY does not stimulate FtsZ filament formation in the absence of PEG8000.

15. In previous papers the Sogaard-Andersen lab has presented high quality negative stain EM. This would be an ideal tool to study the bundling as a function of Y, Y-cc, and PEG concentrations. EM might also reveal details of the substructure of Y and Y-cc clusters.

Response: Many thanks for these suggestions. As you will see in our answers to comments #12-14, we performed new TEM experiments (Fig. 8b) in which we analyze A488-FtsZ filament formation and bundling in the presence of GTP under different conditions. In the TEM and fluorescence microscopy experiments, extensive FtsZ networks of filaments and bundled filaments are only formed in presence of PomY-mCh condensates formed in the presence of PEG8000.

Please note that we also included new TEM experiments to investigate the effect of (1) PEG8000 on the PomX filaments (Fig. 2e), (2) PEG8000 on PomY (Fig. 3g) and (3) PEG8000+PomY on PomX filaments (Fig. 4d).

While we did not study PomY^{CC} by TEM, we added a new experiments in Fig. 6h, in which we demonstrate that the two phase separating PomY variants (PomY^{CC-HEAT-mCh} and PomY-mCh) coated the PomX-A647 filament bundles and also formed condensates that localized on the PomX-A647 bundles, while the non-phase-separating PomY^{HEAT-IDR-mCh} variant simply coated the PomX structures but did not form condensates on them.

Reviewer #2 (Remarks to the Author):

In this manuscript, Ramm and colleagues investigated how the PomX/Y/Z complex directs the formation of the Z-ring, a structure formed through the polymerization of FtsZ for bacterial division. Using *Myxococcus xanthus* as a model organism, they found that the cluster of PomY, but not of PomX, dynamically exchanged molecules with the cytoplasm and was assembled before and disassembled after the cell division. In the presence of a crowding reagent such as PEG8000, purified PomX polymerized into filaments, whereas PomY phase-separated into liquid droplets. The filaments concentrated PomY and induced deformation of PomY condensates. The CC region mediated the LLPS of PomY, whereas the HEAT and IDR regions promoted the LLPS though homotypic interactions. All the regions contributed to the full interaction affinity with PomX. PomY condensates formed in PomX-depleted cells through overexpression was sufficient to induce cell division. Furthermore, PomY condensates enriched FtsZ and promoted its GTP-dependent polymerization in vitro. The authors thus proposed a model in which the cluster of PomX filaments in cells enriched PomY and induced its local phase separation to form a single condensate, which further recruited FtsZ and promoted its polymerization to form the Z-ring.

While the authors proposed an interesting model for the mechanism of Z-ring formation from the angle of LLPS, experimental results still do not sufficiently support their claims due to lack of controls, solid evidence, or detailed information. I do not suggest publication of the manuscript in its current form, unless the concerns listed below can be properly addressed.

Major concerns:

1. In the images presented in Figure 3a and 3b/Video 2, fluorescent signals of the A488 and mCh channels superimposed nicely at fibrous structures and condensates. While this may reflect an extensive interplay between PomX and PomY, it might be due to a leak of fluorescent signals as well, unless the latter possibility is excluded. I would suggest that the authors demonstrate that the signals are not interdependent by bleaching A488 and mCh, respectively, in different small regions of fibers/condensates, followed by dual-color imaging of the entire field. Alternatively, mCh could be mixed with PomX-A488 fibers and an irrelevant A488-labelled protein with PomY-mCh condensates, followed by imaging. If leaking is not a problem, they might want to clarify whether PomY droplets concentrated PomX to in turn facilitate PomX polymerization. In Video 2, PomX filaments appear to display enhanced polymerization after contacting with droplets.

Response: Throughout our experiments, care was taken to minimize bleedthrough. All images were acquired using alternating excitation. Furthermore, all images for samples with only PomX or PomY and when mixed together were acquired at the same excitation and emission setting, and laser power was adjusted in a way that bleedthrough was minimized on the respective

sample that did not contain the other protein. Bleedthrough normally occurs from the channel with the lower excitation wavelength (in this case PomX-A488) into the channel with the higher excitation wavelength (in this case PomY-mCh). However, as PomY-mCh condensates are very dense and thus highly fluorescent, the laser power for PomY-mCh was set very low, an order of magnitude lower than that of the other one. Even though we know that laser powers cannot be compared between different excitations, this is just to give an impression.

To show that the bleedthrough is indeed minimized, please see below an experiment where we mixed 4 μ M PomX and 4 μ M PomY at 4% PEG8000. We did this with both proteins labeled (PomY-mCh, PomX-A488) and only one of them labeled (PomY-mCh, PomX) and (PomY, PomX-A488). Please also note the extremely different Brightness and Contrast settings, showing that the only bleedthrough occurring is from PomY-mCh into the 488 channel, and that this is at a very different scale than the actual fluorescent signal (Brightness 0-0.05 compared to 0-20).

Inspired and motivated by the comment “clarify whether PomY droplets concentrated PomX to in turn facilitate PomX polymerization. In Video 2, PomX filaments appear to display enhanced polymerization after contacting with droplets”, we did the following:

1. Quantified the effect of PEG8000 on the formation of bundled PomX filaments by determining the coefficient of variation of the fluorescence intensity in the images with increasing PomX and PEG8000 concentrations (Supplementary Fig. 5d, e). This analysis demonstrated that PEG8000 stimulates bundling of the PomX filaments (line 228-231).
2. Included TEM analyses of the structures formed by PomX with and without PEG8000 (Fig. 2e) and again found that PEG8000 stimulates bundling of the PomX filaments (line 232-237).
3. Quantified the joint effect of PomY and PEG8000 on the formation of bundled PomX filaments by determining the coefficient of variation of the fluorescence intensity in the images with increasing PomX and PEG8000 concentrations (Fig. 4b) and found that PomY-mCh and PEG8000 have an additive effect especially at lower PEG8000 concentrations on bundling of the PomX filaments (line 301-303). These filament bundles also give the appearance of being more dense than those formed in the presence of either PomY or PEG8000. Because PomX spontaneously polymerizes to form thin filaments *in vitro* and forms a cluster independently of PomY *in vivo* (Schumacher et al., 2017, 2021), we decided not to pursue this observation in the context of the work presented here.
4. Included TEM analyses of the structures formed by PomX in the presence of PomY and with or without PEG8000 (Fig. 4c) and again found that PEG8000 stimulates bundling of the PomX filaments (line 303-308).

Thus, by both methodologies, PEG8000 as well as PomY have a bundling effect on PomX filaments, and together they have an additive effect especially at lower PEG8000 concentrations

on bundling of the PomX filaments. We would like to add that PomX spontaneously polymerizes to form thin filaments in vitro and forms a cluster independently of PomY in vivo (Schumacher et al., 2017, 2021). So, we believe that the only effect PEG8000 and PomY has is on bundling of the filaments (and not in stimulating filament formation).

Finally, concerning Fig. 4e & video 2: We have included in the legend that “Note that in this recording, the PomY-mCh condensates do not stimulate PomX-A488 filament bundle formation, instead the PomX-A488 filament bundle sediments from the solution into the focal plane for imaging”.

2. In a previous publication (PMID: 28486132), the group shows through TEM that PomX alone forms 8.3-nm-thick filaments, which are further bundled into 150-nm ones in the presence of PomY. Fluorescent images in Figure 3a indicate that the fibrous structures formed with PomX and PomX+PomY in the presence of PEG differ markedly from those in the absence of PEG. I would suggest that the authors at least examine the samples of 1% PEG with TEM to clarify how the induced phase separation impacts the filamentous structures.

Response: We thank the reviewer for this suggestion. Please see the last part of our answer to comment #1.

3. Figure 4d: the conclusion that PomY condensates are functional and support cell division in the absence of PomX is drawn without the support of quantification results. Could the authors show how tightly the cell division event is correlated with PomY punctum? What happens in cells with multiple puncta?

Response: Thank you for this suggestion. To address this question, we performed new time-lapse microscopy experiments in which we followed cells that overaccumulate PomY-mCh in the absence of PomX. In four independent experiments, we found that in 49 out of 50 division events, cell division occurred over the PomY-mCh condensate, providing quantitative evidence that cell division site positioning solely depends on the presence of PomY-mCh condensate. We included this quantification in Fig. 5e (previously Fig. 4). Moreover, we included additional images to illustrate cell divisions over PomY-mCh condensates in the absence of PomX in Fig. 5d and Supplementary 8e. Together these images show that these cell divisions can occur over “weakly” as well as “strongly” fluorescent clusters. The reviewer is correct that the $\Delta pomY$ mutant cells still divide, albeit much less frequently than WT.

We would also like to add that we do not know what determines which PomY condensate (when PomY is overexpressed in the absence of PomX) eventually determines the site of cell division. Similarly, we do not know whether these cells divide as frequently as WT cells. We will address these two questions in future experiments. Importantly, the answers to these two questions, are not essential for the take-home-message of the experiments in Fig. 5, i.e. “cell division site positioning solely depends on the presence of a PomY condensate, irrespectively of the presence of PomX” (line 346-347).

4. The enrichment of Strep-A488 into PomY condensates (Figure 6a) raises the question whether the enrichment of A488-FtsZ is specific, which needs to be further clarified. As pointed out earlier, fluorescence leaking might be a problem. If this possibility is excluded, I would suggest that the authors clarify whether the observed partitioning was due to the A488 label. Condensates usually display selectivity on proteins that can be partitioned in them. Using FtsZ and a control protein tagged with a fluorescent protein, e.g., GFP, would help to clarify the

issue. Although the authors managed to show less enrichment of Strep-A488 (Figure 6b), I am not convinced by the quantification results. In the images in Figure 6a, I do not see that Strep-A488 is much less enriched in the droplets than A488-FtsZ, as implicated in Figure 6b. I also don't understand why the authors suddenly chose to use selected optical sections, instead of maximum intensity-projected images, and a different calculation method to measure the partition efficiency of the two proteins (Figure 6b), while measuring the usual separation factor to show the partition efficiency of PomY-mCh (Figure 6c). Without a clear demonstration of the partition specificity, the polymerization results of FtsZ (Figure 6d-g) could just be artifacts of its non-specific enrichment.

Response: As in the experiments with fluorescently labelled PomX and PomY, we took extreme care in minimizing bleedthrough when quantifying the enrichment of molecules in PomY-mCh condensates. Again, images were acquired with alternating excitation and samples that only contained PomY-mCh were imaged concomitantly in every experiment. Laser power was adjusted to obtain minimal bleedthrough, but still a high enough dynamic range on the other samples to allow for a meaningful quantification of the enrichment.

Unfortunately, we could not use FtsZ-GFP for our analyses because this protein under all conditions tested precipitates *in vitro*. We thus generated a cysteine-tagged FtsZ, that is similarly GTPase active as the wildtype protein (see Fig. S12) and that we labeled with Alexa488 for our fluorescence microscopy experiments. As chemical dyes are known to respond to the nature of their chemical environment, we reasoned that the best control would be an inert protein labeled with the same chemical dye, in this case Streptavidin (A488-Strep). We have now also included new experiments with EGFP (Fig. 7b-e). We have also included the information about the degree of labeling of A488-FtsZ and Strep-A488 in the figure legend. The commercially available Streptavidin is labeled with 19 Alexa 488 molecules/protein whereas our own labeled FtsZ only contains 0.5 Alexa 488/protein on average. Hence, if the dye were to change its fluorescence inside the condensates or alter the enrichment of proteins in the condensates this effect should be much stronger for Strep-A488. We now also normalize the enrichment to the respective bleedthrough control in which we did not include any protein, but still acquired images in the 488 channel. The bleedthrough control images were acquired at the same laser settings, and it was very low, as the calculated enrichment was close to the expected value of 1 (1.7 ± 0.5) for equal distribution between background and condensates (see explanation below). Based on these quantifications, we conclude that the observed enrichment of A488-FtsZ is not due to the A488 label (line 450-451).

Regarding the distinct quantification measures used in this study: The separation factor is the total integrated intensity of the condensates divided by the total intensity of the image. This quantifies the total amount of fluorescent molecules that are located in the condensed phase and is thus a good measure to quantify the amount of phase separation in a given sample. The enrichment on the other hand, is the average intensity of the condensate divided by the average intensity of the background. This does not take into account how many condensates are in the sample and so samples with only very few condensates can have a very similar enrichment as a sample with many condensates of a similar intensity. For quantification of the separation factor of the experiments that only contained PomY-mCh, we used the maximum intensity projection of the images as we only acquired one fluorescence channel which increased imaging speed significantly. Thus condensates did not move much, and if any, only few additional condensates sedimented during acquisition of the z-stack. In this case one could of course also use the slice that is best in focus, which would yield very similar results.

In the case of quantifying the enrichment of client proteins inside the PomY-mCh condensates we chose to only analyze a single slice, as acquiring the z-stack using alternating excitation took a long time. During this acquisition, condensates moved and additional condensates sedimented into the field of view. Hence, in the maximum intensity projection of such z-stacks the position and fluorescence intensity of a given condensate in the two channels will be less accurate than with a single slice.

We also chose enrichment as a quantification measure because in this case we are only interested in the intensity of the condensates and not in the number of condensates. (The number of condensates is very similar as shown by a very similar separation factor in the PomY-mCh channel Fig. 7d). Given that in this experiment condensation occurs in all samples, the separation factor is not a meaningful measure. Assume a protein that is equally distributed in the sample and has the same fluorescence in and outside the condensate, it would thus be considered as not enriched in the condensates. However, as we obtain the condensate location and number from segmenting the PomY-mCh channel, the separation factor would not be 0 but lie somewhere between 0 and 1 depending on the number of condensates segmented. In contrast, the enrichment is a more meaningful measure: it is 1 for a protein that has the same distribution inside and outside the condensate. Due to the normalization to the bleedthrough control it is 0 in our case.

5. According to Video 3 (Figure 6d), FtsZ polymerizations appear to occur from short pre-existing seeds but not from the condensates as described by the authors (line 405). If PomY condensates served as nucleation centers (line 418), one would expect that the condensates produce radial arrays of FtsZ bundles, analogous to the situation of abLIM1 condensates on actin (PMID: 35858327). Therefore, PomY may only bundle spontaneously-polymerized FtsZ filaments, which were too small for light microscopy to reveal. EM examinations on samples as those in Figure 6e will clarify whether FtsZ polymerized autonomously in the absence of PomY and how the bundles formed in the presence of PomY look like. Light scattering assays (e.g., PMID: 9922245) would also be very informative to understand how PomY impacts FtsZ polymerization. A nucleator should usually induce polymerization at FtsZ concentrations below the critical concentration of spontaneous polymerization. In addition, as PomY was gradually recruited to the elongating FtsZ bundles following the polymerization process (Video 3), its colocalization with FtsZ bundles does not necessarily underscore the liquid nature of PomY condensates (line 415). The authors need to tone down this or provide evidence that the bundle-associated PomY molecules possess liquid properties, e.g., by monitoring the behavior of photoconverted PomY fusion proteins like demonstrated in (PMID: 35858327).

Response: Many thanks for pointing this out to us. In response to this comment and reviewer 1's concerns, we have now significantly expanded on the effect of PomY on FtsZ polymerization. First, we separated the original Fig. 6 into two figures. In the new Fig. 7, we show that A488-FtsZ is enriched in the PomY condensates. In the new Fig. 8, we focus on the effect of PomY on FtsZ polymerization by including GTP in the experiments. We also completely reworked the main text to these two figures. By including these changes, we now provide extensive additional evidence that PomY condensates stimulate GTP-dependent FtsZ filament formation and bundling at concentrations that are too low for the unaided formation of FtsZ filaments. Specifically, we included the following changes:

1. The addition of new TEM experiments in which we first show that (line 464-468) "A488-FtsZ incubated alone without or with PEG8000 occasionally formed a very small number of

filamentous structures (Fig. 8b); thus, under these conditions, 1 μ M A488-FtsZ and 0.5 mM GTP, FtsZ does not efficiently polymerize by itself. Similarly, in the presence of PomY-mCh, but without PEG8000, 1 μ M A488-FtsZ also only formed a very small number of filamentous structures (Fig. 8b)”.

In the second part of the TEM experiments, we show that (line 469-473) “in the presence of PomY-mCh and PEG8000, i.e. under conditions where PomY-mCh forms condensates, A488-FtsZ filaments and filament bundles were highly abundant forming an extended network (Fig. 8b). These filaments and filament bundles were associated with protein-dense spherical and lenticular structures that tended to elongate along the A488-FtsZ structures”. Thus, A488-FtsZ only polymerizes under conditions where PomY forms condensates.

2. Following up on the TEM experiments, we performed additional snapshot fluorescence microscopy and confirmed that (line 475-477) “A488-FtsZ formed an extensive network of filamentous structures only in the presence of both PEG8000 and PomY-mCh, but not in the absence of either PomY-mCh or PEG8000 (Fig. 8c)”. We also confirmed that (line 478-485) A488-FtsZ was enriched in the PomY-mCh condensates (Fig. 8c), that “the entire network of FtsZ-A488 structures was covered by PomY-mCh (Fig. 8c), and that “the PomY-mCh condensates were often deformed into lenticular shapes (Fig. 8c), suggesting that the spherical to lenticular shaped protein-dense structures observed by TEM (Fig. 8b) are PomY-mCh condensates”.

From these two sets of experiments, we conclude that neither PomY-mCh nor PEG8000 alone is sufficient for the formation of the large-scale filamentous A488-FtsZ network. Therefore, these two different methods support that PomY-mCh condensates stimulate FtsZ polymerization into filaments and further bundle these filaments.

3. Having established that FtsZ polymerization into filaments and filament bundling only occurs under conditions where PomY forms condensates, we performed additional time-lapse fluorescence microscopy experiments at high temporal resolution to follow the formation of A488-FtsZ filaments and bundles. The results from these new experiments are included in Fig. 8f, g and the new movies in Supplementary Video 3. As described in line 489-496, we observed that “Filamentous A488-FtsZ structures formed in solution sedimented into the focal plane and associated with PomY condensates (Fig. 8f, Supplementary Fig. 14b). Importantly, these A488-FtsZ filament bundles were coated with PomY-mCh and/or connected to PomY-mCh condensates. Similarly, within minutes A488-FtsZ filament bundles emerged from PomY-mCh condensates, which were deformed into lenticular shapes in this process (Fig. 8g, Supplementary Fig. 14c, Supplementary Video 3). These A488-FtsZ filament bundles were also coated with PomY-mCh (Fig. 8f)”. We also find in time-lapse recordings of larger field of views (Video 4) that (line 499-503) “the extensive filamentous A488-FtsZ network was generated over time from three types of interactions: (1) A488-FtsZ filament bundles emerging from condensates, (2) fusion and alignment of A488-FtsZ filament bundles, and (3) A488-FtsZ filament bundles associating with a condensate (Fig. 8h, Supplementary Fig. 14d, Supplementary Video 3 and 4)”.

Based on all the experiments in Fig. 7 and Fig. 8 and the new videos, we conclude (line 504-508) “PomY condensates are enrichment centers for FtsZ and drive local FtsZ filament nucleation in the presence of GTP. In this process, PomY condensates are deformed and PomY associates with the FtsZ filaments, thereby bundling these filaments. In this way, PomY

condensates stimulate GTP-dependent FtsZ filament formation and bundling at concentrations that are too low for the unaided formation of FtsZ filaments”.

Please also note that the conditions for TEM and fluorescence microscopy of A488-FtsZ/PomY-mCh are identical, i.e. FtsZ at 1 μ M and GTP at 0.5mM final concentrations.

Light scattering assays that the reviewer is referring to cannot be performed in the presence of a phase-separating protein as the size and wide size distribution of the condensates would obscure any signal originating from the FtsZ polymers.

We agree with the reviewer that colocalization of PomY-mCh with A488-FtsZ filament bundles does not provide evidence that PomY has liquid-like properties on the filaments. However, in our new high-resolution time-lapse recordings we observe (line 496-498) “A488-FtsZ enriched PomY-mCh condensates also underwent fusion/relaxation events into a spherical shape indicating their liquid-like nature (Fig. 8f, Supplementary Fig. 14b, Supplementary Video 3)”.

6. I would suggest that authors briefly discuss how the PomY condensate disintegrates during or after cytokinesis.

Response: Good question. Our answer is that we do not know. However, in the discussion we comment on possible mechanisms (line 646-656).

Minor concerns:

7. Line 164: “In snapshots of 1000s of cells”: what does 1000s mean here ?

Response: We apologize for this confusion. We rewrote to “In snapshot images of >1000 cells”.

8. Extended Data Figure 3b: full names of the species should be listed instead of abbreviations to increase readability.

Response: Sorry & done.

9. Figure 3a: the white arrow in the PomY channel of the bottommost panel should be either removed or relocated to the merged channel to indicate a condensate.

Response: Thanks & arrow removed (now Fig. 4b).

10. It is not clear whether the initial time point (0 min) of Figure 3b (and video 2) started after the 1-hour incubation (as indicated in the diagram in Figure 3a) or quickly after the mixing of the reagents.

Response: This time-series was indeed started quickly after mixing the proteins together. We have clarified this point in the legend (now Fig. 4e).

11. Figure 4c: one of the two arrows at 60 min does not point to a punctum.

Response: Thanks & arrow corrected (now Fig. 5c).

12. Figure 7: The model does not include the bundling effect of PomY on PomX filaments. The authors might want to consider adding this to the revised manuscript.

Response: Thanks & done (now Fig. 9c)

13. Line 1359: “under the conditions in d and de Scale bar”.

Response: Thanks. This figure panel has been replaced.

Reviewer #3 (Remarks to the Author):

In this well-written manuscript, the Andersen and Schwille groups have collaborated to explore the PomXYZ system, which aids in the positioning of the Z-ring and therefore division in Myxo. Specifically, their focus is on the phase separating properties of PomY and its interactions with PomX and FtsZ filaments. The researchers first characterize the growth and division of PomX and PomY clusters through mCh labeled PomX and PomY, finding that while the PomX cluster divides between daughter cells without dissolution, PomY foci dissolve into the cytoplasm following division. Next the authors demonstrated through in vitro fluorescent imaging and FRAP that PomX forms a cluster of filaments while PomY undergoes phase separation. They go on to show that Csat of PomY can be lowered through PEG8000 induced crowding and/or the presence of PomX. In vivo, they then show that over-expression of PomY allows for condensates to form without PomX that can still position the Z-ring. To characterize the interactions between PomX and PomY, the authors produce PomY domain fragments and show that all three domains bind PomX through pull-down experiments. These PomY fragments were then studied in vitro and in vivo, analyzing which domains and combinations of domains best undergo phase separation. Finally, the authors show PomY condensates can locally induce/stimulate FtsZ polymerization, and that PomY subsequently coats and condensates around the FtsZ bundles. This impressive body of work significantly contributes to our knowledge of the PomXYZ system; primarily through their identification of the phase separation behavior of PomY, its surface-assisted condensation provided by PomX, and the localized induction of FtsZ polymerization both in vivo and in vitro. The authors are successful in effectively introducing the most relevant information about condensates to their study and in contrasting the PomXYZ system from other means of division localization in bacteria. This work is significant as it details a thus-far understudied means of divisome positioning in bacteria. The work's significance extends beyond the immediate scope of bacterial division and has implications on the potential role of phase separation for the regulation of other ParA/MinD ATPase systems. These systems are so widespread that this paper will pave the way for future studies of phase separation roles in bacterial subcellular organization. I think the paper is suitable for publication as is. But there is, as always, a few major (to better justify conclusions made) and several minor suggestions that would improve the manuscript.

Major Issues:

1. The authors claim that while PomY clusters depend on PomX to form, PomX clusters are independent of PomY. Data in columns 1 and 3 in figure 3B slightly conflict with the claim that PomY has no effect on PomX clusters. At 0% PEG8000, PomX mixed with PomY is much more filamentous than PomX alone. Likewise, at 4% and 10% PEG8000, PomX mixed with PomY forms more clumped and aggregated rather than long and thin filaments compared to PomX alone. It is a potentially important finding that PomY may affect PomX filament morphology, the researchers should provide an explanation as to why these PomX morphologies differ.

Response: We thank the reviewer for bringing up this important point. To quantify the effects of PEG8000 and PomY on PomX filaments, we did the following:

1. Quantified the effect of PEG8000 on the formation of bundled PomX filaments by determining the coefficient of variation of the fluorescence intensity in the images with increasing PomX and PEG8000 concentrations (Supplementary Fig. 5d, e). This analysis demonstrated that PEG8000 stimulates bundling of the PomX filaments (line 228-231).

2. Included TEM analyses of the structures formed by PomX with and without PEG8000 (Fig. 2e) and again found that PEG8000 stimulates bundling of the PomX filaments (line 232-237).
3. Quantified the joint effect of PomY and PEG8000 on the formation of bundled PomX filaments by determining the coefficient of variation of the fluorescence intensity in the images with increasing PomX and PEG8000 concentrations (Fig. 4c (previously Fig. 3b)) and found that PomY-mCh and PEG8000 have an additive effect especially at lower PEG8000 concentrations on bundling of the PomX filaments (line 301-303). These filament bundles also give the appearance of being more dense than those formed in the presence of either PomY or PEG8000. Because PomX spontaneously polymerizes to form thin filaments in vitro and forms a cluster independently of PomY in vivo (Schumacher et al., 2017, 2021), we decided not to pursue this observation in the context of the work presented here.
4. Included TEM analyses of the structures formed by PomX in the presence of PomY and with or without PEG8000 (Fig. 4d) and again found that PEG8000 stimulates bundling of the PomX filaments (line 303-308).

Thus, by both methods, PEG8000 as well as PomY have a bundling effect on PomX filaments, and together they have an additive effect especially at lower PEG8000 concentrations. We would like to add that PomX spontaneously polymerizes to form thin filaments in vitro and forms a cluster independently of PomY in vivo (Schumacher et al., 2017, 2021). So, we believe that the only effect PEG8000 and PomY has is on bundling of the filaments (and not in stimulating filament formation).

Finally, concerning Fig. 4e & video 2: We have included in the legend that “Note that in this recording, the PomY-mCh condensates do not stimulate PomX-A488 filament bundle formation, instead the PomX-A488 filament bundle sediments from the solution into the focal plane for imaging”.

2. (Line 311) “Stunningly, not only PomX/PomY-mCh clusters but also the “pure” PomY-mCh clusters were proficient in determining the site of cell division (Fig. 4d), demonstrating that cell division site positioning solely depends on the presence of a PomY condensate, irrespectively of the presence of PomX.” This is a strong statement coming from only two divisions shown in 4C. Please quantify and provide statistics for the locations of pure PomY clusters and the locations of division events across a cell population (similar to what was done in 6h).

Response: We thank the reviewer for pointing this out to us. To address this question, we performed new time-lapse microscopy experiments in which we followed cells that overaccumulate PomY-mCh in the absence of PomX. In four independent experiments, we found that in 49 out of 50 division events, cell division occurred over the PomY-mCh condensate, providing quantitative evidence that cell division site positioning solely depends on the presence of PomY-mCh condensate. We included this quantification Fig. 5e (previously Fig. 4). Moreover, we included additional images to illustrate cell divisions over PomY-mCh condensates in the absence of PomX in Fig. 5d and Supplementary 8e. Together these images show that these cell divisions can occur over “weakly” as well as “strongly” fluorescent clusters. The reviewer is correct that the $\Delta pomX$ mutant cells still divide, albeit much less frequently than WT.

We would also like to add that we do not know what determines which PomY condensate (when PomY is overexpressed in the absence of PomX) eventually determines the site of cell division. Similarly, we do not know whether these cells divide as frequently as WT cells. We will address

these two questions in future experiments. Importantly, the answers to these two questions, are not essential for the take-home-message of the experiments in Fig. 5, i.e. “cell division site positioning solely depends on the presence of a PomY condensate, irrespectively of the presence of PomX” (line 346-347).

3. Based on the data in Fig. 5C, the authors claim that both the IDR and HEAT domains of PomY stimulate phase separation. However, the PomY CC-HEAT fragment alone separates better at PEG8000 2% than the full PomY CC-HEAT-IDR protein. While it's true at 4% PEG8000 the full PomY protein separates more completely, this discrepancy casts doubt on the claim that the IDR region is improving separation. I would argue the IDR assists in fluidizing the condensates and makes them larger due to increasing solvent interactions, as opposed to stimulating condensation (Line 347). Comparisons of droplet fluidity, such as FRAP, could provide insight.

Response: We agree with the reviewer that our data do not support the statement that the PomY^{HEAT} and the PomY^{IDR} domains stimulate phase separation, but they do support that these domains alter the phase separation behavior. We have thus spelled out the respective section more clearly and changed stimulate to modulate (see lines 380-386).

4. I am very excited about the PomY(HEAT-IDR) mutant (Fig 5C-D, specifically). Although the authors don't explicitly mention this in the paper, could it be that this LLPS- mutant still forms the oblong structure in vivo because it is still interacting with PomX (pre-wetting interactions)? If so, do you think this mutant has allowed you to separate pre-wetting interactions from PomY's condensation activity?

Response: Yes, please see the answer to this comment below.

5. (Line 372-376) The claim that LLPS of PomY is necessary for Z ring positioning by the PomXYZ system hinges on the idea that the cluster seen in Figure 5D with PomY [HEAT-IDR-mCh] is not a condensate given its oblong shape. This evidence of a lack of phase separation is weak given that Figure 1B already explored the fact that the PomY condensates form oblong shapes with long and short axes. Perhaps FRAP could be used to show the PomY [HEAT-IDR] cluster lacks fluidity? Or PomX could be mixed with PomY HEAT-IDR in vitro (as in Fig 3A) to show that PomY is associating with PomX, but not phase separating? This would be a nice addition because it shows that with this truncation mutant you have likely separated the initial (pre-wetting) PomX association and surface-assisted condensation activity of PomY.

Response: We thank the reviewer for this suggestion. Inspired by this comment, we included a new *in vitro* experiment in which we mix PomY^{HEAT-IDR}-mCh, PomY^{CC-HEAT}-mCh or wildtype PomY-mCh with PomX at 4% PEG8000 (Fig. 6h, Supplementary Fig. 11). Indeed, we observe (line 414-419) that all three proteins interact with the PomX scaffold, but only PomY^{CC-HEAT} and wildtype PomY-mCh form condensates on the PomX bundles. By contrast, PomY^{HEAT-IDR}-mCh simply coated the PomX structures but did not form condensates (Fig. 6h, Supplementary Fig. 11).

Minor Issues:

1. (Line 42) “Although bacterial cells generally lack organelles....” Protein-based organelles are ubiquitous in bacteria and archaea. Please change to – “Although bacterial cells generally lack membrane-bound organelles.”

Response: Thanks & corrected.

2. (Lines 45-46) The authors should change “LLPS” to just “phase separation” or “condensation”. Many eukaryotic condensates indeed form via phase separation, but many (most) of these papers do not conclusively show that Liq-Liq phase separation is the actual mechanism.

Response: Thanks & done.

3. Why weren't the 6-His tags cleaved off? These tags have been shown in several papers to have drastic effects on all features of condensates (assembly, maintenance, dissolution). The authors are making relative comparisons among protein variants that all have the tags, so I think the results still support the conclusions made. However, I'd be cautious of any definitive statements regarding empirical measurements such as a partitioning coefficient or csat for these proteins.

Response: Throughout the manuscript, our aim is to compare *in vivo* and *in vitro* experiments as closely as possible. The fluorescent protein fusions mCh-PomX and PomY-mCh are active and can fully complement the deletion phenotype when expressed at native levels. In the case of PomX, we used three different proteins to investigate its behavior PomX-His6, mCh-His6-PomX and PomX-His6-Cys and they all behaved similarly *in vitro*. For PomY, we used two variants, i.e. PomY-His6 and PomY-mCh-His6, that also behave similarly. Also, we have previously shown that both His6-PomX and PomY-His6 stimulate the ATPase activity of PomZ *in vitro*, suggesting that they are both functional (Schumacher *et al.*, 2017 & 2021).

4. Figure 1: Do you think PomZ pulling on the PomXY cluster (specifically PomX) is responsible for the elongated PomX structure at mid-cell? For example, if you delete the N-terminal peptide on PomX do you still get an elongated structure? It would be nice to propose a hypothesis for the elongation of the PomX cluster, as one is provided for the elongation of the PomY cluster.

Response: Good question & the answer is that we do not know how the PomX clusters gets its shape. Our data suggest that the PomX cluster *in vivo* is composed of one or several PomX filaments. Incorporation of new proteins into the filamentous structure *in vivo* could lead to an elongation of the filaments and the cluster that we observe before cell division (Fig. 1d). This hypothesis is supported by the finding that overexpression of PomX also results in elongation of PomX clusters into elongated shape (Fig. 1g, and Schumacher *et al.*, 2017). In the future, we plan to investigate the detailed structure of the PomX cluster.

5. Figure 6a: Do you have any hypotheses as to why partitioning of FtsZ into PomY condensates is heterogeneous? It's intriguing. Do you think FtsZ in the PomY condensates is dynamic (liquid-like?) before the addition of GTP?

Response: Thank you for drawing our attention to this heterogeneity. In the revised manuscript, we included an additional experiment in which we quantify this heterogeneity with the coefficient of variation of the intensity of FtsZ and the other client proteins in the PomY condensates (Fig. 7e). Indeed we observe that A488-FtsZ fluorescence in the condensates was more variable than for the control proteins. We speculate that strong interactions between FtsZ and PomY cause this heterogeneity (line 456).

6. Figure 6: Why not add PomX to the *in vitro* reconstitution with PomY and FtsZ? I know you show *in vivo* that overexpressed PomY is necessary and sufficient. But since you have all three labelled and purified, did you try putting all 3 together? Seemed like the obvious thing to do

next.

Response: We have thought about this experiments, but as two filament-forming proteins, PomX and FtsZ, are mixed together with a phase-separating protein the resulting structures may be really hard to interpret. We have thus decided not to do these experiments but plan to try to do them in the future.

7. Do you think overexpression of both PomY and FtsZ would allow for multiple division events in a PomX deletion?

Response: Good question. As alluded to above, we do not know what determines which PomY condensate (when PomY is overexpressed in the absence of PomX) eventually determines the site of cell division. Similarly, we do not know whether these cells divide as frequently as WT cells. We will address these two questions in future experiments.

8. Any reason why two-color labelling of PomX and PomY wasn't performed, as the authors did previously in the 2017 paper? Would have been helpful for several points made throughout this paper, such as co-localization data and pairwise comparisons of PomX-PomY cluster shape. I have the same question for two-color labelling of PomY and FtsZ.

Response: In this manuscript, we focused on expressing the fluorescently tagged PomY and PomX at native levels. Unfortunately, we do not have differently labelled PomX and PomY variants at native levels. We plan to do these experiments in the future.

9. (Line 567) "PomX-PomX" should be "PomX-PomY"

Response: Thanks & corrected.

REVIEWERS' COMMENTS

Reviewer #1 (Remarks to the Author):

The authors have constructively addressed all concerns raised in my original review. I recommend publication.

Reviewer #2 (Remarks to the Author):

The authors have addressed my concerns and I can recommend publication of this manuscript after minor revision: In the abstract, the comma following "social" should be removed; "PomY condensates ...nucleate ... filament bundling" does not appear to be a correct statement.

Reviewer #3 (Remarks to the Author):

The authors have done an excellent job responding to my concerns. I recommend publication.

- Anthony Vecchiarelli